# Unifying VXAI: A Systematic Review and Framework for the Evaluation of Explainable AI

**David Dembinsky**                                    *david.dembinsky@dfki.de*
*German Research Center for Artificial Intelligence (DFKI) GmbH*
*RPTU University Kaiserslautern-Landau, Department of Computer Science*

**Adriano Lucieri**                                    *adriano.lucieri@dfki.de*
*German Research Center for Artificial Intelligence (DFKI) GmbH*
*RPTU University Kaiserslautern-Landau, Department of Computer Science*

**Stanislav Frolov**                                    *stanislav.frolov@dfki.de*
*German Research Center for Artificial Intelligence (DFKI) GmbH*
*RPTU University Kaiserslautern-Landau, Department of Computer Science*

**Hiba Najjar**                                    *hiba.najjar@dfki.de*
*German Research Center for Artificial Intelligence (DFKI) GmbH*
*RPTU University Kaiserslautern-Landau, Department of Computer Science*

**Ko Watanabe**                                    *ko.watanabe@dfki.de*
*German Research Center for Artificial Intelligence (DFKI) GmbH*
*RPTU University Kaiserslautern-Landau, Department of Computer Science*

**Andreas Dengel**                                    *andreas.dengel@dfki.de*
*German Research Center for Artificial Intelligence (DFKI) GmbH*
*RPTU University Kaiserslautern-Landau, Department of Computer Science*

**Reviewed on OpenReview:** *https://openreview.net/forum?id=wAvFLe7oOE*

## Abstract

Modern AI systems frequently rely on opaque black-box models, most notably Deep Neural Networks, whose performance stems from complex architectures with millions of learned parameters. While powerful, their complexity poses a major challenge to trustworthiness, particularly due to a lack of transparency. Explainable AI (XAI) addresses this issue by providing human-understandable explanations of model behavior. However, to ensure their usefulness and trustworthiness, such explanations must be rigorously evaluated. Despite the growing number of XAI methods, the field lacks standardized evaluation protocols and consensus on appropriate metrics. To address this gap, we conduct a systematic literature review following the *Preferred Reporting Items for Systematic Reviews and Meta-Analyses (PRISMA)* guidelines and introduce a unified framework for the *eValuation of XAI (VXAI)*. We identify 362 relevant publications and aggregate their contributions into 41 functionally similar metric groups. In addition, we propose a three-dimensional categorization scheme spanning explanation type, evaluation contextuality, and explanation quality desiderata. Our framework provides the most comprehensive and structured overview of VXAI to date. It supports systematic metric selection, promotes comparability across methods, and offers a flexible foundation for future extensions.

# 1 Introduction

Explainable AI (XAI) is a research area of growing interest to both AI researchers and practitioners. It aims to alleviate the black-box issue of current deep-learning models, which can reach stunning performances at the expense of their interpretability (Vilone & Longo, 2021). Government-affiliated initiatives, such as the European Union High-Level Expert Group on AI (2019), the U.S. National Institute of Standards and Technology (2023), and the DARPA initiative (Gunning & Aha, 2019), identified XAI as a crucial part of Trustworthy AI. Especially as it helps AI systems in serving the "right to explain" its decisions (Goodman & Flaxman, 2017) and fosters user trust through understanding of the system (Morandini et al., 2023). XAI already plays a fundamental role in making high-stakes AI systems more trustworthy (Saarela & Podgorelec, 2024; Xua & Yang, 2024), with broad applications in areas such as healthcare, finance, autonomous driving, natural disaster detection, energy management, military and remote sensing (Adadi & Berrada, 2018; Markus et al., 2021; Saraswat et al., 2022; Kadir et al., 2023; Hosain et al., 2024; Höhl et al., 2024). Furthermore, explainability is used to help with other dimensions of trustworthiness like privacy, robustness, or fairness (Doshi-Velez & Kim, 2017; Yang et al., 2019; Arrieta et al., 2020; Das & Rad, 2020; Markus et al., 2021; Rawal et al., 2021; Agarwal et al., 2022b).

However, XAI is not a silver bullet. Van der Waa et al. (2021) point out that humans tend to trust predictions more readily when an explanation is provided, often without carefully examining the explanation itself. This lack of critical scrutiny can lead to unwarranted trust, especially when decisions are taken based on incorrect or misleading explanations (Eiband et al., 2019; Jesus et al., 2021). To make matters worse, different XAI algorithms may result in conflicting explanations for the same model and sample (Krishna et al., 2022). Therefore, simply providing *any* explanation is not sufficient, but it is important to assess the quality of the explanation at hand (Sovrano et al., 2021). Unfortunately, while there is a plethora of XAI methods, evaluation of explanations is still an immature research area (Ribera & Lapedriza, 2019), with many studies relying on the notion of a good explanation as "You'll know it when you see it", providing anecdotal evidence (i.e. small-scale qualitative validation) (Doshi-Velez & Kim, 2017; Nauta et al., 2023; Saarela & Podgorelec, 2024). Especially in computer vision tasks, evaluation through qualitative inspection of a few examples can be appealing (Ibrahim & Shafiq, 2023). However, unstructured qualitative examination yields highly subjective results, as humans struggle at judging the value of XAI explanations (Adebayo et al., 2018; Buçinca et al., 2020; Hase & Bansal, 2020). In addition, such evaluations risk cherry-picking favorable examples and offer no reliable foundation for comparing different explanation methods across studies or practitioners. For the same explanation, human ratings vary depending on both the task itself (Franklin & Lagnado, 2022) and the participant's cultural background (Peters & Carman, 2024). Evaluation is further complicated by the lack of ground-truth for the explanations, as it requires knowledge about the model's internal reasoning process (Samek et al., 2019; Markus et al., 2021; Samek et al., 2021; Bommer et al., 2024; Ortigossa et al., 2024).

Because evaluation is still not performed consistently and seldom systematically (Adadi & Berrada, 2018; Lipton, 2018; Payrovnaziri et al., 2020; Messina et al., 2022; Lopes et al., 2022; De Camargo et al., 2023; Kadir et al., 2023; Nauta et al., 2023; Mohamed et al., 2024; Naveed et al., 2024; Saarela & Podgorelec, 2024; Salih et al., 2024a), the community frequently calls to develop comprehensive and unified evaluation standards (Pinto & Paquette, 2024; Saarela & Podgorelec, 2024; Xua & Yang, 2024). A central motivation behind such efforts is to enable the comparison of explanations and asses whether explainability is achieved (Markus et al., 2021; Zhou et al., 2021)

One of the most prevalent taxonomies reported in the literature (Vilone & Longo, 2021; Zhou et al., 2021; Elkhawaga et al., 2023), and illustrated in Figure 1, is the distinction proposed by Doshi-Velez & Kim (2017) between human-grounded and functionality-grounded evaluation methods. The former includes qualitative and quantitative evaluations by laypeople and experts, while the latter consists of (semi-)automatic metrics.

Since explanations are meant to aid humans, human-grounded evaluation remains the gold standard to assess their effectiveness in assisting humans (Doshi-Velez & Kim, 2017; Gunning & Aha, 2019; Miller et al., 2017). However, the faithfulness (i.e., technical correctness) of an explanation and the plausibility to humans do not necessarily correlate (Wiegreffe & Pinter, 2019; Jacovi & Goldberg, 2020; Atanasova, 2024a). Therefore, human-grounded evaluation of comprehensibility should be distinguished from functionality-grounded evaluation of faithfulness (Nauta et al., 2023). Especially humans cannot confidently attribute whether an

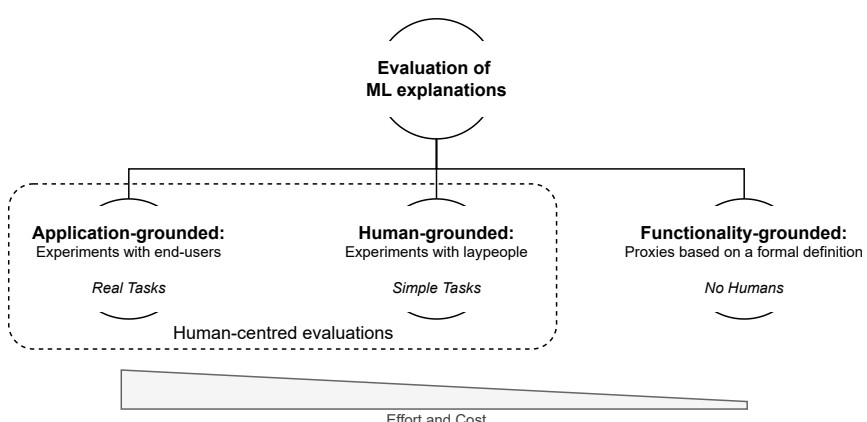

Figure 1: XAI evaluation classified into human-grounded and functionality-grounded evaluation, adapted from the classification framework by Doshi-Velez & Kim (2017) and its visualization by Zhou et al. (2021).

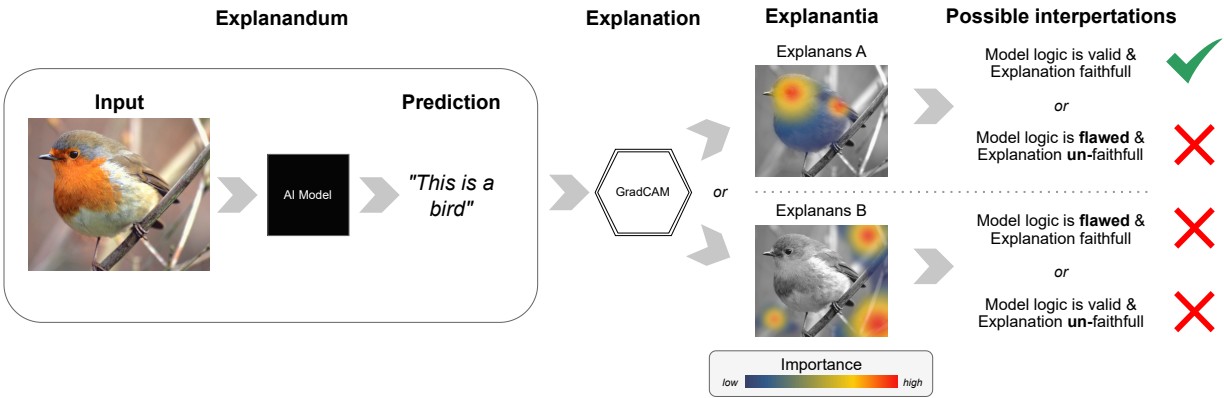

Figure 2: The heatmaps show two alternative example explanantia[1], indicating which input regions were deemed decisive for the model's decision (the explanandum[1]). A qualitative inspection allows for multiple interpretations, as it is unclear whether a) both the model and the explanation process (explanation[1]) are correct or flawed (top), or b) one is correct and the other failed (bottom). The green checkmark marks the only scenario in which a human relying on visual plausibility would arrive at a correct conclusion; the red crosses indicate cases where such qualitative judgment would be misleading.

unexpected explanans (i.e., the information provided to explain a decision[1]) is caused by a faulty explanation (process[1]) or a flawed black-box model (Robnik-Šikonja & Bohanec, 2018; Zhang et al., 2019a); see Figure 2 for an illustration. In both cases, the consequences can be severe, either reducing trust in a well-functioning model or, more critically, reinforcing trust in a flawed one. Further, human evaluation, especially through the system's developers, is prone to confirmation bias (Doshi-Velez & Kim, 2017; Lipton, 2018).

There exist a number of surveys and guidelines that address human-centered evaluations (Hoffman et al., 2018; Miller, 2019; Chromik & Schuessler, 2020; Holzinger et al., 2020; Franklin & Lagnado, 2022; Hsiao et al., 2021; Jesus et al., 2021; Langer et al., 2021; Mohseni et al., 2021; van der Waa et al., 2021; Silva et al., 2023). Unfortunately, the range of reviews dedicated to functionality-grounded evaluation is still limited. This is despite the advantage of offering objective, quantitative metrics without requiring human experiments, which can save both time and cost (Doshi-Velez & Kim, 2017; Samek et al., 2019; Zhou et al., 2021). Most existing studies are narrow and restricted to a specific application domain (Giuste et al., 2022; Arreche et al., 2024), including cybersecurity (Pawlicki et al., 2024), medical image classification (Patrício et al., 2023; Chaddad et al., 2024), electronic health record data (Payrovnaziri et al., 2020), data and knowledge engineering (Li

---

[1]The exact definitions of explanandum, explanation, and explanans are provided at the end of Section 1.

et al., 2020b), or time-series classification (Theissler et al., 2022). Others focus on particular XAI approaches, such as visual explanations in CNNs (Mohamed et al., 2022) or instance-based explanations (Bayrak & Bach, 2024). Moreover, many surveys dedicate only limited attention to evaluation metrics, mainly focusing on the XAI methods themselves (Carvalho et al., 2019; Ding et al., 2022; Minh et al., 2022; Mohamed et al., 2022; Ali et al., 2023; Clement et al., 2023; Patrício et al., 2023; Chaddad et al., 2024; Gongane et al., 2024; Xua & Yang, 2024). By contrast, this review focuses exclusively on functionality-grounded evaluation across domains and is applicable to a wide range of XAI approaches. We focus on both in-hoc and post-hoc explanations for black-box models. Although this work does not consider directly interpretable (white-box) models, many of the presented metrics can also be applied to such models.

### Contributions

Despite the growing number of proposed metrics, a comprehensive and unified framework for functionality-grounded evaluation is still missing. Further, the inconsistent use of terms such as interpretability, comprehensibility, understandability, transparency, and explainability (Koh & Liang, 2017; Guidotti et al., 2018; Arrieta et al., 2020; Markus et al., 2021) hampers conceptual clarity and comparability across approaches. To address this gap, we introduce a framework called **eValuation of Explainable Artificial Intelligence (VXAI)**, aimed at unifying functionality-grounded evaluation for XAI. An interactive version of the framework is available at `https://vxai.dfki.de/`.

Our contributions are as follows:

- We perform a systematic literature review based on the Preferred Reporting Items for Systematic Reviews and Meta-Analyses (PRISMA) guidelines by Page et al. (2021), identifying 362 relevant publications that introduce or utilize evaluation metrics.
- We aggregate these into 41 functionally similar metric groups, capturing common methodological patterns across the literature.
- We propose a three-dimensional categorization scheme consisting of desiderata, explanation type, and evaluation contextuality, and use it to organize the identified metrics.
- To our knowledge, this results in the most comprehensive and unified VXAI framework to date and provides an extensible foundation for future research.

The remainder of this review is structured as follows: In Section 2, we first present related studies on the topic of VXAI to motivate the need for this systematic review. Further, Section 3 outlines our literature research, with full details provided in Appendix A. The VXAI framework is presented in Section 4, where we introduce our new categorization scheme (Subsection 4.1) and summarize the identified metrics (Subsection 4.2), complemented by a visual overview in Figure 5. A deeper discussion of these findings is provided in Section 5, while comprehensive descriptions of the metrics alongside references are listed in Appendix C. We conclude the review in Section 6, discussing the results and future paths for the area of VXAI.

### Terminology

To avoid ambiguous language, throughout the paper we stick to the terminology of the XAI Handbook by Palacio et al. (2021): The goal of XAI is to facilitate understanding by providing insights into an *explanandum* ("What is to be explained"), usually a model or a model's decision. To accomplish this, we leverage an *explanation*, which is the process of getting insight into the explanandum. The resulting output of this process is the *explanans*, which provides the user with information about the model's inner workings. In a mathematical sense, the explanation can be viewed as a function that maps an explanandum to an explanans. For example, the explanandum could be a CNN's classification of a given input image. The explanation might be an algorithm such as GradCAM (Selvaraju et al., 2017), and the resulting heatmap is the explanans, which highlights important features. We use the Latin plural forms explananda (explanandum) and explanantia (explanans) throughout. When we refer to VXAI, we include both the evaluation of the method (explanation) and its output (explanans), since most evaluation metrics necessarily assess the quality of explanations through the quality of their generated outputs.

As defined by the XAI Handbook, *interpretation* (or *interpretability*) refers to the subsequent assignment of meaning to an explanation. It describes the process through which a human infers knowledge about the explanandum using the explanans. This step significantly influences the success of the explanation and also depends on the receiving human's (the *explainee*'s) mental model.

---

**Terminology**

**explanandum** (pl. **explananda**): What is to be explained, i.e. a model and its prediction.
**explanation** (pl. **explanations**): The process of explaining, i.e. the XAI algorithm.
**explanans** (pl. **explanantia**): The explaining information, i.e. the output of an explanation.

---

## 2 Related Work

Although the field of XAI has gained popularity over the past years, there is still no extensive and unified evaluation framework for XAI metrics. Various surveys have explored XAI and VXAI from different angles, ranging from human-grounded evaluation to technical metrics. Table 1 gives an overview over 30 such XAI reviews from the past years.

While evaluation of XAI is frequently given less attention in XAI surveys, 23 of these reviews directly focus on the topic of VXAI. Besides functionality-grounded evaluation, a second school of thought is concerned with human-grounded evaluation of explanations through qualitative expert evaluations or quantitative user studies, with representative surveys for this domain available as well (Sokol & Flach, 2020; Rawal et al., 2021; Naveed et al., 2024). Nevertheless, a considerable number of 19 reports focus specifically on the topic of functionality-grounded evaluation. Unfortunately, most of these surveys focus on a subset of well-known metrics, whereas only 14 surveys gathered VXAI metrics in a systematic or semi-systematic literature review. Further, numerous of the referenced reviews either lack an extensive list of desiderata and focus only on a subset of them, or limit their research to specific types of explanations[2] or application domains.

There are five reviews, that we consider most similar to this work, as they present systematic functionality-grounded VXAI surveys. Le et al. (2023) and Nauta et al. (2023) both categorize the identified metrics based on a scheme of 12 properties, namely the Co-12 framework, which we discuss in more detail in Subsection B.1. However, Le et al. (2023) restrict their analysis to metrics available through public XAI or VXAI toolkits (e.g., Quantus (Hedström et al., 2023)) and do not report on metrics introduced in the literature but not implemented in such libraries. Although there is some overlap with the study by Nauta et al. (2023), particularly in the inclusion of some identical metrics, their review also incorporates studies that merely apply VXAI metrics rather than introducing them. In contrast, our work provides detailed descriptions and categorizations of each identified metric. The review by Kadir et al. (2023) covers a broad range of domains and explanation types[2] and reports a wide variety of metrics. It also groups several metrics by method, a strategy shared by our work. However, it does not adopt a categorization scheme based on the desiderata fulfilled by individual metrics. Notably, the recent reviews from Bayrak & Bach (2024) and Pawlicki et al. (2024) report a high number of individual metrics for VXAI. However, both limit the scope of their review considerably, either in terms of application domain (Pawlicki et al., 2024) or explanation type[2] (Bayrak & Bach, 2024). In contrast, our work includes all metrics reported to date and introduces a categorization scheme based on three individual dimensions. Finally, many of the metrics we identified were introduced only recently, underlining the need for this more recent literature review.

While previous reviews report between 10 and 90 individual metrics, our work introduces a unified structure by aggregating over 360 individual metrics into 41 conceptually related groups. This enables clearer comparison and interpretation across metrics. Unlike most surveys, we do not limit our analysis to specific explanation types or application domains, ensuring broader applicability across the XAI landscape.

---

[2]The *explanation type* refers to both the design of the explanation algorithm and, consequently, the nature of the resulting explanans, as introduced in Subsection 4.1.3.

| Work | VXAI Focus | Functionality-Grounded VXAI | (Semi-)Systematic | Date ↓ | Desiderata | Limited to | Reported Metrics |
|---|---|---|---|---|---|---|---|
| This work | ✓ | ✓ | ✓ | Jan 2025 | Parsimony, Plausibility, Coverage, Fidelity, Continuity, Consistency, Efficiency | | 41 metrics from 362 sources |
| Klein et al. | ✓ | ✓ | | Jan 2025 | Faithfulness, Robustness, Complexity | Feature Attributions; Computer Vision | 20 metrics |
| Pawlicki et al. | ✓ | ✓ | ✓ | Oct 2024 | | Cybersecurity | 86 metrics |
| Awal & Roy | ✓ | ✓ | | Jun 2024 | Reliability, Consistency | Rule Explanations | 6 metrics |
| Bayrak & Bach | ✓ | ✓ | ✓ | Apr 2024 | | Counterfactuals | 66 metrics |
| Bommer et al. | ✓ | ✓ | | Mar 2024 | Robustness, Faithfulness, Complexity, Localization, Randomization | Climate Science | 10 metrics |
| Li et al. | ✓ | ✓ | | Dec 2023 | Faithfulness | Feature Attributions | 6 metrics |
| Alangari et al. | ✓ | | | Aug 2023 | Correctness, Comprehensibility, Stability | | 59 metrics |
| Le et al. | ✓ | ✓ | ✓ | Aug 2023 | Co-12† | | 86 metrics from 17 toolkits |
| Salih et al. | ✓ | | ✓ | Aug 2023 | | Cardiology | 27 metrics |
| Kadir et al. | ✓ | ✓ | ✓ | Jul 2023 | | | 80 metrics |
| Hedström et al. | ✓ | ✓ | | Apr 2023 | Faithfulness, Robustness, Localization, Complexity, Axiomatic, Randomization | Feature Attribution | 27 metrics |
| Schwalbe & Finzel | | | ✓ | Jan 2023 | | | 11 metrics (already grouped) |
| Agarwal et al. | ✓ | ✓ | | Nov 2022 | Faithfulness, Stability | Feature Attributions | 11 metrics |
| Coroama & Groza | ✓ | | | Nov 2022 | | | 26 metrics |
| Verma et al. | | ✓ | | Nov 2022 | | Counterfactuals | 9 metrics (already grouped) |
| Belaid et al. | ✓ | ✓ | | Oct 2022 | Fidelity, Fragility, Stability, Simplicity, Stress, Other | Feature Attributions | 22 metrics |
| Cugny et al. | ✓ | ✓ | | Oct 2022 | | | 6 metrics |
| Lopes et al. | ✓ | | ✓ | Aug 2022 | Fidelity (Completeness, Soundness), Interpretability, Broadness, Simplicity, Clarity) | | 43 metrics |
| Yuan et al. | | ✓ | ✓ | Jul 2022 | Fidelity, Sparsity, Stability, Accuracy | Graph Neural Networks | 7 metrics |
| Löfström et al. | ✓ | | ✓ | Mar 2022 | | | 10 metrics |
| Vilone & Longo | ✓ | | ✓ | Dec 2021 | | | 36 metrics |
| Bodria et al. | | ✓ | ✓ | Nov 2021 | | | 6 metrics |
| Sovrano et al. | ✓ | | | Oct 2021 | Similarity, Exactness, Fruitfulness | | 22 metrics |
| Ras et al. | | | | Sep 2021 | | | *13 sources in text* (metrics not listed) |
| Mohseni et al. | | | ✓ | Aug 2021 | Fidelity, Trustworthiness | | 15 metrics |
| Yeh & Ravikumar | ✓ | ✓ | | Jun 2021 | | | 7 metrics |
| Nauta et al. | ✓ | ✓ | ✓ | May 2021 | Co-12† | | 28 metrics (already grouped) |
| Zhou et al. | ✓ | ✓ | | Jan 2021 | Fidelity (Completeness, Soundness), Interpretability, Broadness, Simplicity, Clarity) | | 17 metrics |
| Samek & Müller | | | | Sep 2019 | | | *16 sources in text* (metrics not listed) |
| Yang et al. | ✓ | ✓ | ✓ | Aug 2019 | Generalizability, Fidelity, Persuasibility | | *40 sources in text* (metrics not listed) |

†Co-12: Correctness, Output-Completeness, Consistency, Continuity, Contrastivity, Covariate Complexity, Compactness, Composition, Confidence, Context, Coherence, Controllability

Table 1: Overview of recent XAI reviews, sorted by date. The table indicates whether each survey primarily focused on evaluation metrics, whether it reported mainly functionality-grounded metrics, and whether a (semi-)structured review was conducted. The date refers to the earliest available point in the article's timeline; either the database query, submission, or publication, depending on what was reported. For each survey, we also report the desiderata used to classify the metrics and any limitations regarding explanation type or application domain. Note that not all surveys systematically listed their assessed metrics, so the reported metric count may vary depending on the method of extraction.

## 3    Method

In our review, we aim to systematically collect and classify all functionality-grounded metrics relevant to evaluating explanations in the context of XAI. We base our review on the PRISMA (Page et al., 2021) guidelines to make the process transparent. Figure 3 gives an overview of our process.

The overall procedure consisted of two stages: an initial structured database search to identify secondary literature (e.g., surveys and reviews), followed by a recursive backward snowballing stage targeting primary sources that introduced new metrics. In total, we reviewed 1,459 papers, screened 866 in full, and included 362 that proposed an original VXAI metric or one of its variants. Comprehensive details of the database queries, screening criteria, and inclusion statistics are reported in Appendix A.

## 4    The VXAI Framework

In this section, we present the twofold contribution that constitutes the VXAI framework: (i) a three-dimensional categorization scheme for evaluating explainability metrics, and (ii) a comprehensive overview of the metrics identified through our systematic literature review. The categorization scheme provides the conceptual structure used to organize and analyze the collected metrics.

From the literature review, we identified a total of 362 individual references, some of which introduce or modify multiple metrics simultaneously. We grouped these metrics based on methodological similarity, aggregating those that are functionally similar and measure the same underlying property into distinct *aggregated metrics*. For each aggregated metric, we determined the associated desiderata, the metric's contextuality, and the suitable explanation types as defined in our categorization scheme below.

We first formalize the dimensions of this categorization scheme before presenting the overview of aggregated metrics in Subsection 4.1. To maintain readability, this section this section mainly focusses on a broader overview of the desiderata, and meta-results and statistics of the metrics. Table 2 summarizes the VXAI framework by listing all aggregated metrics structured along the categorization scheme, while detailed descriptions and corresponding references are provided in Appendix C. An interactive version of the framework is available at `https://vxai.dfki.de/`.

### 4.1    Categorization Scheme

#### 4.1.1    Overview

To facilitate the selection of evaluation metrics for testing explanations, we propose a categorization scheme that groups the identified metrics into functionally similar approaches. Serving both as a conceptual overview and a practical guide, it is structured along three orthogonal dimensions:

(i) the **Desiderata** each metric contributes to,
(ii) the suitable **Explanation Types** to which a metric applies, and
(iii) the **Contextuality**, given by the degree of dependency on the model and data.

#### 4.1.2    Desiderata

We annotate each metric with the functional desiderata it addresses. To justify our selection, we compared several of the most common desiderata formulations proposed in related works (see Subsection B.1). We find substantial overlap among these frameworks in their core desiderata, yet also notable gaps: some desiderata are inconsistently covered, while others included in prior work do not directly translate to functionality-grounded evaluation. Building on this comparison, we derive a coherent set of desiderata that capture both interpretability and technical soundness.

Specifically, we follow a two-stage view of explaining: an *Interpretability dimension (I)*, concerned with how the explanation is perceived and understood by humans, and a *Technical dimension (T)*, focused on the reliability and rigor of the explanans. The seven desiderata (fully defined in Subsection B.2) are:

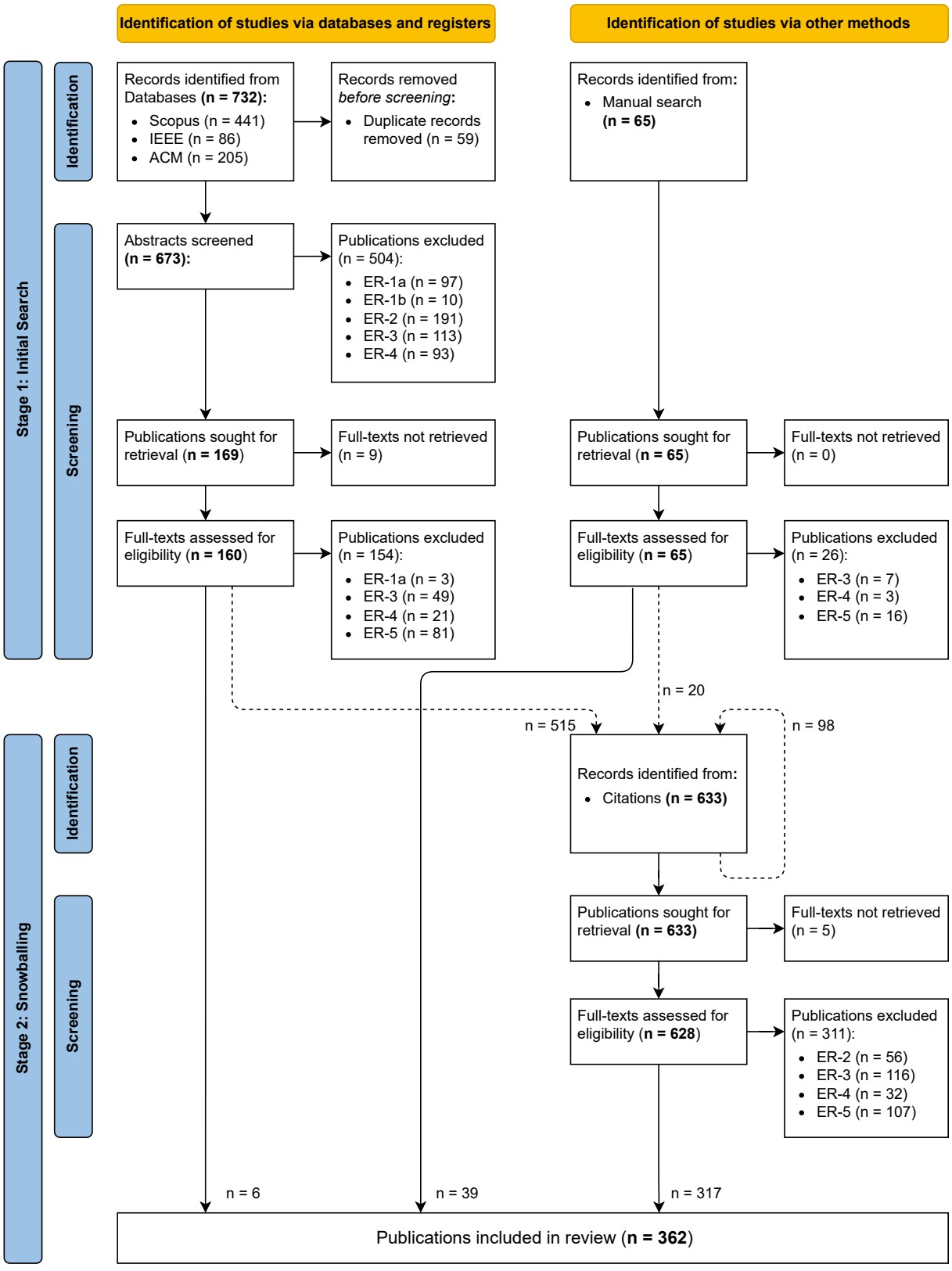

Figure 3: Our search strategy building upon the PRISMA guidelines (Page et al., 2021). Notably, we split the process into an initial database search phase and a snowballing phase, identifying the most relevant literature in the second phase.

- **Parsimony (I):** The explanation should keep the explanans concise to support interpretability.
- **Plausibility (I):** The explanation should shape the explanans to align with human expectations.
- **Coverage (T):** The explanation should provide an explanans for every explanandum.
- **Fidelity (T):** The explanation should make the explanans reflect the model's true reasoning.
- **Continuity (T):** The explanation should ensure that similar explananda yield similar explanantia.
- **Consistency (T):** The explanation should produce stable explanantia across repeated evaluations.
- **Efficiency (T):** The explanation should compute the explanans efficiently and broadly.

Because various metrics contribute to multiple desiderata, we do not enforce a one-to-one mapping between metrics and desiderata.

### 4.1.3 Explanation Types

We categorize VXAI metrics based on the accepted input. Apart from a few exceptions, most metrics are agnostic to the underlying black-box model or data format (e.g., tabular, image, or graph). Therefore, we do not consider this dimension separately. Instead, we follow prior XAI classification schemes that organize methods based on the explanation approach or resulting explanans (Carvalho et al., 2019; Markus et al., 2021; Zhang et al., 2021; Speith, 2022; Nauta et al., 2023).

For XAI algorithms, a distinction is often made between local and global methods (Zhang et al., 2021; Speith, 2022; Bedi et al., 2024). We do not translate this distinction to VXAI metrics (unlike Robnik-Šikonja & Bohanec (2018)), as most metrics can be adapted accordingly, for instance, by computing changes in logits for a single instance rather than accuracy over an entire dataset. Conversely, metrics that provide local scores can be aggregated across explanantia to obtain global results. Another frequent distinction is made between explanations that can be built directly into a model (in-hoc) or derived after training (post-hoc) (Carvalho et al., 2019; Zhang et al., 2021; Speith, 2022; Nauta et al., 2023; Bedi et al., 2024). The metrics in the VXAI framework are largely agnostic to this distinction and can generally be applied to either approach.

We differentiate between five principal explanation types: **Feature Attributions**, **Concept Explanations**, **Example Explanations**, **White-Box Surrogates**, and **Natural Language Explanations**. We introduce each of these types below. Similar to the formulation of desiderata, our categorization is extensible and not mutually exclusive. Because many XAI methods combine multiple forms of explanations, metrics designed for one type often transfer to others. Examples include LIME, which produces a local surrogate yielding attribution scores (Ribeiro et al., 2016), or the generation of counterfactuals by leveraging prior surrogates (Pornprasit et al., 2021) or attributions (Ge et al., 2021; Albini et al., 2022). This overlap enhances the framework's flexibility and supports systematic comparison across different explanation families.

**Feature Attributions (FAs)** return a vector $e \in \mathbb{R}^d$, typically (but not necessarily) matching the dimensionality of the input $x$. Each element $x_j$ represents an input feature, such as a column in tabular data, a (super-)pixel in an image, or a node in a graph, with value $e_j$ indicating its relevance to the prediction. Attribution values may be positive or negative and can be continuous, discrete, or thresholded, depending on the underlying method. Saliency maps form a structured subtype commonly used in computer vision, where features exhibit spatial relations (Bach et al., 2015; Ribeiro et al., 2016; Lundberg, 2017; Shrikumar et al., 2017; Sundararajan et al., 2017). Although most FAs are local, aggregating individual explanations can yield global feature-importance estimates (Lundberg, 2017; Molnar, 2020).

**Concept Explanations (CEs)** capture higher-level, human-interpretable properties beyond individual features, such as visual patterns or abstract semantic ideas. They are typically extracted from intermediate model representations, resulting in a reduced and interpretable set of dimensions compared to the input space. Unlike feature-level relevance, concepts are meaningful on their own and may appear across multiple inputs. Consequently, they establish a middle ground between local and global explanations, as their detection in a single instance is local, while their overall contribution to model behavior is global (Kim et al., 2018).

**Example Explanations (ExEs):** reside directly in the input space and explain models through representative instances. They include counterfactuals identifying minimal input changes that alter the prediction outcome (Wachter et al., 2017; Karimi et al., 2020; Mothilal et al., 2020; Verma et al., 2024), as well as

prototypes or factuals representing typical or minimally changed samples within a class (Kim et al., 2016; Dhurandhar et al., 2019; Koh & Liang, 2017; Molnar, 2020). Local ExEs describe instance-specific changes, while global ones summarize model behavior through sets of influential or representative samples.

**White-Box Surrogates (WBSs)** approximate the behavior of a black-box model using interpretable surrogate models that themselves serve as the explanans. These surrogates can capture global behavior by reconstructing the overall decision logic of the black box, as commonly done with decision trees or rule sets (Craven & Shavlik, 1995; Friedman & Popescu, 2008), or describe local neighborhoods that provide explanantia for specific subsets of inputs (Ribeiro et al., 2016; 2018).

**Natural Language Explanations (NLEs)** provide textual justifications that accompany or follow model predictions. They include models that generate explanations jointly during inference (Ras et al., 2022; Camburu et al., 2018; Wei et al., 2022) as well as post-hoc generation through large language models (Bills et al., 2023). Template-based approaches that merely verbalize existing explanantia (e.g., FAs) are not regarded as genuine NLEs, as they reformulate rather than generate new content (Lucieri et al., 2022; Das et al., 2023). We therefore propose evaluating the underlying explanans and resulting explanation instead.

### 4.1.4 Contextuality

Finally, we distinguish metrics by their evaluation context, which defines how strongly they depend on or intervene in the underlying model or data. We identify five levels, each introducing progressively deeper contextual interaction:

I) **Explanans-Centric:** Evaluates only the explanans in relation to the raw input instance, fully independent of the model.
II) **Model Observation:** Relies on access to model outputs or internal activations to assess behavior.
III) **Input Intervention:** Perturbs input data and observes resulting changes in predictions or explanantia.
IV) **Model Intervention:** Alters the model itself, e.g., by retraining or parameter randomization.
V) **A Priori Constrained:** Requires specific data, architectures, or experimental setups.

We conceptualize Contextuality as a technical property describing how a metric interacts with or intervenes in the evaluated model or explanation. This dimension was derived inductively from the surveyed metrics, where recurring technical dependencies naturally formed five distinct contextual levels.

These levels reflect a gradual shift from **In-Situ** to **Ex-Situ** evaluation,[3] moving from metrics that operate directly on given explanantia and predictions (Levels I–III) to those that evaluate explanation methods under modified or constrained conditions (Levels IV–V). This gradual structure loosely parallels the distinction between ante-hoc, in-hoc, and post-hoc explainability (Carvalho et al., 2019; Zhang et al., 2021; Speith, 2022; Nauta et al., 2023; Bedi et al., 2024), as both progress from intrinsic to externally applied processes; though our formulation pertains specifically to evaluation rather than explanation generation. While Ex-Situ evaluations support method-level benchmarking, only In-Situ metrics inform the quality of explanations in their specific deployment context. By distinguishing these levels, Contextuality assists practitioners in selecting metrics suited to their evaluation setup and interpreting results consistently. Since all surveyed metrics align with these five levels, we currently consider the Contextuality dimension comprehensive.

## 4.2 Identified Metrics

Now that we have established our classification scheme, we turn to the results of our literature review and report key statistics about the identified metrics.

We identified metrics across 362 sources. Since some works proposed multiple variants, the total number of found metrics exceeded 400. Many of these metrics followed closely related approaches or represented minor variants of one another, e.g., differing only in hyperparameters, evaluation setup, or dataset-specific

---

[3]From Latin for "in place" and "off site", respectively.

| Contextuality | Aggregated Metric | Desiderata | | | | | | | Explanation Type | | | | | #References |
|---|---|---|---|---|---|---|---|---|---|---|---|---|---|---|
| | | Parsimony | Plausibility | Coverage | Fidelity | Continuity | Consistency | Efficiency | FA | ExE | CE | WBS | NLE | |
| **I** | (1) Explanans Size | ✓ | | | | | | | ✓ | ✓ | (✓) | ✓ | (✓) | 51 |
| | (2) Overlap | ✓ | | | | | | | | | | ✓ | | 5 |
| | (3) Explanans Cohesion | ✓ | ✓ | | | | | | ✓ | | (✓) | | | 2 |
| | (4) Minimality | ✓ | ✓ | | | | | | | ✓ | | | | 25 |
| | (5) Autoencoder Plausibility | | ✓ | | | | | | | ✓ | | | | 1 |
| | (6) Diversity | | ✓ | | | | | | | ✓ | | | | 6 |
| | (7) Input Similarity | | ✓ | | | | | | | ✓ | | | | 11 |
| | (8) Input Contrastivity | | ✓ | | (✓) | | | | ✓ | (✓) | (✓) | (✓) | (✓) | 2 |
| | (9) Actionability | | ✓ | | ✓ | | | | | ✓ | | | | 7 |
| | (10) Model-Agnostic Explanation Consistency | | ✓ | | | | (✓) | | ✓ | ✓ | (✓) | (✓) | (✓) | 4 |
| | (11) Input Coverage | | | ✓ | | | | | ✓ | ✓ | (✓) | ✓ | (✓) | 8 |
| | (12) Output Coverage | | | ✓ | | | | | | | | ✓ | | 1 |
| | (13) Output Mutual Information | | | | ✓ | | | | ✓ | | ✓ | | | 1 |
| **II** | (14) Input Mutual Information | ✓ | | | | | | | ✓ | | ✓ | | | 1 |
| | (15) Output Contrastivity | | ✓ | | | | | | ✓ | (✓) | (✓) | (✓) | (✓) | 5 |
| | (16) Output Similarity | | ✓ | | | | | | | ✓ | | | | 1 |
| | (17) Mutual Coherence | | ✓ | | (✓) | | | | ✓ | (✓) | (✓) | (✓) | (✓) | 19 |
| | (18) Significance Check | | | | ✓ | | | | ✓ | (✓) | ✓ | (✓) | (✓) | 15 |
| | (19) (Counter-)Factual Relevance | | | | ✓ | | | | | ✓ | | | | 1 |
| | (20) Prediction Validity | | | | ✓ | | | | | ✓ | | | | 18 |
| | (21) Sufficency | | | | ✓ | | | | | | ✓ | | | 2 |
| | (22) Output Faithfulness | | | | ✓ | | | | | | | ✓ | | 34 |
| | (23) Internal Faithfulness | | | | ✓ | | | | | | | ✓ | | 3 |
| | (24) Setup Consistency | | | | | | ✓ | | ✓ | ✓ | (✓) | ✓ | (✓) | 11 |
| | (25) Hyperparameter Sensitivity | | | | | | ✓ | | ✓ | (✓) | (✓) | (✓) | (✓) | 6 |
| | (26) Execution Time | | | | | | | ✓ | ✓ | ✓ | (✓) | ✓ | (✓) | 34 |
| **III** | (27) Unguided Perturbation Fidelity | | | | ✓ | | | | ✓ | | | | | 10 |
| | (28) Guided Perturbation Fidelity | | | | ✓ | | | | ✓ | | ✓ | | | 75 |
| | (29) Counterfactuability | | | | ✓ | | | | | | | ✓ | | 1 |
| | (30) Prediction Neighborhood Continuity | | | | ✓ | ✓ | | | | | | ✓ | | 1 |
| | (31) Quantification of Unexplainable Features | | | (✓) | ✓ | | | | ✓ | | (✓) | | | 2 |
| | (32) Neighborhood Continuity | | | | ✓ | | | | ✓ | ✓ | (✓) | ✓ | ✓ | 19 |
| | (33) Adversarial Input Resilience | | | | ✓ | | | | ✓ | (✓) | (✓) | (✓) | (✓) | 10 |
| **IV** | (34) Model Parameter Randomization Test | | | | ✓ | | | | ✓ | (✓) | (✓) | (✓) | (✓) | 5 |
| | (35) Data Randomization Test | | | | ✓ | | | | ✓ | (✓) | (✓) | (✓) | (✓) | 2 |
| | (36) Retrained Model Evaluation | | | | ✓ | | | | ✓ | | | | | 10 |
| | (37) Influence Fidelity | | | | ✓ | | | | | ✓ | | | | 2 |
| | (38) Normalized Movement Rate | | | | | ✓ | | | ✓ | | | | | 2 |
| | (39) Adversarial Model Resilience | | | | ✓ | | | | ✓ | (✓) | (✓) | (✓) | (✓) | 4 |
| **V** | (40) GT Dataset Evaluation | | ✓ | | ✓ | | | | ✓ | | ✓ | | ✓ | 119 |
| | (41) White Box Model Check | | | | ✓ | | | | ✓ | (✓) | | | (✓) | 12 |

Table 2: The 41 aggregated metrics identified in our study, as presented in Appendix C. Each metric is classified according to its associated desiderata, applicable explanation types, and level of contextuality. ✓ indicates full alignment or reported usage in the literature, while (✓) denotes partial contribution or unreported but plausible applicability. The final column shows the number of references per metric.

details. To provide a more coherent and interpretable overview, we therefore summarized related metrics into broader, high-level *aggregated metrics* that capture a shared conceptual core. In total, we derived 41 functionally distinct aggregated metrics based on shared goals, methods, or assumptions. Each was then categorized using the three-dimensional scheme introduced above. Since a single metric may serve multiple desiderata and apply to different explanation types, we organize them primarily by their contextuality level. Table 2 presents a complete overview of all metrics and their classification. Key patterns are summarized below; for detailed descriptions of each individual metric, see Appendix C. Four metrics deemed conceptually flawed or misaligned with any desideratum were excluded and are discussed in Appendix D.

**Metric Popularity and References:** On average, each metric is supported by 13.4 references (standard deviation 22.5), though the median is only 5. While a few metrics are backed by large reference sets (e.g., 119, 75, and 51 citations), eight metrics are supported by just one. This should not be interpreted as lack of relevance: our focus was on original proposals, not reuse or popularity. The most cited metric, Ground-Truth Dataset Evaluation, reflects the ubiquity of using annotated or synthetic datasets for evaluating explanation quality; an intuitive strategy with virtually unlimited implementation variants.

**Desiderata:** Most metrics target *Fidelity*, with 18 doing so directly and 3 more partially. *Plausibility* follows with 12 metrics. By contrast, *Efficiency*, *Coverage*, and *Consistency* are least represented, with only 1, 2, and 2 metrics respectively. The majority of metrics (32) address a single desideratum, while 4 target two equally and 5 contribute partially to a secondary one.

**Explanation Types:** A total of 25 metrics are applicable to FA methods. An equal number is available for ExE, although only 15 are directly supported by the literature; the remaining 10 are marked as potentially applicable (see (✓) in Table 2 and the opaque bars in Figure 4). For other explanation types, direct literature support is more limited: 11 metrics for WBS, 6 for CE, and just 2 for NLE. However, we consider many metrics adaptable even in the absence of published usage, increasing the totals to 22 for CE, 21 for WBS, and 17 for NLE. In total, 15 metrics are applicable (or adaptable) across all explanation types, and 19 are proprietary to a single type. Notably, no metric is exclusive to NLE, and only one is unique to CE. The largest overlap exists between FA and CE, with 21 metrics covering both. FA also has the highest number of adaptable metrics: while only 3 are exclusive to FA alone, 22 are shared with other types.

**Contextuality:** Most metrics fall into the less restrictive categories. Specifically, 13 are classified as *Contextuality I* (Explanans-Centric) and another 13 as *Contextuality II* (Model Observation), both of which do not require altering model or input. As contextual demands increase, metric availability declines: 7 fall into *Contextuality III* (Input Intervention), and 6 into *Contextuality IV* (Model Intervention). The most restrictive category, *Contextuality V* (A Priori Constrained), includes only two metrics that require specific setups with known ground-truth rationales, either via synthetic data or interpretable white-box models.

**Desideratum-Contextuality Interactions:** Metrics targeting the interpretability desiderata *Parsimony* and *Plausibility* are found almost exclusively in Contextualities I and II. Similarly, the technical desiderata *Coverage*, *Consistency*, and *Efficiency* are addressed only through observational strategies belonging to the same Contextualities of I and II. In contrast, all metrics measuring *Continuity* require intervention, either through modified inputs (Contextuality III) or altered model internals (Contextuality IV). *Fidelity*, in turn, is represented across all contextuality levels.

**Contextuality-Explanation Interactions:** There are no striking anomalies in how metrics available for specific explanation types are distributed across contextuality levels. The only exception is that no metric in Contextuality V targets WBS, likely because it is difficult to define a single ground-truth surrogate when multiple equivalent surrogates may exist.

**Desideratum-Explanation Interactions:** Every explanation type is covered by at least one metric for every desideratum, ensuring that all evaluation dimensions are, in principle, addressable regardless of the explanation form. The overall distribution across types is balanced. The only notable concentration is *Plausibility* in ExE, with 10 metrics. This may reflect the inherently interpretable nature of many ExE techniques (e.g., counterfactuals, prototypes), which naturally suit human-aligned plausibility assessments.

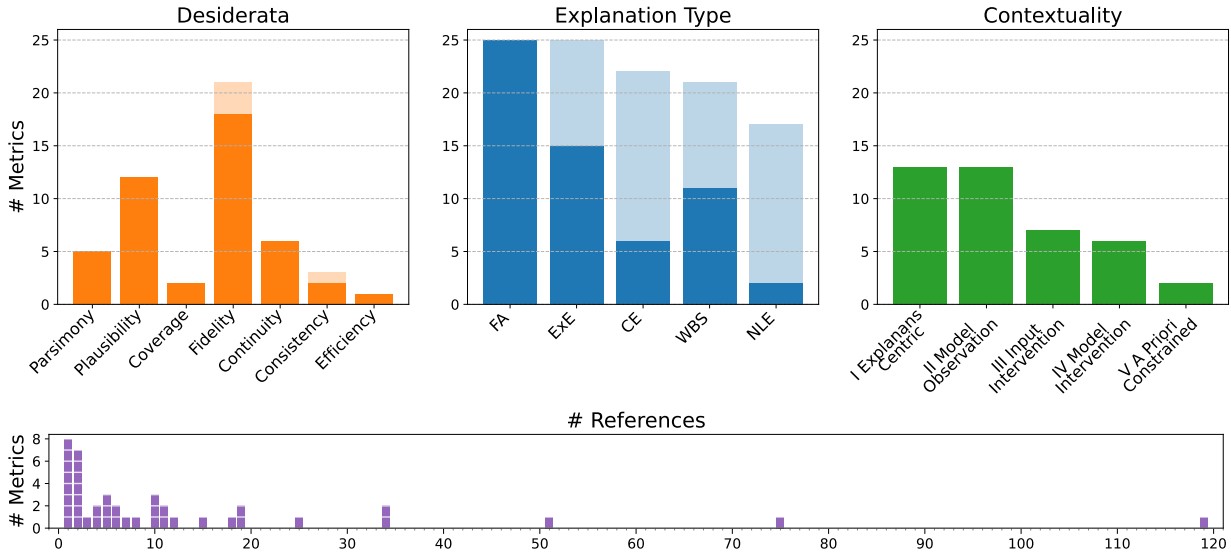

Figure 4: Overview of the distribution of metrics within the categorization scheme. Each metric may be associated with multiple desiderata and explanation types, but only a single level of contextuality. Light-colored bars indicate partial alignment with a desideratum. For explanation types, light bars denote cases where no usage has been reported in the literature, though the metric is considered adaptable. The bottom histogram shows the number of metrics grouped by their reference count.

These patterns highlight the breadth of our framework. Overall, the proposed categorization encompasses a wide range of metrics across explanation types, desiderata, and evaluation contexts, offering a comprehensive foundation for structured VXAI assessment.

## 5 Discussion

Finally, we reflect on trends observed in the identified VXAI metrics and offer considerations for their interpretation and future application.

**General observations:** Our framework reveals a wide and diverse landscape of VXAI metrics. While each aims to quantify a specific property of explanations, few metrics offer clear thresholds or benchmarks to determine what constitutes "good" explanation quality. This lack of consensus limits interpretability and comparability across studies. Furthermore, although several metrics address similar desiderata, explanation types, or contextuality levels, each typically serves a distinct niche. These differences often stem from variations in what aspect of a desideratum is targeted, or from adapting to specific explanation structures.

**Emphasis on Fidelity:** Fidelity stands out as the most frequently addressed desideratum, both in terms of metric count and methodological variety. This supports our perspective of Fidelity as a fundamental aspect of explanation quality (see Subsection B.3). However, the range of proposed metrics also demonstrates that there is no universally accepted approach to quantifying it. The same holds for the desiderata Parsimony, Plausibility, and Continuity, each offering a range of possible evaluation metrics. In contrast, Coverage, Consistency, and Efficiency are addressed by fewer metrics. This is likely due to their more straightforward quantifiability; for instance, Coverage is trivially satisfied if explanantia exists, and Efficiency can be assessed via computation time. Portability, a part of Efficiency, is similarly important, but we identified no functionality-grounded metrics that are associated with it. Hence, it remains a descriptive property of the method rather than something measurable on individual explanantia. More broadly, the number of metrics associated with a desideratum does not necessarily reflect how well it can be quantified. Some desiderata may be better captured by a single strong metric than others by a diverse set of weaker proxies.

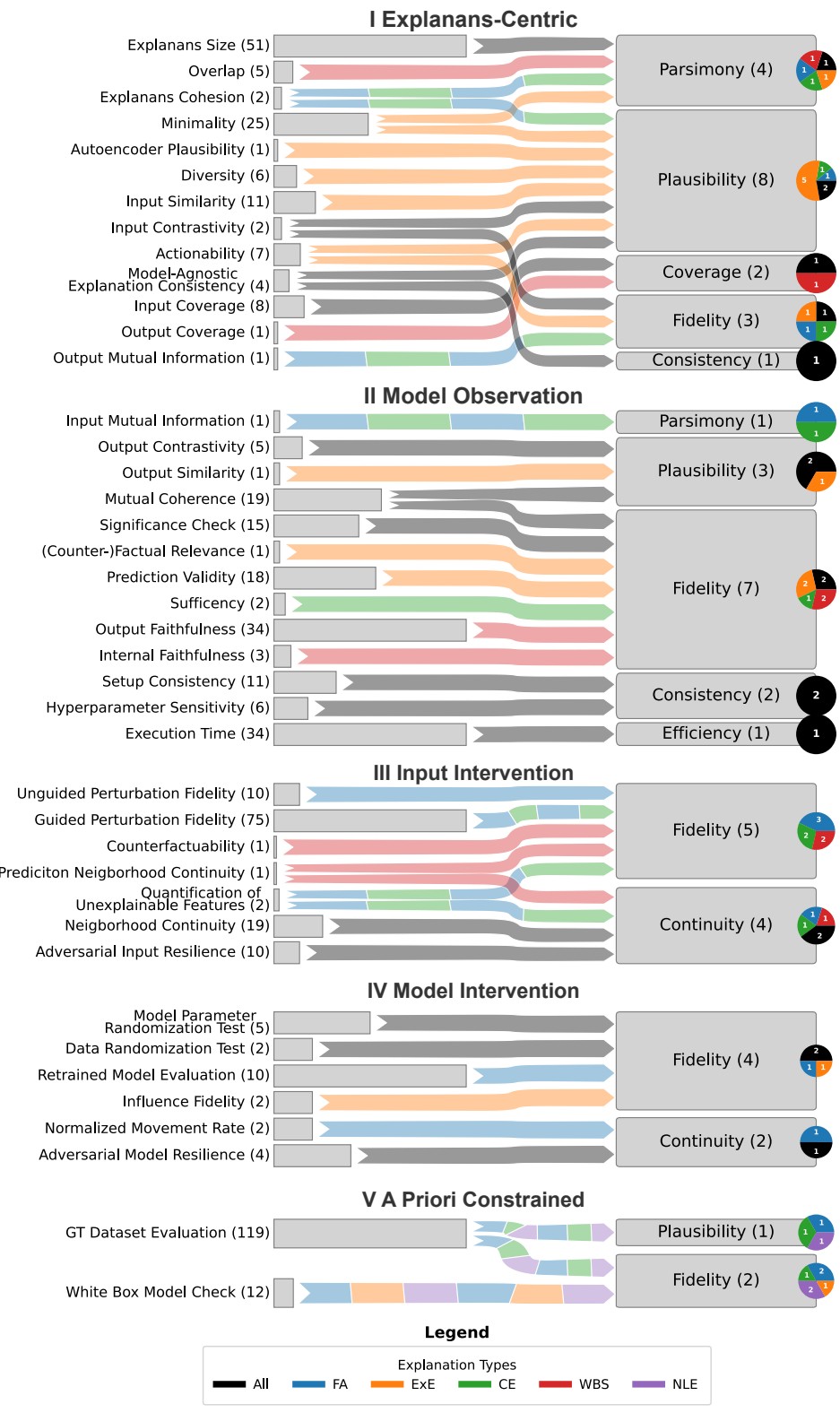

Figure 5: Overview of the 41 identified metrics, grouped by contextuality. Each metric is represented by a horizontal bar indicating the number of supporting references. Metrics are mapped to their associated desiderata, with arrow colors denoting the corresponding explanations type. These associations do not distinguish between full (✓) and partial (✓) alignment (unlike Table 2). For each desideratum, a pie chart summarizes the distribution of linked metrics by explanation type.

**Explanation type focus:** Among explanation types, FA is the most extensively studied, which is reflected both in literature prevalence and metric availability. This focus has also enabled the adoption of classical techniques from adjacent fields. For example, treating FA explanations as feature selectors allows the use of established stability metrics from feature selection research, such as those used by Nogueira et al. (2018). Conversely, the explanation type NLE is still underrepresented in dedicated metrics. Nonetheless, our framework is flexible enough to incorporate emerging metrics in this and further areas.

**Transferability across explanation types:** A notable pattern is that many metrics are rarely applied beyond one or two explanation types. In our analysis, several were marked as "potentially applicable" to additional types, but lacked evidence of actual transfer. This suggests an isolated view of explanation types in current research. We view this as a missed opportunity, as generalizing and adapting metrics across different explanation types could foster a more integrated evaluation practice.

**Metrics embedded in the explanation process:** Some explanation types, notably ExE and WBS, commonly embed evaluation metrics as part of their generation objectives. For example, counterfactuals often optimize for plausibility and proximity directly during their construction (Wachter et al., 2017; Dandl et al., 2020; Kanamori et al., 2020). While this tight integration may enhance the generated explanans, it raises concerns about circularity in evaluation, especially in light of Goodhart's Law (Strathern, 1997): If a metric is optimized for, it may no longer serve as a reliable post-hoc assessment.

**Subjectivity and the limits of automation:** Metrics for Parsimony and Plausibility, while quantifiable, are ultimately shaped by subjective human interpretation. Functionality-grounded scores offer useful proxies that can be evaluated automatically and without costly human studies. However, they cannot replace human-centered assessments of whether an explanation is truly accessible or helpful in practice.

**Caveats in specific metric designs:** Several metric designs warrant closer scrutiny. Ground-truth-based metrics, specifically those relying on human-annotated rationales (see Metric 40), are widely used but have notable limitations. These include their tendency to reflect plausibility rather than explanation fidelity (Camburu et al., 2019; Carmichael & Scheirer, 2023), vulnerability to trivial baselines like central focus (Gu et al., 2019), and their inability to capture model reliance on context or background (Arras et al., 2022; Brandt et al., 2023). There's no reason to assume that the human rationale and the models' rationale are aligned (Khakzar et al., 2022), resulting in low scores for truthful explanations generated from an implausible model (Samek & Müller, 2019). Likewise, metrics based on input perturbation or retraining (e.g., Metric 28, Metric 32, or Metric 36) can yield misleading results. Feature removal may not affect model output if redundant features are present (D'Amour et al., 2022), and retraining can alter the model in uncontrolled ways, obscuring meaningful evaluation (Hooker et al., 2019).

**Metric aggregation:** While most studies report multiple metric scores separately, some recent work proposes aggregating them into a single composite score (Farruque et al., 2021; Margot & Luta, 2021; Poppi et al., 2021; Bommer et al., 2024). Suggested methods include scaling by theoretical bounds, comparison to random or perfect baselines, and aggregation via weighted sums or harmonic means. Despite these proposals, the field lacks consensus on how to combine metrics meaningfully and robustly.

**The role of white-box models in VXAI:** These limitations prompt reflection on whether post-hoc explanation of black-box models is always justified. Some authors recommend comparing black-box performance to that of directly trained white-box models (Zhang et al., 2019b; Rosenfeld, 2021; Margot & Luta, 2021). If performance is comparable, choosing an interpretable model may be preferable, especially in high-stakes settings, as advocated by Rudin (2019). The same logic applies when considering surrogate models. They should be benchmarked against white-box models trained directly on the data (Barakat et al., 2010).

Although not the main focus of this work, many of the identified metrics can also be extended to directly interpretable white-box models. Here, matching metrics can usually be found in the White-Box Surrogate explanation type. The primary exception are metrics under the Fidelity desideratum, as Fidelity measures the alignment between the explanans and the explanandum, which presupposes a black-box reference. In

such cases, the Fidelity desideratum could instead be replaced by a Performance desideratum, where standard predictive metrics such as accuracy or mean squared error serve as proxies for model quality. Subsection B.2.8 points to other frameworks in related work, that have such desiderata.

**Beyond discriminative tasks:** While the majority of existing XAI algorithms were developed for discriminative settings, the metrics compiled within the VXAI framework are formulated independently of the underlying task. They evaluate explanation properties rather than the model's predictive objective and can, in principle, be applied to generative or other non-discriminative tasks. Some adaptations may still be required; for instance, metrics relying on predictive performance (e.g., accuracy or related scores) should employ task-appropriate alternatives such as likelihood-based or reconstruction-based measures.

### 5.1 Practical Guidance

While this work enables researchers to gain a comprehensive overview of the field of VXAI and the metrics proposed in the literature, we acknowledge that the vast number and diversity of existing metrics pose a major challenge for practitioners when selecting suitable ones. As comprehensive selection guidelines remain an open research topic, we offer the following practical considerations derived from our analysis.

First, we encourage researchers and practitioners to address the issue of anecdotal evaluation by systematically applying quantitative metrics rather than relying solely on qualitative or illustrative examples. Using established metrics supports comparability across studies and strengthens the validity of evaluation results.

To select suitable metrics, practitioners should begin by assessing their test case according to the categorization scheme (Subsection 4.1) to identify the appropriate explanation type, as this determines the subset of applicable metrics. Next, they should identify the key desiderata that are most relevant for their goals and focus their evaluation on these aspects. We particularly emphasize that, when in doubt, Fidelity is the most critical property from a trustworthiness perspective, while Parsimony is often the most practical consideration for ensuring the accessibility and interpretability of explanations.

Metrics from higher Contextuality stages are typically of greater interest to researchers aiming to evaluate an XAI method in general. In contrast, such metrics (especially those belonging to Contextuality stage V) are usually less suitable for practitioners operating within a fixed application setup. Once the relevant subset of metrics has been identified, the final choice should be guided by a careful comparison of their detailed descriptions, as multiple metrics within the same category may still follow different rationales and address distinct aspects. The metric summaries in Appendix C also include the corresponding references, which can serve as a valuable starting point for further reading and understanding of practical use cases.

To achieve a more comprehensive evaluation, we recommend employing several complementary metrics, ideally with different but reasonable hyperparameter configurations. Even in the absence of established benchmarks or threshold values, practitioners can still gain valuable insights by comparing their explanations against naïve or random baselines, as it is generally known for each metric whether higher or lower scores indicate better performance.

## 6 Conclusion, Limitations & Future Work

**Conclusion**

In this survey, we introduced a unified and comprehensive framework for the evaluation of XAI. We conducted a systematic review of the literature, identifying original metrics designed to assess explanation quality. These were grouped into 41 aggregated metrics and categorized using a three-dimensional scheme based on **desideratum**, **explanation type**, and **contextuality**.

Our analysis reveals a broad range of available metrics, covering diverse use cases and goals. At the same time, it highlights the lack of consensus regarding when and how specific metrics should be applied. Moreover, we find that many existing metrics are not yet adapted for all explanation types, indicating potential for extension and generalization.

Beyond providing the most extensive synthesis of evaluation metrics to date, the VXAI framework advances the field conceptually and organizationally. It introduces a structured and extensible taxonomy that enables practitioners and researchers to systematically analyze, compare, and select suitable metrics for their evaluation goals. Unlike prior reviews, VXAI supports both *horizontal* extension (i.e., incorporating new desiderata, explanation types, or contextuality levels) and *vertical* extension (adding new metrics within the existing structure). This flexibility allows the framework to evolve in step with emerging explanation paradigms and evaluation needs.

Overall, VXAI provides a foundation for more transparent, comprehensive, and methodologically grounded evaluation practices in XAI. It facilitates comparability across studies, encourages reproducible metric application, and paves the way toward standardized evaluation strategies.

**Limitations and Future Work**

This work focuses exclusively on surveying and categorizing existing metrics. Although we provide structured comparisons and conceptual insights, we do not conduct empirical benchmarking. Future work should investigate how different metrics behave in practice, under which conditions they agree or contradict, and what trade-offs they impose. This would support the development of a more standardized benchmarking suite for explanation evaluation.

Establishing practical thresholds, studying alignment between metrics and desiderata, and exploring aggregation strategies remain open and important questions. Ideally, this could lead to a broadly accepted set of evaluation standards and protocols for verifying explanations across different models and tasks.

Lastly, while many metrics are labeled by us as potentially applicable to additional explanation types, their use remains unvalidated in literature. Future work should adapt and extend these metrics to underexplored explanation types, particularly for natural language and concept-based explanations.

**Acknowledgment**

This research is funded by the German Federal Ministry for Digital Transformation and Government Modernisation (BMDS) as part of the project *MISSION KI - Nationale Initiative für Künstliche Intelligenz und Datenökonomie* with the funding code 45KI22B021.

We would like to thank our colleagues Kevin Iselborn and Jayanth Siddamsetty for reviewing this survey and sharing helpful comments and insights throughout its development. We also gratefully acknowledge the anonymous TMLR reviewers, whose valuable feedback helped improve the clarity and quality of this work.

# A    Detailed Literature Search Procedure

This appendix provides the complete methodological details of our systematic literature review, expanding upon the summary presented in Section 3. It includes a full description of the search strategy, databases and queries, screening procedure, inclusion and exclusion criteria, and intermediate statistics. The review followed the PRISMA guidelines (Page et al., 2021) to ensure transparency and reproducibility.

**Preliminary Consideration**

The research was preceded by two key observations: First, searching for general XAI terms is not feasible, as the database is intractable (searching Google Scholar for "XAI" gives over 200,000 results). Second, VXAI metrics are usually introduced alongside an XAI method rather than in a dedicated publication, and there exists no unified vocabulary, making it hard to identify relevant sources from keyword searches in titles or abstracts alone. Hence, we decided to split our research into two stages. First, we performed a database search using a limited set of search terms to build an initial body of potentially relevant sources. The first stage was designed primarily to identify *secondary literature* (e.g., reviews and surveys) that could serve as an entry point for the second, snowballing stage. In the second stage, we expanded this body through recursive backward snowballing by reviewing references of the already identified publications. This design explains why comparatively few papers were directly included from Stage 1, while Stage 2, which targeted the *primary literature* containing original metric proposals, yielded a substantially larger number of inclusions.

**Stage 1: Initial Search**

We started with a database meta-search using **Scopus**[4], **IEEE**[5], and **ACM**[6], to identify existing reviews and surveys that point towards further evaluation metrics. Using the advanced search features of each database, we designed research queries based on the following key terms:

```
[ Explain∗ | Interpret ∗]
              ×[7]
[AI | Artificial Intelligence | ML | Machine Learning]
         AND
[Evaluation | Metric | Quantification]
         AND
[Survey | Review]
```

The exact search strings for each database are provided in Subsection A.1. The last search was conducted on January 15, 2025.

This resulted in a total of 673 identified articles after de-duplication. We first screened titles and abstracts, excluding sources that were not research articles, did not involve XAI or VXAI, or focused on human-based evaluation. Through this first screening, we excluded 504 articles, using the following exclusion criteria:

- ER-1: General Issues
  - ER-1a: Not a research article
  - ER-1b: Not English
- ER-2: No XAI (or VXAI) at all, or with explainability outside our scope (e.g., ante-hoc data analysis)
- ER-3: Contains XAI, but no systematic evaluation of XAI (e.g., "anecdotal evidence")
- ER-4: Contains systematic VXAI, but only human-centered methods

Out of the 169 articles that passed the initial title and abstract screening, 9 could not be retrieved. For the remaining 160 articles, we conducted full-text screening, applying the exclusion criteria described above. To

---

[4] https://www.scopus.com/search/form.uri?display=advanced
[5] https://ieeexplore.ieee.org/search/advanced/command
[6] https://dl.acm.org/search/
[7] The operator × denotes the concatenation of terms into composite search phrases (e.g., "Explain* AI", "Interpret* ML").

avoid an intractable number of limited-value sources, we further excluded works which used functionality-grounded VXAI metrics already introduced by other articles, introducing the following exclusion criterion:

- ER-5: Contains functionality-grounded evaluation, but does not introduce a new metric itself.

Instead, we added the cited metric to our corpus for the snowballing phase, yielding 515 potentially relevant sources. Of the articles retrieved through the initial database search, only 6 were included in our review.

To complement the structured database search, we also conducted an unstructured manual search using Google Scholar and other sources such as personal literature databases, citation alerts, and colleague recommendations. This additional search yielded 65 articles, of which 39 were included directly in the review. A further 20 references from these articles were added to the corpus for the snowballing phase. Duplicates already identified during the database search were not counted again, and duplicates among citations from the manual search were likewise removed. We did not specifically search preprint repositories such as arXiv, but relevant preprints that appeared through other channels were included if they met our inclusion criteria.

In total, during the first stage, we included 45 articles in our review (6 from the database search and 39 from additional sources). We also compiled 535 additional references for the second-stage backward snowballing (515 from the database search and 20 from the additional sources).

**Stage 2: Snowballing**

Starting with the corpus of 535 references identified in the first stage, we conducted full-text screening on all records as described above. During the recursive backward snowballing process, we identified and assessed an additional 98 references for eligibility. We observed that the set of relevant records quickly began to converge: most newly encountered papers that did not present original metrics (ER-5) cited articles already included in our corpus. We did not perform forward snowballing (i.e., identifying articles that cite our included works), as our focus was on original metric proposals. Including such follow-up papers would have significantly increased the number of records beyond a manageable scope.

In total, we assessed 628 articles during the second phase (535 from the first stage and 98 from recursive snowballing). Out of these, we included 317. Compared to the 20% inclusion rate in the first stage, the nearly 50% inclusion rate in the second phase confirms the effectiveness of our strategy. The initial database search primarily yielded secondary sources (e.g., XAI and VXAI reviews), which helped identify original metric proposals during snowballing.

Overall, we reviewed $1,459$ articles, screened 866 in full, and included 362 that originally proposed a VXAI metric or one of its variants.

### A.1 Queries

**Scopus**

```
TITLE–ABS–KEY (
  ("Explain* AI" OR "Explain* Artificial Intelligence" OR "Explain* ML" OR "
      Explain* Machine Learning" OR
  "Interpret* AI" OR "Interpret* Artificial Intelligence" OR "Interpret* ML"
    OR "Interpret* Machine Learning")
  AND
  ("Evaluation" OR "Metric" OR "Quantification")
  AND
  ("Survey" OR "Review")
)
```

**IEEE Access**

```
("All Metadata":"Explain* AI" OR "All Metadata":"Explain* Artificial
    Intelligence" OR "All Metadata":"Explain* ML" OR "All Metadata":"Explain*
    Machine Learning" OR
"All Metadata":"Interpret* AI" OR "All Metadata":"Interpret* Artificial
    Intelligence" OR "All Metadata":"Interpret* ML" OR "All Metadata":"
    Interpret* Machine Learning")
AND
("All Metadata":"Evaluation" OR "All Metadata":"Metric" OR "All Metadata":"
    Quantification")
AND
("All Metadata":"Survey" OR "All Metadata":"Review")
```

**ACM Digital Library**

```
ANYWHERE:[
    ("Explain* AI" OR "Explain* Artificial Intelligence" OR "Explain* ML" OR "
        Explain* Machine Learning" OR
    "Interpret* AI" OR "Interpret* Artificial Intelligence" OR "Interpret* ML"
        OR "Interpret* Machine Learning")
    AND
    ("Evaluation" OR "Metric" OR "Quantification")
    AND
    ("Survey" OR "Review")
    ]
```

# B    Desiderata of XAI

*This appendix expands on the concise outline of desiderata used in the categorization scheme (see Subsection 4.1) and provides the full background and definitions.* A well-founded evaluation of XAI methods requires clearly defined criteria for what constitutes a good explanation. To establish such criteria, we must first reflect on the role of explanations in the context of XAI. According to the definition in the XAI Handbook (Palacio et al., 2021), explaining a model and its behavior is a two-stage process: first, factual information about the model's decision process is generated (the explanans); this is then interpreted by the human user. The first stage can be evaluated using technical criteria that assess whether the model's reasoning has been captured truthfully and reliably. The second stage depends on the interpretability of the explanation, which can be assessed using general cognitive principles, even in the absence of a specific user model.

To capture the multifaceted nature of explanation quality, a number of desiderata have been proposed in the literature. We interpret these as functionality-grounded expectations that reflect the demands of both stages of the explanation process. We first provide an overview of existing desiderata proposed in prior work, then introduce a unified set that systematically describes the requirements for ensuring technical soundness and for bridging the interpretation gap.

## B.1    Common Formulation of Desiderata

Several XAI surveys report that there is no ubiquitous consensus on appropriate desiderata, with some of the categories related to goals pursued *through* XAI, rather than standalone desiderata of XAI, e.g., Trustworthiness, Acceptance, or Fairness (Doshi-Velez & Kim, 2017; Langer et al., 2021; Vilone & Longo, 2021; Elkhawaga et al., 2023). Hence, we conduct a scoping review, reporting the main desiderata used by different authors and analyzing the similarities as well as differences in their formulations. For the sake of brevity, we exclude some of the papers listed in Table 1, as the missing ones either overlap considerably (e.g., Awal & Roy (2024) and Klein et al. (2024)), rely on a different notion of desiderata (e.g., Sovrano et al. (2021)), or use no desiderata at all. We first present these frameworks using the authors' original terminology before introducing our own categorization scheme.

The famous **Co-12** properties, introduced by Nauta et al. (2023) and reused by Le et al. (2023), form one of the most extensive existing frameworks for categorizing XAI metrics. They group properties along three dimensions: Content (*Correctness, Completeness, Consistency, Continuity, Contrastivity, Covariate Complexity*), Presentation (*Compactness, Composition, Confidence*), and User (*Context, Coherence, Controllability*). While the first dimension focuses on information contained in the explanans, the second and third dimensions concern how this information is conveyed. Although some of these human-centered properties can be measured through proxies, others may mainly be assessed through human-grounded evaluation.

Zhou et al. (2021), based on the taxonomy of Markus et al. (2021), define *Interpretability* and *Fidelity* as the two major components of explainability. The former is concerned with providing understandable explanantia and includes the properties of *Clarity*, *Broadness*, and *Parsimony*. Fidelity, refers to how accurately an explanans reflects the model's behavior, and consists of the properties of *Completeness* and *Soundness*.

The framework proposed by Robnik-Šikonja & Bohanec (2018), which was adopted by Carvalho et al. (2019) and Molnar (2020), differentiates between properties of explanations and individual explanantia. Investigating the properties of methods (i.e., explanations), they consider *Translucency*, *Portability*, and *Algorithmic Complexity*, which can all be interpreted as desiderata, while *Expressive Power* is a descriptive property. The properties of individual explanantia include *Comprehensibility*, *Importance*, *Representativeness*, *Fidelity*, and *Stability*. However, their categorization encompasses further properties, which we do not consider as proper desiderata of XAI: *Accuracy*, *Novelty*, *Certainty*, and *Consistency*. Accuracy is a measurement of the underlying black-box model, while Novelty and Certainty are rather explanantia themselves than properties of general explanations. Further, Consistency between different black-box models is not necessarily a useful measure, as different models may derive similar predictions based on different reasoning (see Rashomon Effect (Breiman, 2001; Leventi-Peetz & Weber, 2022)).

The famous XAI review by Guidotti et al. (2018), inspired by earlier works such as Andrews et al. (1995) and Johansson et al. (2004), reports three less fine-grained desiderata: *Interpretability*, *Fidelity*, and *Accuracy*.

Similar to previously discussed reviews, Interpretability describes human understandability, while Fidelity measures how well the explanans imitates the black box, and Accuracy focuses on predictive performance, which is outside the scope of XAI in our context. Additionally, *Consistency* is introduced by Andrews et al. (1995), expecting reproducible explanations, while Johansson et al. (2004) emphasize the explanation's algorithmic *Scalability* and *Generality*.

Alvarez-Melis & Jaakkola (2018b), Jesus et al. (2021), and Alangari et al. (2023a) all report a similar set of desiderata. The understandability of explanations is measured in terms such as *Interpretability*, while *Faithfulness* and the corresponding desiderata give insight into how truthful the explanation is to the underlying black-box model. All three works further report the *Stability* of explanations as a desired property, assessing whether explanantia on similar inputs are similar.

In their Quantus toolkit, Hedström et al. (2023) (and the follow-up study by Bommer et al. (2024) as well), categorize their metrics partly through desiderata, namely *Faithfulness*, *Robustness*, and *Complexity*. Simultaneously, part of their metrics are grouped by their conceptual similarity, including *Localization*, *Randomization*, and *Axiomatic*.

Finally, the Compare-xAI benchmark by Belaid et al. (2022) organizes functional tests into six categories, namely *Fidelity*, the robustness-related *Stability* and *Fragility*, the interpretability desideratum *Simplicity*, and the explanation-methods-focused *Stress* and *Portability* (which they integrate under "Other").

While many existing frameworks overlap conceptually, a unified and practically usable categorization scheme for VXAI metrics is still lacking. This requires a structured set of desiderata that defines what makes a good explanation and supports consistent classification of metrics. Prior work often enforces a rigid one-to-one mapping between metrics and desiderata; in contrast, we decouple these dimensions, defining a set of mostly independent desiderata to which each metric may contribute individually or jointly. Lightweight frameworks tend to omit critical aspects of explanation quality, while broader ones sometimes include goals that are not intrinsic to the explanation itself (e.g., accuracy). We restrict our scope to properties that reflect the explanation rather than the underlying model and clarify excluded cases after presenting our set. Although all desiderata rely on proxies, we limit ourselves to properties that are quantifiable in principle. Highly abstract or vague notions lacking empirical grounding are omitted. Lastly, our framework is designed to be extensible, allowing the integration of future desiderata as the field evolves.

### B.2 Proposed Set of Desiderata

Building on the desiderata established above and our findings on VXAI metrics, we propose a set of seven desiderata to serve as a categorization scheme for VXAI.

Our goal is to offer a principled yet practical structure that enables consistent classification while avoiding the limitations of prior definitions. These are either too narrow to accommodate relevant metrics or too broad and include properties beyond explainability. While properties such as fairness are often measured using XAI methods, we consider them beyond the scope of VXAI, because they assess the model's behavior rather than the explanation itself.

Building on the two-stage view of explaining described by Palacio et al. (2021) (i.e., presenting factual information followed by human interpretation), we define two complementary dimensions of explanation quality. The **Technical** (T) dimension comprises desiderata that assess the factual correctness, robustness, and reliability of the explanation, ensuring that it faithfully reflects the model's reasoning. In contrast, the **Interpretability** (I) dimension captures how the explanation is conveyed and how accessible, intuitive, and useful it is to a general-purpose user. This separation is aligned with existing frameworks such as the Co-12 properties (Nauta et al., 2023) and the taxonomy by Zhou et al. (2021). The desiderata are designed to be as independent from each other as possible, allowing for reliable quantification of different aspects relevant to trustworthy XAI.

We present our set of desiderata and its relation to other frameworks in Figure 6 and introduce them in more detail in the following paragraphs. In total, we define seven desiderata, two associated with Interpretability and five belonging to the Technical dimension:

(I) **Parsimony**: The explanation should keep the explanans concise to support interpretability.
(I) **Plausibility**: The explanation should shape the explanans to align with human expectations.
(T) **Coverage**: The explanation should provide an explanans for every explanandum.
(T) **Fidelity**: The explanation should make the explanans reflect the model's true reasoning.
(T) **Continuity**: The explanation should ensure that similar explananda yield similar explanantia.
(T) **Consistency**: The explanation should produce stable explanantia across repeated evaluations.
(T) **Efficiency**: The explanation should compute the explanans efficiently and broadly.

### B.2.1 Parsimony

> The explanation should keep the explanans concise to support interpretability.

The primary purpose of an explanation is to convey information about the black-box model or its decision process to humans. Therefore, the resulting explanans must be expressed in a way that the human mind can grasp easily to increase the explanations success. While actual interpretability can only be evaluated through human-grounded evaluation, Parsimony is one of the most prevalent proxies defined to serve as a functionality-grounded approximation. Since our mental capacity is limited and we tend to struggle with an overload of information (Miller, 1956; 2019; Alangari et al., 2023a), providing short and simple explanantia helps humans understand more effectively.

Nauta et al. (2023) introduce the property of Compactness, arguing that a briefer explanans is easier to understand. Similarly, they use Covariate Complexity to assess how complex the features are that constitute the explanans, where higher interpretability is supported by providing a few high-level concepts in favor of a very granular explanans. Both of these aspects are summarized under Parsimony by Markus et al. (2021) and Zhou et al. (2021), preferring simpler explanantia over longer or more complex ones. The scheme used in the Quantus library (Hedström et al., 2023; Bommer et al., 2024) defines a group called Complexity. It specifically tests for concise explanantia and aims to have as few features as possible to be easier to understand. The associated interpretability desiderata from other authors are defined less explicitly, but similarly favoring simpler explanantia (Andrews et al., 1995; Johansson et al., 2004; Guidotti et al., 2018), proposing that simple explanantia should be short (Alvarez-Melis & Jaakkola, 2018b; Jesus et al., 2021; Alangari et al., 2023a), promoting small explanantia and only focusing on relevant parts (Robnik-Šikonja & Bohanec, 2018; Carvalho et al., 2019; Molnar, 2020), and expecting an explanans with concentrated information to facilitate human understanding (Belaid et al., 2022).

Following the proposed definitions, we include Parsimony as one of our desiderata. It expects explanantia to be as brief and concise as possible, to ensure that rationales can be understood easily and fast. We focus Parsimony exclusively on this aspect, as other associated properties are either covered by separate desiderata (such as truthfulness of the explanation) or excluded entirely as they are not functionality-grounded (general understandability of the explanation).

### B.2.2 Plausibility

> The explanation should shape the explanans to align with human expectations.

To improve the acceptance of explanations and facilitate their interpretation, the Plausibility desideratum is contained in most authors' interpretability desiderata. Coherence as defined by Nauta et al. (2023) is the accordance of an explanans with the user's previous knowledge and expectations. The metrics classified under Localization by Hedström et al. (2023) evaluate whether an explanation shows a rationale similar to what humans would expect. This is concordant with the definition of Comprehensibility (Alvarez-Melis & Jaakkola, 2018b; Jesus et al., 2021; Alangari et al., 2023a), which states that an explanans should be similar to what a human expert would choose as the correct rationale. Furthermore, Nauta et al. (2023) introduce Contrastivity, which supports Plausibility, as an explanans should be specific to the given explanandum. Similarly, Clarity is introduced by Markus et al. (2021) and Zhou et al. (2021), expecting explanantia to be unambiguous.

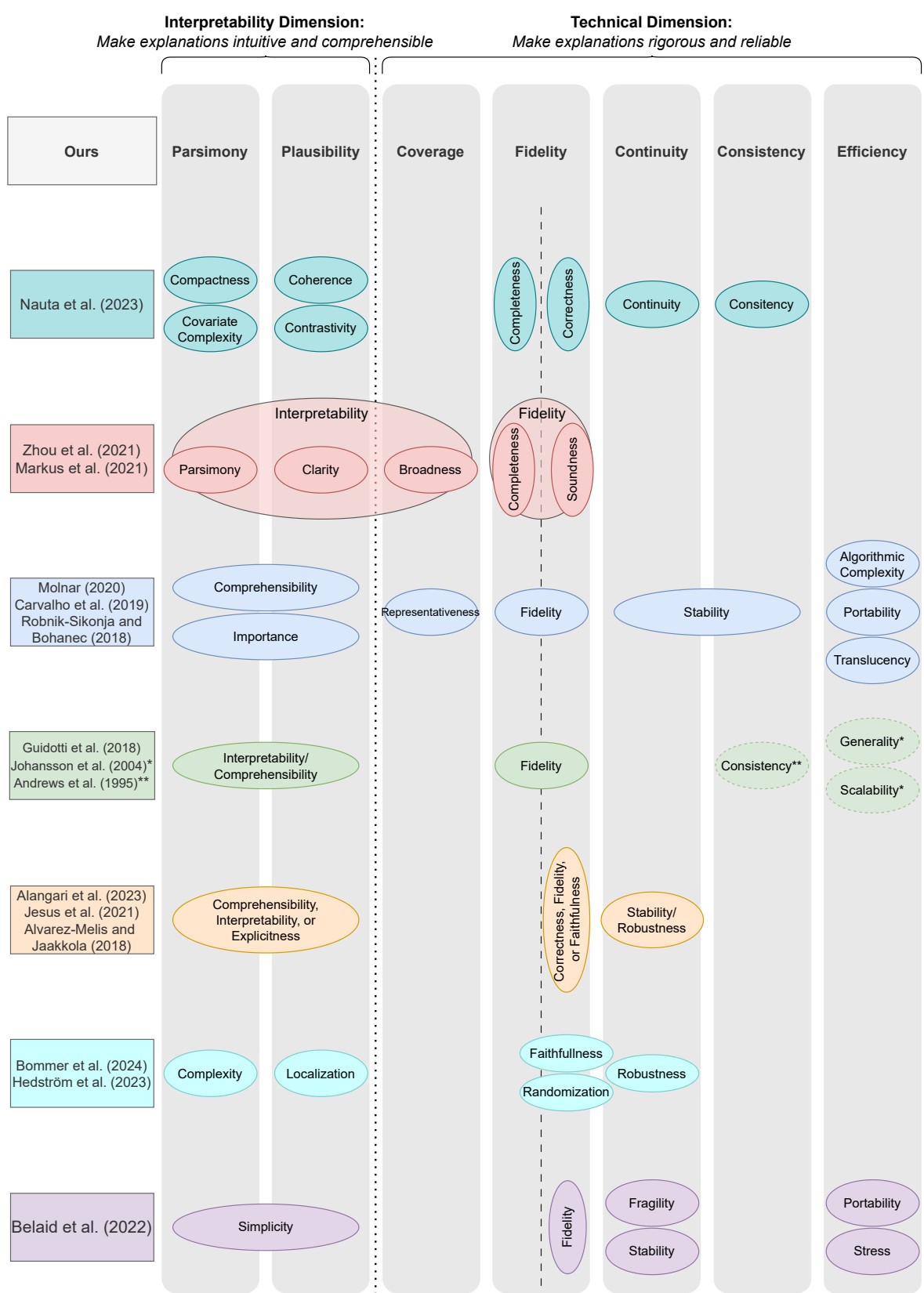

Figure 6: Comparison of our desiderata with those used in related work. Desiderata outside our framework are omitted. Desiderata reported by only individual sources (per row) are denoted by an asterisk (*).

We include the Plausibility desideratum, which encompasses the idea that explanations should align with human knowledge and intuition. On one hand, this includes human expectations towards the result (explanans), e.g., "The model focuses on what a human would focus on". On the other hand, the XAI methods' behavior (explanation) should also be aligned with human intuition, e.g., "The outputs for individual inputs should differ".

### B.2.3  Coverage

> The explanation should provide an explanans for every explanandum.

The extent to which an explanation or explanans can be applied is considered by two frameworks. Unfortunately, definitions from both surveys are vague. Markus et al. (2021) and Zhou et al. (2021) define the Broadness of an explanation as "how generally applicable" it is, without further elaboration on the implications of this definition. More concretely, Representativeness is presented by Robnik-Šikonja & Bohanec (2018), Carvalho et al. (2019), and Molnar (2020). It reflects the number of explananda that are covered by an individual explanans, although this definition focuses mainly on the distinction between global and local explanation methods.

To add more clarity to these definitions, we include Coverage with an alternative definition. It defines the amount of explananda that are covered by the explanation, i.e., reflecting whether there exists an explanans for every data input or output.

### B.2.4  Fidelity

> The explanation should make the explanans reflect the model's true reasoning.

Fidelity is one of the most frequently discussed concepts in the literature and combines two closely related aspects: Correctness and Completeness. While some works introduce these as separate desiderata, others group them together under the umbrella of Fidelity.

Correctness refers to whether the explanation truthfully represents the internal logic and decision process of the black-box model. It is one of the most frequently emphasized desiderata across the reviewed frameworks. Without correctness, even the most interpretable or simple explanation may provide no meaningful insight. Terms like Faithfulness, Truthfulness, and Fidelity are often used interchangeably in literature to describe this idea. Correctness encompasses both local fidelity for individual explanantia and global alignment across the dataset (Robnik-Šikonja & Bohanec, 2018; Carvalho et al., 2019; Molnar, 2020). The general consensus is that an explanation should reveal what truly drives the model's outputs (Alvarez-Melis & Jaakkola, 2018b; Markus et al., 2021; Zhou et al., 2021; Belaid et al., 2022; Alangari et al., 2023a; Nauta et al., 2023). It is commonly assessed by how well the explanation reflects or mimics the model's behavior (Andrews et al., 1995; Johansson et al., 2004; Guidotti et al., 2018).

Completeness, in contrast, describes how much of the model's reasoning is captured by the explanation. According to the Co-12 properties by Nauta et al. (2023), an explanation should ideally include the full scope of the model's rationale. Some authors treat Completeness as a sub-aspect of Fidelity (Markus et al., 2021; Zhou et al., 2021), while others define Fidelity itself as the capacity to capture all of the information embodied in the model (Andrews et al., 1995; Johansson et al., 2004; Guidotti et al., 2018).

Although it is theoretically possible to have an explanation that is partially correct but incomplete (e.g., providing a heatmap that highlights only one of several relevant features), or complete but partially incorrect (e.g., including all the right features alongside irrelevant ones), neither scenario is desirable. If key features are missing or irrelevant ones are included, the explanans ultimately misrepresents the model's behavior. While Correctness and Completeness can be distinguished conceptually, they are tightly interwoven in practice and difficult to evaluate in isolation. Since our desiderata are intended to capture orthogonal evaluation dimensions, and these two cannot be meaningfully disentangled, we combine them under the unified criterion of Fidelity.

### B.2.5 Continuity

> The explanation should ensure that similar explananda yield similar explanantia.

Just as the robustness or stability of a standard AI model is of great interest, similar expectations apply to explainability. Most frameworks highlight this desideratum, and various metrics have been proposed to assess how stable or reliable explanations are. However, the terminology used in literature is inconsistent, at times overlapping and at other times diverging in meaning.

Nauta et al. (2023) introduce the term Continuity as the smoothness of the explanation, i.e., similar explananda should yield similar explanantia. Others refer to this idea as Stability (Robnik-Šikonja & Bohanec, 2018; Carvalho et al., 2019; Molnar, 2020), describing it as the resilience against slight variations in input features that do not alter the model's prediction. The term Robustness is used by Alvarez-Melis & Jaakkola (2018b), Jesus et al. (2021), and Alangari et al. (2023a) to describe the same behavior, referring to it as a key requirement for trustworthy XAI.

The Quantus toolkit reflects the prevalence of this concept, providing a "Robustness" metric category (Hedström et al., 2023; Bommer et al., 2024), which assesses the similarity of explanantia under minor changes in input. Finally, Belaid et al. (2022) cover the same idea under the term Stability. In addition, they assess Fragility, which they define as the resilience of explanations against malicious manipulation, such as adversarial attacks.

Our Continuity desideratum covers both of the mentioned properties. It includes the smoothness of explanations with respect to "naïve" changes in the explanandum that ideally do not affect the model's behavior, as well as the resilience of explanations against malicious manipulation attempts. Note that this includes changes over the input data as well as the model. We adopt the term *Continuity* instead of Stability or Robustness to reduce possible confusion with model robustness.

### B.2.6 Consistency

> The explanation should produce stable explanantia across repeated evaluations.

While Continuity investigates the smoothness and similarity between similar but different inputs, the Consistency of explanations for identical inputs also needs to be considered. However, few authors explicitly consider this desideratum.

Consistency is introduced by Nauta et al. (2023) as a direct measure of the determinism of an XAI algorithm. Similarly, one part of the definition of Stability by Robnik-Šikonja & Bohanec (2018), Carvalho et al. (2019), and Molnar (2020) considers variations in explanations based on non-determinism. The oldest formulation of Consistency is given by Andrews et al. (1995) and considers explanation methods to be consistent when they produce equivalent results under repetition.

However, several frameworks additionally consider the similarity of explanantia generated from different models trained on the same data (Robnik-Šikonja & Bohanec, 2018; Carvalho et al., 2019; Molnar, 2020; Nauta et al., 2023). Yet, different models can produce the same prediction while relying on entirely different internal reasoning. This is especially true as there are often multiple valid reasons for the same event, also known as the *Rashomon Effect* (Breiman, 2001; Leventi-Peetz & Weber, 2022).

We include Consistency using the initial formulations, i.e., explanations should be deterministic or self-consistent, always presenting the same explanans for identical explananda. While the latter definition is present in one of the identified metrics, we do not explicitly add it to the definition of our Consistency desideratum, as we do not believe that different explananda (inputs), i.e., different models, necessarily result in identical explanantia (outputs).

### B.2.7    Efficiency

> The explanation should compute the explanans efficiently and broadly.

Finally, out of practical considerations, we want explanations to be conveniently applicable. This includes considering the range of models or situations in which the algorithm can be effectively applied. Simultaneously, it also includes the time it takes to compute an individual explanans.

The first property is introduced as Portability and Translucency throughout literature (Robnik-Šikonja & Bohanec, 2018; Carvalho et al., 2019; Molnar, 2020). Portability is the variety of models for which an explanation can be used, while Translucency is the necessity of the explanation algorithm to have access to the internals of the model. Similarly, Johansson et al. (2004) measure Generality, given by the restrictions or overhead necessary to apply an explanation to specific models. Belaid et al. (2022) refer to Portability as the diverse set of models to which the explanation can be applied.

Secondly, the Algorithmic Complexity (Robnik-Šikonja & Bohanec, 2018; Carvalho et al., 2019; Molnar, 2020) considers the time it takes to generate an explanans. Naturally, the necessary time depends not only on the inherent complexity of the explanation algorithm but also on Scalability, i.e., its ability to efficiently handle larger models and input spaces (Johansson et al., 2004). Using the "Stress test", Belaid et al. (2022) explicitly evaluate the runtime behavior with respect to increasing input size.

We subsume both of these aspects under a general desideratum called Efficiency. It includes the algorithmic or computational properties of the explanation, which might influence the choice of a specific XAI algorithm over another.

### B.2.8    Excluded Desiderata

Several of the desiderata introduced in the referenced frameworks were not included in our catalog. We briefly present each of these and justify why they were excluded.

In the Co-12 properties for XAI, Nauta et al. (2023) introduce **Controllability**, which we exclude for two reasons. First, it measures the extent to which a user can interact with the explanans, which can be relevant in given applications, but is a property of the *presentation* and not necessarily the explanans itself. Second, the only measurement they provide is through user interaction, which we avoid in this study, as we focus on functionality-grounded metrics. Their **Composition** property is not measurable either, as it describes the explanans' presentation format. Similarly, **Expressive Power** (Robnik-Šikonja & Bohanec, 2018; Carvalho et al., 2019; Molnar, 2020) describes the format of the explanans and is therefore not included.

The **Context** property (Nauta et al., 2023) describes how relevant a given explanans is to a user. This is mostly covered in the Plausibility desideratum, although their definition is more focused on the needs of specific users and hence is very subjective and not measurable through proxies.

Further, both **Confidence** (Nauta et al., 2023) and the closely related **Certainty** (Robnik-Šikonja & Bohanec, 2018; Carvalho et al., 2019; Molnar, 2020) are excluded. They both describe whether confidence scores of the model's decisions or the explanans are displayed. However, this is not a general desideratum for explanations, but can rather be seen as providing assisting information, which is also an explanans in itself, and is then subject to all the desiderata of our framework. Related to this, **Novelty** (Robnik-Šikonja & Bohanec, 2018; Carvalho et al., 2019; Molnar, 2020) refers to whether data instances lie within or outside the training distribution. While relevant to assessing model behavior or uncertainty, this property does not characterize the quality of the explanation itself. Instead, it serves as contextual information that may inform or accompany an explanans, but is not an intrinsic requirement for explainability.

As discussed in Subsection B.2.6, we do not include some of the aspects of **Consistency** as defined by Robnik-Šikonja & Bohanec (2018), Carvalho et al. (2019), and Molnar (2020), who expect several models to generate similar explanantia. This is rooted in the assumption that different models should all follow the same reasoning, which we reject. In contrast, Anders et al. (2020) showed that two models exhibiting the same external behavior, such as predictions, can have radically different inner workings.

While it is a standard metric during model training, **Accuracy** of explanations is also reported by several frameworks (Andrews et al., 1995; Johansson et al., 2004; Guidotti et al., 2018; Robnik-Šikonja & Bohanec, 2018; Carvalho et al., 2019; Molnar, 2020). Importantly, this is only meaningful for specific types of explanations, such as white-box surrogate models (e.g., decision trees; see Subsection 4.1.3 for a definition). It is different from "surrogate fidelity", since this measures performance against the black-box model's predictions and not the ground truth. We restrict our framework to explanantia derived from black-box models, rather than evaluating interpretable models that are directly trained on the data as predictive models themselves.

Lastly, the Quantus toolkit (Hedström et al., 2023; Bommer et al., 2024) includes a category labeled as **Axiomatic**, which groups metrics based on their functionality. Specifically, it refers to whether they test for formal properties. This categorization reflects the underlying mechanism of the metric rather than a specific quality criterion. Accordingly, we reassign these metrics to the corresponding desiderata they evaluate.

### B.3 Considerations

We conclude this presentation of desiderata with a few final considerations.

First, all seven desiderata are introduced as independent dimensions. However, in practice, they cannot always be separated entirely, which is reflected in some metrics being associated with multiple desiderata. Especially where a single criterion in other authors' frameworks covers multiple of our desiderata, an inherent connection is given. This can be seen especially in Consistency and Continuity, which both measure some sort of robustness. Conversely, some of our desiderata subsume multiple sub-aspects themselves. This applies especially to Fidelity, which is comprised of Correctness and Completeness.

Second, a truly useful explanation can only be achieved when considering the interplay between the individual desiderata. Otherwise, trivial explanantia can be constructed to optimize individual desiderata, as we illustrate with two examples: a) A saliency map that highlights the entire input image is certainly complete, as it encompasses all relevant features; however, it is hardly parsimonious and unlikely to be correct, as not every region is truly decisive for the prediction. b) A concept explanation that identifies the same concept regardless of the input may be maximally consistent, continuous, and parsimonious; nevertheless, it will unlikely be either correct or complete, as it disregards input-specific variation.

Further, to ensure the usefulness of XAI to humans, we believe that some hierarchy exists between our defined desiderata. While an exact ordering is implausible, as it depends on the exact setting and needs, at least some general guidelines can be given: **Fidelity** is the foundation of the entire evaluation process. If there is no necessity for an explanation to be correct and complete, it can be completely arbitrary and will not necessarily give any true insights into the underlying model. This would undermine the entire purpose of XAI. If important parts of the rationale are excluded in the explanans, it may result in an incomplete (and hence incorrect) understanding of the decision process. While not as crucial, but still important, **Consistency** and **Continuity** considerably contribute to the trustworthiness of XAI. Without them, explanations may vary unpredictably, as explanantia can change without reasons rooted in the model's behavior. Given these four desiderata, the explanation can be expected to give solid results from a technical viewpoint. Hence, the interpretability dimensions, **Parsimony** and **Plausibility**, can be considered next, as they facilitate understanding but do not assess whether an explanation is actually reliable. Finally, **Coverage** and **Efficiency** are useful properties of an explanation, but can be seen as a bonus, making them readily applicable to a wide set of explananda, including data and models. However, depending on the use case and context, individual desiderata may be weighted differently or even disregarded entirely, particularly when some aspects are less relevant for a specific application or domain.

Finally, the framework is based on the works presented by other authors, as well as the metrics we identified through our systematic survey. While we consider it extensive for now, it is possible to expand it horizontally in the future. Depending on the requirements, further desiderata may be added. We do not rigorously map each metric to a single desideratum, but instead list several desiderata it contributes to. Therefore, our proposed categorization can be readily extended with further desiderata.

## C   VXAI Metrics

This appendix details all metrics identified in our literature survey and included in the VXAI framework, as summarized in Table 2. An interactive version of the framework is available at https://vxai.dfki.de/.

### C.1   Notation

The mathematical expressions used throughout the appendix are intended to clarify concepts. We do not enforce strict formalism as long as the notation remains unambiguous. In cases where clarity is not compromised, we slightly overload notation, but specific meanings are always disambiguated by context. While the notation is predominantly tailored to classification tasks, where the model assigns scores to discrete class labels, it can be adapted to suit regression settings or, in part, other paradigms. We define the following symbols, which are used across all metrics:

- Let $\mathcal{X}$ and $\mathcal{Y}$ denote the input (data) and output (label) spaces, respectively.
- The black-box model is defined as a scoring function $\theta : \mathcal{X} \times \mathcal{Y} \to \mathbb{R}$ that assigns a real-valued predictive score (e.g., logits) to each input–label pair.
- For a given input $x \in \mathcal{X}$, we write $\theta(x)$ as shorthand for the vector of scores across all labels in $\mathcal{Y}$, i.e., $\theta(x) := (\theta(x, y_1), \theta(x, y_2), \ldots, \theta(x, y_k))$.
- The predicted label is then given by $\hat{y}_x := \arg\max_{y \in \mathcal{Y}} \theta(x, y)$.
- We denote a second input as $x' \in \mathcal{X}$ and a perturbed version of $x$ as $\dot{x}$ (see Subsection C.2.1).
- Let $\mathcal{X}_y \subseteq \mathcal{X}$ be the set of all instances with ground-truth class $y$.

- An explanandum is a triple $(\theta, x, y)$, where $\theta$ is the model, $x \in \mathcal{X}$ is the input instance, and $y$ is the label to be explained (which is often, but not necessarily, $\hat{y}$).
- An explanans for a given explanandum is denoted $e_{\theta,x,y}$, or simply $e$ when unambiguous.
- For ExEs, we write $z := e_{\theta,x,y_z^*}^{(\text{ExE})}$, where $z \in \mathcal{X}$ is typically a counterfactual input such that $\hat{y}_z = y_z^*$.
- For WBS, the surrogate model is denoted $\theta^e := e_{\theta,x,\hat{y}}^{(\text{WBS})}$. The surrogate returns a label prediction for arbitrary input via $\hat{y}_{x'}^e := \theta^e(x')$.

- $\delta(\cdot, \cdot)$ denotes an arbitrary distance or dissimilarity function, which may be applied to inputs, explanations, or predictions depending on context. Similarity measures (see Subsection C.2.3) may be applied by inverting them appropriately (e.g., via negation or reciprocal).
- $|\cdot|$ indicates size or cardinality; $\|\cdot\|_p$ denotes the $L_p$-norm.
- $k$ refers to a generic parameter (e.g., number of selected features, neighbors, or examples); $\epsilon$ denotes a small threshold or tolerance constant.
- $\Phi_y$ denotes an autoencoder trained specifically on inputs from class $y$ (i.e., $x \in \mathcal{X}_y$).

For convenience, we recall the abbreviations of the explanation types introduced in Subsection 4.1:

- Feature Attribution (FA)
- Concept Explanation (CE)
- Example Explanation (ExE)
- White-Box Surrogate (WBS)
- Natural Language Explanation (NLE)

Additionally, we adopt the following terminology: **Metrics** refer to the individual or aggregated VXAI criteria we evaluate and identified from the literature survey. **Measures** denote performance-scoring functions (e.g., accuracy, similarity, overlap), often also called metrics in literature; but we use the term "measure" to distinguish them clearly.

To improve readability while maintaining completeness throughout the Appendix, we move extensively long reference lists into footnotes using the notation:[a].

---

[a]Author et al.

## C.2   Helper Functions

The following components are recurring elements that serve as functional building blocks across multiple metrics. They can be understood as interchangeable parameters that shape how specific evaluation metrics are instantiated and interpreted.

### C.2.1   Perturbation Approach

Perturbations are small changes typically applied to input features and are recurring components in metrics targeting Fidelity and Continuity. There exists a wide range of perturbation strategies, and the choice of approach can significantly affect both metric results and their interpretation (Brunke et al., 2020; Funke et al., 2022; Rong et al., 2022).

We distinguish first by the **Perturbation Scope**, i.e., the parts of the input that are modified. Perturbations can be applied at a fine-grained level (e.g., individual features) or on more structured, high-level groupings. For image data, scope definitions may involve aggregating pixels into fixed grids (Schulz et al., 2020) or segmenting into superpixels (Ribeiro et al., 2016; Kapishnikov et al., 2019; Rieger & Hansen, 2020). In time-series data, one may perturb fixed-length windows, with the target time-step at the beginning or middle (Schlegel et al., 2019; 2020). In topologically ordered domains (e.g., images, time-series, or graphs), adjacent features can be perturbed together, such as modifying the area surrounding a focal pixel (Samek et al., 2016; Brahimi et al., 2019). Higher-level approaches include perturbing Concepts (El Shawi et al., 2021) or internal activations associated with object parts (Zhang et al., 2019b).

Once the scope is defined, a variety of **Perturbation Functions** can be applied. In fact, any perturbation function might be suitable (Hameed et al., 2022; Schlegel & Keim, 2023). However, here we present some of the most common in literature. At the simplest level, features may be removed by setting them to zero or dropping them entirely, especially in structured domains such as graphs, text, or time-series data, as employed by many authors[b]. Alternatively, features can be replaced by a fixed value, e.g., the per-channel or per-instance mean (Petsiuk, 2018; Schlegel et al., 2019; 2020; Hameed et al., 2022; Jin et al., 2023).

To generate less deterministic perturbations, authors propose adding random noise (e.g., Gaussian) or drawing values from uniform distributions (Yeh et al., 2019; Bhatt et al., 2020; Sturmfels et al., 2020; Bajaj et al., 2021; Funke et al., 2022; Veerappa et al., 2022). Other strategies leverage the spatial structure of the data: for instance, applying blurring or interpolation (Sturmfels et al., 2020; Rong et al., 2022), or reordering spatial regions (Schlegel et al., 2019; Chen et al., 2020). Instead of applying synthetic noise, values can be resampled from the marginal distribution of a feature, from its nearest neighbor, or even from an opposite-class example (Guo et al., 2018b; Hameed et al., 2022). Where influence regions are known, perturbations can be constrained to lie inside or outside these intervals (Velmurugan et al., 2021a).

In the NLP domain, word embeddings can be noised, or tokens substituted using synonym sets and domain knowledge (Yin et al., 2021). For image inputs, another strategy is cropping and resizing to emphasize or suppress local information (Dabkowski & Gal, 2017).

Finally, when perturbations are guided by a FA, their intensity can be scaled proportionally to the assigned importance scores (Chattopadhay et al., 2018; Guo et al., 2018b; Jung & Oh, 2021).

### C.2.2   Normalization of Explanantia

Since FAs and CEs are typically represented as real-valued vectors, computed through various mechanisms, their value ranges are not inherently standardized. However, many metrics either explicitly require the explanans to lie within a fixed range or implicitly assume comparability across explanantia, making normalization a necessary preprocessing step. A widely used normalization method is Min-Max Scaling (Binder et al., 2023; Brandt et al., 2023), which maps all values into a fixed interval (typically $[0, 1]$). Alternative strategies include normalization based on the square root of the average second-moment estimate, offer-

---

[b]Bach et al. (2015); Ancona et al. (2017); Alvarez-Melis & Jaakkola (2018b); Chu et al. (2018); Arya et al. (2019); DeYoung et al. (2019); Schlegel et al. (2019); Cong et al. (2020); Singh et al. (2020); Warnecke et al. (2020); Bajaj et al. (2021); Faber et al. (2021); Singh et al. (2021); Jin et al. (2023)

ing robustness to outliers and variance shifts (Binder et al., 2023). This limited selection can be extended through any suitable normalization approach.

### C.2.3 Similarity Measures

Across various metrics, it is necessary to calculate similarities or distances between two explanantia, especially when concerned with the desideratum of Continuity and in metrics relying on Ground-Truth evaluations. Throughout the reported literature, various approaches have been reported, which differ based on the type of explanation. While some measures directly compute similarity, others quantify distance or disparity. In this work, we adopt a similarity-based framing, either directly or by transforming distance measures, to ensure that higher values uniformly indicate greater explanatory agreement. Analogously, we can compute the similarity between explananda, adopting measures that are presented in the following.

For **FAs**, similarity may be measured using arbitrary inverted distance or loss measures (e.g., $L_p$, MSE, cosine distance, JS-Divergence, or Bhattacharyya Coefficient), potentially normalized (e.g., by standard deviation)[c]. Alternatively, rank correlation measures such as Spearman's or Kendall's Tau can be applied[d]. When binarizing FA outputs through thresholding, feature-wise evaluation measures such as accuracy, precision, $F_1$, or AUROC are commonly used[e]. Similarly, IoU can be calculated over binarized features[f], or top-$k$ intersection measures may be used[g]. For saliency maps, specialized similarity measures are available, such as SSIM[h], Earth Movers Distance (Park et al., 2018; Wu & Mooney, 2018), Normalized Cross Correlation (Baumgartner et al., 2018; Bass et al., 2020), or Mutual Information (Sun et al., 2023). While primarily established for FAs, similar similarity functions can be naturally applied to **CE**-based explanations as well.

For **WBSs** and **ExEs**, the choice of similarity measure depends strongly on the underlying model or domain. For linear predictive models, coefficient mismatch is a common choice (Lakkaraju et al., 2020), whereas rule- and tree-based explanantia may be compared by their rule overlap, feature usage, or node structures[i].

**NLEs** can be compared using standard natural language processing measures[j]. Those include for instance BLEU (Papineni et al., 2002), METEOR (Banerjee & Lavie, 2005), ROUGE (Lin, 2004), CIDEr (Vedantam et al., 2015), or SPICE (Anderson et al., 2016). In addition, several measures have been proposed specifically for evaluating natural language explanations (Xie et al., 2021; Du et al., 2022; Rodis et al., 2024; Park et al., 2018).

When comparing similarities over multiple instances, the most natural aggregation is to compute the mean similarity (Fan et al., 2020; Fouladgar et al., 2022; Yeh et al., 2019). Depending on the evaluation goal, alternative aggregation strategies may offer more informative insights. For example, worst-case stability, defined as the minimum similarity across inputs, can be used to quantify robustness[k].

---

[c]Alvarez-Melis & Jaakkola (2018a;b); Chu et al. (2018); Wu & Mooney (2018); Jain & Wallace (2019); Jia et al. (2019); Mitsuhara et al. (2019); Pope et al. (2019); Trokielewicz et al. (2019); Yeh et al. (2019); Zhang et al. (2019a); Jia et al. (2020); Agarwal et al. (2022a); Atanasova et al. (2022); Dai et al. (2022); Fouladgar et al. (2022); Agarwal et al. (2023); Huang et al. (2023a); Nematzadeh et al. (2023)

[d]Das et al. (2017); Adebayo et al. (2018); Chen et al. (2019b); Dombrowski et al. (2019); Ghorbani et al. (2019); Nguyen & Martínez (2020); Rajapaksha et al. (2020); Sanchez-Lengeling et al. (2020); Liu et al. (2021a); Yin et al. (2021); Krishna et al. (2022); Huang et al. (2023a)

[e]Chen et al. (2018a); Yang et al. (2018b); Jia et al. (2019; 2020); Sanchez-Lengeling et al. (2020); Bykov et al. (2021); Joshi et al. (2021); Park & Wallraven (2021); Amoukou et al. (2022); Chen et al. (2022); Funke et al. (2022); Tjoa & Guan (2022); Wilming et al. (2022); Agarwal et al. (2023)

[f]Oramas et al. (2017); Fan et al. (2020); Kim et al. (2021); Situ et al. (2021); Vermeire et al. (2022)

[g]Ghorbani et al. (2019); Mishra et al. (2020); Rajapaksha et al. (2020); Warnecke et al. (2020); Amparore et al. (2021); Bajaj et al. (2021)

[h]Adebayo et al. (2018); Dombrowski et al. (2019); Rebuffi et al. (2020); Graziani et al. (2021); Sun et al. (2023)

[i]Bastani et al. (2017); Guidotti et al. (2019); Lakkaraju et al. (2020); Rajapaksha et al. (2020); Margot & Luta (2021)

[j]Camburu et al. (2018); Chuang et al. (2018); Liu et al. (2018a); Wu & Mooney (2018); Chen et al. (2019d); Rajani et al. (2019); Wickramanayake et al. (2019); Li et al. (2020a); Sun et al. (2020); Jang & Lukasiewicz (2021); Atanasova (2024b)

[k]Alvarez-Melis & Jaakkola (2018a;b); Yeh et al. (2019); Yin et al. (2021); Fouladgar et al. (2022)

### C.3 Metrics

### C.3.1 Explanans-Centric

The following metrics directly evaluate the explanans, potentially looking into the data inputs but without having any access to the underlying model.

---

**Metric I.1: Explanans Size**

**Desiderata:** Parsimony
**Explanation Type:** FA, ExE, WBS, (CE, NLE)                    **References:** (51)

Craven & Shavlik (1995); Stefanowski & Vanderpooten (2001); Nauck (2003); Alonso et al. (2008); Augasta & Kathirvalavakumar (2012); Lakkaraju et al. (2016); Samek et al. (2016); Zilke et al. (2016); Lakkaraju et al. (2017); Guidotti et al. (2018); Hara & Hayashi (2018); Rustamov & Klosowski (2018); Wang et al. (2018b); Wang (2018); Wang et al. (2018a); Wu et al. (2018a;b); Deng (2019); Evans et al. (2019); Fong et al. (2019); Guidotti et al. (2019); Ignatiev et al. (2019); Lakkaraju et al. (2019); Polato & Aiolli (2019); Pope et al. (2019); Shakerin & Gupta (2019); Slack et al. (2019); Topin & Veloso (2019); Verma & Ganguly (2019); Wang (2019); Yoo & Sael (2019); Bhatt et al. (2020); Chalasani et al. (2020); Molnar et al. (2020); Nguyen & Martínez (2020); Panigutti et al. (2020); Rajapaksha et al. (2020); Rawal & Lakkaraju (2020); Stepin et al. (2020); Warnecke et al. (2020); Wu et al. (2020); Liu et al. (2021b); Margot & Luta (2021); Moradi & Samwald (2021); Poppi et al. (2021); Rosenfeld (2021); Samek et al. (2021); Dai et al. (2022); Funke et al. (2022); Huang et al. (2023b); Stevens & De Smedt (2024)

---

The size of an explanans $|e|$ is a common indicator of its complexity. Smaller or more compact explanantia are generally easier to understand and more plausible to human users. The exact method to measure size depends on the explanation type and context.

A generally applicable method is to compute the file size (in bytes) of the explanans, based on the assumption that sparse explanantia can be more easily compressed (Samek et al., 2016; 2021).

For **WBS**, size is typically measured via structural properties of the surrogate model: tree-based models are assessed by depth, number of nodes, or number of leaves[a], while rule-based systems are evaluated using the number of rules or predicates per rule[b]. For explanation-graphs, the number of nodes and edges can serve as a proxy for size (Rustamov & Klosowski, 2018; Topin & Veloso, 2019). Conversely, the (relative) number of instances covered per rule can also express parsimony (Deng, 2019; Guidotti et al., 2019).

In **FA**, size is typically based on the number of relevant features. This may be computed via:

- The $L_0$ norm[c],
- A count of features exceeding a relevance threshold[d],
- Normalization by input dimensionality, e.g., in graph settings (Pope et al., 2019),
- Integration over multiple thresholds to form a size curve (Warnecke et al., 2020), or
- Threshold-free statistics like entropy (Samek et al., 2016; Bhatt et al., 2020; Funke et al., 2022) and Gini index (Chalasani et al., 2020).

While not common in the literature, for **CE**, similar counting mechanisms may be applied. One may count concepts exceeding a relevance threshold or use the total number of tested concepts (which is not tied to input size).

In **ExE**, size naturally corresponds to the number of examples used in the explanans (Nguyen & Martínez, 2020; Huang et al., 2023b).

For **NLE**, the number of words or sentences provides a straightforward measure of size.

In general, size can be measured per-instance or aggregated over the dataset. For instance, one can report the number of predicates in the rule explaining a single instance (Augasta & Kathirvalavakumar, 2012), or aggregate the number of predicates across a global rule set with statistics such as the average (Lakkaraju et al., 2016; 2017; 2019), sum (Margot & Luta, 2021; Moradi & Samwald, 2021), or maximum (Moradi & Samwald, 2021). Similarly, rule counts can be aggregated over classes (e.g., average number per class (Nauck, 2003)). Optional adjustments include applying a tolerance margin

(e.g., $\max(0, |e| - k)$ (Rosenfeld, 2021)) or aggregating over multiple class-specific explanantia per instance (Pope et al., 2019).

[a]Craven & Shavlik (1995); Alonso et al. (2008); Guidotti et al. (2018); Hara & Hayashi (2018); Wu et al. (2018b); Evans et al. (2019); Slack et al. (2019); Yoo & Sael (2019); Molnar et al. (2020); Rawal & Lakkaraju (2020); Wu et al. (2020)

[b]Craven & Shavlik (1995); Stefanowski & Vanderpooten (2001); Nauck (2003); Alonso et al. (2008); Augasta & Kathirvalavakumar (2012); Lakkaraju et al. (2016); Zilke et al. (2016); Lakkaraju et al. (2017); Wang (2018); Wu et al. (2018a); Deng (2019); Lakkaraju et al. (2019); Polato & Aiolli (2019); Shakerin & Gupta (2019); Wang (2019); Panigutti et al. (2020); Rajapaksha et al. (2020); Stepin et al. (2020); Margot & Luta (2021); Moradi & Samwald (2021); Rosenfeld (2021)

[c]Wang et al. (2018a;b); Fong et al. (2019); Polato & Aiolli (2019); Verma & Ganguly (2019); Nguyen & Martínez (2020); Poppi et al. (2021); Rosenfeld (2021); Stevens & De Smedt (2024)

[d]Ignatiev et al. (2019); Pope et al. (2019); Warnecke et al. (2020); Liu et al. (2021b); Dai et al. (2022)

---

## Metric I.2: Overlap

**Desiderata:** Parsimony
**Explanation Type:** WBS                                    **References:** (5)
Lakkaraju et al. (2016; 2017; 2019); Moradi & Samwald (2021); Hosain et al. (2024)

The overlap within a rule-based explanans (e.g., rule sets or decision trees) can be measured with respect to either the input space or the rules themselves. A lower degree of overlap is generally preferred, as it implies more distinct, non-redundant rules and enhances interpretability.

- **Input Overlap**: Measures how often input instances are covered by multiple rules or decision paths (Lakkaraju et al., 2016; 2017; 2019; Hosain et al., 2024). High overlap may indicate redundant or conflicting logic. Overlap can also be broken down by class label to distinguish intra-class from inter-class coverage.
- **Rule Overlap**: Measures how many rules share identical or highly similar predicates (Moradi & Samwald, 2021), indicating structural redundancy within the rule set.

---

## Metric I.3: Explanans Cohesion

**Desiderata:** Parsimony, Plausibility
**Explanation Type:** FA,(CE)                                **References:** (2)
Fong & Vedaldi (2017); Saifullah et al. (2024)

In domains where input features are ordered (e.g., images or time series, as opposed to tabular data), we expect relevant information to be located in a smooth and coherent region of the input. This property can be evaluated in two complementary ways.

First, Fong & Vedaldi (2017) assess the locality of the explanans: a higher degree of cohesion is achieved when relevant information lies within a small, contiguous region of interest, making the explanans more condensed and easier to interpret. In saliency maps, this can be quantified by computing the area of the smallest bounding box that encloses the thresholded explanans.

Second, Saifullah et al. (2024) assess the smoothness of the explanans: a more continuous attribution map is often easier to understand. This can be measured by summing the absolute differences in attribution between neighboring features (e.g., in x- and y-directions for images). A higher score indicates a more fragmented, less interpretable explanans.

Both variants are applicable to data with ordered features and extend naturally to temporal domains. While originally proposed for FAs, these metrics can also be applied to CE when concept-based saliency maps are available (e.g., as presented by Lucieri et al. (2020)).

---

**Metric I.4: Minimality**

---

**Desiderata:** Parsimony, Plausibility
**Explanation Type:** ExE                                                              **References:** (25)
Tolomei et al. (2017); Wachter et al. (2017); Karlsson et al. (2018); Zhang et al. (2018c); Guidotti et al. (2019); Albini et al. (2020); Artelt & Hammer (2020); Dandl et al. (2020); Kanamori et al. (2020); Karimi et al. (2020); Le et al. (2020); Mothilal et al. (2020); Pawelczyk et al. (2020); Ramon et al. (2020); Sharma et al. (2020); Abrate & Bonchi (2021); Hvilshøj et al. (2021); Pawelczyk et al. (2021); Rasouli & Yu (2021); Van Looveren & Klaise (2021); Albini et al. (2022); Chou et al. (2022); Bayrak & Bach (2023a); Huang et al. (2023b); Verma et al. (2024)

---

To ensure that counterfactuals are understandable and believable, the induced changes should be minimal. This can be assessed by evaluating both the proximity of the counterfactual to the original instance and the sparsity of the changes.

**Proximity** is commonly measured as the distance $\delta(x, z)$ between the counterfactual $z$ and the original instance $x$. A variety of distance measures are used, with the choice significantly impacting the results (Wachter et al., 2017; Artelt & Hammer, 2020; Bayrak & Bach, 2023a; Verma et al., 2024). Common options include (feature-wise weighted) $L_p$ distances (especially $L_2$)[a], as well as Jaccard or cosine distance (Tolomei et al., 2017), or combinations thereof (Karimi et al., 2020; Hvilshøj et al., 2021). Other notable choices are Mahalanobis distance (Artelt & Hammer, 2020; Kanamori et al., 2020; Verma et al., 2024), Gower distance (Dandl et al., 2020; Karimi et al., 2020), or feature-wise cumulative density-based distances (Pawelczyk et al., 2020). Dataset-specific alternatives include computing the quantile-shift per feature (Albini et al., 2022). Different data types require suitable distance measures, e.g.: for graphs, the symmetric difference of adjacency matrices or cosine similarity between node features (Abrate & Bonchi, 2021); for images, the inverse SSIM score (Sharma et al., 2020); and for time series, distances measured per time step (Karlsson et al., 2018).

**Sparsity**, in contrast, concerns the number of features changed rather than the extent of change. It is often quantified via the $L_0$ norm, that is, the number (or fraction) of altered features[b].

Bridging sparsity and proximity, measures like $L_1$ (Wachter et al., 2017) or elastic-net regularization (Van Looveren & Klaise, 2021) are sometimes used. Depending on the representation, alternative definitions may apply, for example, counting the number of changed rules in rule-based counterfactuals (Guidotti et al., 2019).

---

[a]Wachter et al. (2017); Artelt & Hammer (2020); Le et al. (2020); Mothilal et al. (2020); Sharma et al. (2020); Rasouli & Yu (2021); Chou et al. (2022); Verma et al. (2024)
[b]Albini et al. (2020); Dandl et al. (2020); Le et al. (2020); Mothilal et al. (2020); Ramon et al. (2020); Pawelczyk et al. (2021); Bayrak & Bach (2023a); Verma et al. (2024)

---

**Metric I.5: Autoencoder Plausibility**

---

**Desiderata:** Plausibility
**Explanation Type:** ExE                                                              **References:** (1)
Van Looveren & Klaise (2021)

---

Autoencoders are trained to capture the underlying structure of the dataset. Based on this idea, Van Looveren & Klaise (2021) propose evaluating the plausibility of counterfactual examples via reconstruction errors from class-specific and general-purpose autoencoders.
Two specific scores are introduced:

- **IM1** (Class-specific reconstruction comparison): This metric compares how well a counterfactual $z$ is reconstructed by an autoencoder trained on the target class ($\Phi_{y_z^*}$) versus the originally predicted class ($\Phi_{\hat{y}_x}$): $\frac{\|z - \Phi_{y_z^*}(z)\|_2^2}{\|z - \Phi_{\hat{y}_x}(z)\|_2^2 + \epsilon}$.
  A low IM1 score implies that the counterfactual is more representative of the target class than of its original class, indicating class-specific plausibility.

- **IM2** (General manifold plausibility): This score compares the reconstructions from a general autoencoder $\Phi_{\mathcal{Y}}$ and a class-specific one: $\frac{\|\Phi_{\mathcal{Y}}(z) - \Phi_{y_z^*}(z)\|_2^2}{\|z\|_1 + \epsilon}$.
  A low IM2 score implies that the counterfactual aligns well with both the general data manifold and the target class distribution, thus indicating higher plausibility.

While these metrics provide valuable insight into how realistic or semantically valid counterfactuals are, they come with caveats. Training multiple autoencoders (e.g., per class) introduces significant computational overhead. Further, reconstruction quality may vary across classes, and Hvilshøj et al. (2021) showed that autoencoders can be sensitive to small perturbations, potentially undermining the consistency of the plausibility assessment.

---

**Metric I.6: Diversity**

**Desiderata:** Plausibility
**Explanation Type:** ExE                                                    **References:** (6)
Wachter et al. (2017); Karimi et al. (2020); Mothilal et al. (2020); Nguyen & Martínez (2020); Stepin et al. (2020); Verma et al. (2024)

In cases where multiple counterfactuals are generated for a single instance, they should be diverse to offer distinct and meaningful alternatives (Wachter et al., 2017; Stepin et al., 2020; Verma et al., 2024).
Diversity is typically quantified by computing pairwise distances among the set of counterfactuals, either as the average pairwise distance (Mothilal et al., 2020; Nguyen & Martínez, 2020), or as the number of counterfactual pairs that exceed a predefined distance threshold (Karimi et al., 2020).

---

**Metric I.7: Input Similarity**

**Desiderata:** Plausibility
**Explanation Type:** ExE                                                    **References:** (11)
Laugel et al. (2019a;b); Mahajan et al. (2019); Singla et al. (2019); Artelt & Hammer (2020); Dandl et al. (2020); Kanamori et al. (2020); Delaney et al. (2021); Pawelczyk et al. (2021); Rasouli & Yu (2021); Smyth & Keane (2022)

To ensure that counterfactuals remain realistic and trustworthy, they should lie close to the true data manifold. Counterfactuals that deviate significantly from the distribution of training data are unlikely to be plausible. Two principal approaches are used to evaluate this alignment: direct estimation of data conformity and distance-based proximity to the training data.
In the **direct approach**, several statistical or unsupervised methods determine whether the counterfactual is an outlier. This includes calculating its likelihood under a kernel density estimator (Artelt & Hammer, 2020) or from known data distributions in synthetic setups (Mahajan et al., 2019). Other methods include Local Outlier Factor (Kanamori et al., 2020; Delaney et al., 2021) and Isolation Forests (Delaney et al., 2021), which can be applied either across the full dataset or restricted to samples of the counterfactual's target class (Artelt & Hammer, 2020). Pawelczyk et al. (2021) assess how many of a counterfactual's nearest neighbors share its class label.
Alternatively, proximity can be evaluated through **distance-based** approaches, relying on any suitable distance measure (or inverted similarity function, see Subsection C.2.3). These include distances to the $k$ nearest neighbors in the training data (Laugel et al., 2019a; Dandl et al., 2020), or to the nearest training instances that share the same class label as the counterfactual or the original instance (Rasouli & Yu, 2021; Smyth & Keane, 2022). Specific domains may require tailored metrics such as the Fréchet Inception Distance (FID) for images (Singla et al., 2019).
Raw distance scores are often used directly (Dandl et al., 2020), but some methods normalize distances, for example, by comparing the counterfactual–input distance $\delta(x, z)$ to average distances

between random pairs $\delta(x', x'')$ in the dataset (Laugel et al., 2019a;b), or to the original instance–counterfactual distance (Smyth & Keane, 2022).

---

**Metric I.8: Input Contrastivity**

**Desiderata:** Plausibility, (Fidelity)
**Explanation Type:** FA, (ExE, CE, WBS, NLE)          **References:** (2)
Honegger (2018); Pornprasit et al. (2021)

Explanantia should be specific to their corresponding explananda and not overly generic. That is, distinct inputs should typically yield distinct explanantia.
To assess this, we calculate the fraction of instances or instance pairs that result in different explanantia (Honegger, 2018; Pornprasit et al., 2021). A higher fraction of distinguishable explanantia indicates greater contrastivity and, by extension, higher plausibility.
While this metric has been primarily proposed for FAs, it may be applicable to other explanation types as well.

---

**Metric I.9: Actionability**

**Desiderata:** Plausibility, Fidelity
**Explanation Type:** ExE                   **References:** (7)
Mahajan et al. (2019); Karimi et al. (2020); Le et al. (2020); Pawelczyk et al. (2021); Ma et al. (2022); Smyth & Keane (2022); Verma et al. (2024)

For a counterfactual to be helpful, it should only introduce changes that are feasible or allowed. This is often enforced by defining explicit constraints, such as:

- Immutable features (e.g., sex, age) (Karimi et al., 2020; Pawelczyk et al., 2021; Smyth & Keane, 2022; Verma et al., 2024), or
- Value boundaries for individual features (Karimi et al., 2020; Le et al., 2020; Ma et al., 2022; Smyth & Keane, 2022).

The actionability of a counterfactual set can then be assessed by calculating the fraction of counterfactuals that satisfy all constraints[a], or by computing the harmonic mean of satisfied constraints per counterfactual (Mahajan et al., 2019).

---
[a]Karimi et al. (2020); Le et al. (2020); Pawelczyk et al. (2021); Ma et al. (2022); Smyth & Keane (2022)

---

**Metric I.10: Model-Agnostic Explanation Consistency**

**Desiderata:** Plausibility, (Consitency)
**Explanation Type:** FA, ExE, (CE, WBS, NLE)        **References:** (4)
Fan et al. (2020); Nguyen et al. (2020); Hvilshøj et al. (2021); Jiang et al. (2023)

To assess whether explanations reflect generalizable patterns rather than model-specific artifacts (such as adversarial shortcuts in counterfactual explanations), several authors evaluate explanations across different models trained on the same dataset.
In general, the approach involves training additional black-box models (oracles) on the same data and task. These oracles are then used to evaluate explanations from the original model.
One strategy is to directly compute the similarity between explanantia generated by different models for the same input (Fan et al., 2020). While this has been reported for FAs, it is applicable to any explanation type, provided a suitable similarity metric is chosen (see Subsection C.2.3).
Alternatively, explanations from the original model are evaluated using the oracles:

- Nguyen et al. (2020) use perturbation-based evaluation (e.g., Metric 28) on a secondary model to assess whether the explanation highlights features that are generally important across models.
- Hvilshøj et al. (2021) propose that counterfactuals should change the prediction in both the original model and the oracle. Only such counterfactuals are considered plausible.
- An extension of this approach trains one oracle per class and computes the Jensen-Shannon-Divergence between its predictions on the original and counterfactual input. Ideally, only the target and original class should exhibit strong divergence (Hvilshøj et al., 2021).
- Jiang et al. (2023) define a neighborhood of models via small weight perturbations and count how many counterfactuals remain valid across all neighbors.

This approach might be extended to other explanation types. However, its interpretive strength remains limited: it is unclear whether explanations should be similar across models, as differing model architectures may learn distinct (yet valid) rationales. As a result, high or low agreement does not always reflect explanation quality, making this metric inherently context-dependent.

---

**Metric I.11: Input Coverage**

**Desiderata:** Coverage
**Explanation Type:** FA, ExE, WBS, (CE,NLE                                    **References:** (8)
Lakkaraju et al. (2016; 2017; 2019); Ribeiro et al. (2018); Rawal & Lakkaraju (2020); Warnecke et al. (2020); Moradi & Samwald (2021); Huang et al. (2023b)

An explanation method should ideally be capable of generating a valid explanans for every input instance, regardless of its position on the data manifold. To assess this, we calculate the fraction of inputs for which the method provides a valid result.
The notion of "valid result" is intentionally broad and depends on the explanation type and the use case. A few prominent definitions include:

- For perturbation-based FAs like LIME (see Ribeiro et al. (2016)), a valid explanans can be given if a sufficient number of perturbed samples belong to the target or opposite class, enabling a reliable local approximation (Warnecke et al., 2020).
- For rule-based or tree-based WBSs, coverage is the number of instances that are captured by at least one rule or decision path (Lakkaraju et al., 2016; 2017; 2019; Ribeiro et al., 2018; Rawal & Lakkaraju, 2020), optionally restricted to rules associated with the correct class (Moradi & Samwald, 2021).
- For global ExEs, coverage may be defined as the number of inputs that have at least one sufficiently similar explaining instance within a pre-defined distance (Huang et al., 2023b).

---

**Metric I.12: Output Coverage**

**Desiderata:** Coverage
**Explanation Type:** WBS                                                      **References:** (1)
Lakkaraju et al. (2016)

Inherently interpretable WBS models should be capable of producing predictions for all possible output classes. Lakkaraju et al. (2016) evaluate the model's class-level coverage by calculating the fraction of target classes that appear in the explanans. This could, for instance, be the number of classes that are assigned to at least one rule in a rule set or to a leaf in a decision tree.

---

**Metric I.13: Output Mutual Information**

**Desiderata:** Fidelity
**Explanation Type:** FA,CE **References:** (1)
Nguyen & Martínez (2020)

When high-level features (such as concepts or aggregated input features) are provided by an explanans, they should capture as much information as possible about the model's output. In turn, the prediction should be reflected in the explanans, implying a high degree of mutual dependency between both.
To quantify this relationship, Nguyen & Martínez (2020) leverage the *Mutual Information (MI)* (see Cover (1999)) between the explanans and the output: $\mathbf{MI}(e_{\theta,x,\hat{y}}, \theta(x))$. Since MI is symmetric, a high score indicates that the explanation both reflects (i.e. correctness) and encompasses (i.e., completeness) the relevant reasoning of the model.

---

### C.3.II   Model Observation

This set of metrics requires access to the model to obtain information about the model's output or its internal activations.

---

**Metric II.14: Input Mutual Information**

**Desiderata:** Parsimony
**Explanation Type:** FA,CE **References:** (1)
Nguyen & Martínez (2020)

When high-level features are provided by an explanans, they should ideally represent an abstracted, human-understandable form of the input, omitting unnecessary detail. This promotes parsimony by simplifying the input space. Such features can be obtained either through feature selection in FAs or through concept-level representations in CEs.
To quantify this abstraction, Nguyen & Martínez (2020) propose measuring the *Mutual Information (MI)* (see (Cover, 1999)) between the input and the explanans: $\mathbf{MI}(x, e_{\theta,x,\hat{y}})$ A lower mutual information score suggests that the explanans captures less granular input detail and thus constitutes a more compact and parsimonious representation.

---

**Metric II.15: Output Contrastivity**

**Desiderata:** Plausibility
**Explanation Type:** FA, (ExE, CE, WBS, NLE) **References:** (5)
Nie et al. (2018); Pope et al. (2019); Li et al. (2020c); Rebuffi et al. (2020); Sixt et al. (2020)

To enhance plausibility, explanations should be class-discriminative, that is, they should differ depending on the target class. In image classification, for instance, the explanantia supporting a "car" label should highlight different regions than those supporting "dog". This makes it easier for humans to understand the specific rationale for each class.
Class discriminativeness is commonly measured by comparing explanantia generated for different classes on the same input. Several comparison setups have been proposed: between the most and least likely classes (Li et al., 2020c; Rebuffi et al., 2020), the top predicted vs. a randomly chosen class (Sixt et al., 2020), or simply between the two classes in binary settings (Pope et al., 2019).
Any suitable similarity or distance measure can be used to assess the degree of overlap between explanantia, such as the $L_0$ or $L_2$ distance (Nie et al., 2018; Pope et al., 2019), rank correlation (Rebuffi et al., 2020), or SSIM for image domain (Sixt et al., 2020). A lower similarity implies a clearer distinction between the rationales for each class.

Although the reported implementations target FA, the idea of class-discriminative explanations can be naturally extended to other type, wherever explanations can be generated for multiple class hypotheses.

---

**Metric II.16: Output Similarity**

**Desiderata:** Plausibility
**Explanation Type:** ExE                                    **References:** (1)
Plumb et al. (2020)

To ensure plausibility, counterfactuals should yield output distribution similar to real instances of the target class. Plumb et al. (2020) evaluate whether each counterfactual $z$ matches the output activation of at least one training sample, i.e.: $\exists x' \in \mathcal{X}_{y_z^*} \; : \; \delta\big(\theta(x'), \theta(z)\big) < \epsilon$
This confirms that the counterfactual aligns with typical model behavior for that class.

---

**Metric II.17: Mutual Coherence**

**Desiderata:** Plausibility, (Fidelity)
**Explanation Type:** FA, (ExE, CE, WBS, NLE)                **References:** (19)
Selvaraju et al. (2017); Guo et al. (2018a); Ancona et al. (2019); Fernando et al. (2019); Fusco et al. (2019); Jain & Wallace (2019); Zhang et al. (2019a); Marques-Silva et al. (2020); Nguyen & Martínez (2020); Wang et al. (2020b); Warnecke et al. (2020); Graziani et al. (2021); Malik et al. (2021); Rajbahadur et al. (2021); Krishna et al. (2022); Mercier et al. (2022); Jin et al. (2023); Duan et al. (2024); Tekkesinoglu & Pudas (2024)

Several authors propose to evaluate their explanations by comparing them against other XAI methods that are assumed to produce trustworthy or well-understood outputs. Although only reported for FAs, this approach is applicable to any explanation type, as long as a suitable reference XAI method is available and an appropriate similarity metric can be defined.
A common reference is the Shapley value framework (see Shapley (1953)), often operationalized via SHAP (see Lundberg (2017)) due to its solid theoretical grounding. FAs are frequently compared to SHAP estimates using similarity measures[a]. In the context of saliency maps, Image Occlusion (see Zeiler & Fergus (2013)) is similarly treated as a proxy ground truth (Selvaraju et al., 2017). Comparisons against further explanation methods are also reported in literature[b].
Other works compute similarity across multiple explanation methods, assuming that high agreement between methods reflects convergence toward a reliable explanans[c]. Any suitable similarity measure may be applied (see Subsection C.2.3 for an overview).
However, this evaluation approach is not without criticism. Kumar et al. (2020) caution that validating one explanation method using another may propagate shared biases or assumptions, providing limited evidence for actual correctness.

---

[a]Ancona et al. (2019); Zhang et al. (2019a); Malik et al. (2021); Jin et al. (2023); Tekkesinoglu & Pudas (2024)
[b]Guo et al. (2018a); Fernando et al. (2019); Fusco et al. (2019); Marques-Silva et al. (2020); Nguyen & Martínez (2020)
[c]Jain & Wallace (2019); Wang et al. (2020b); Warnecke et al. (2020); Graziani et al. (2021); Rajbahadur et al. (2021); Krishna et al. (2022); Mercier et al. (2022)

---

**Metric II.18: Significance Check**

**Desiderata:** Fidelity
**Explanation Type:** FA, CE, (ExE, WBS, NLE)                **References:** (15)
Samek et al. (2016); Adel et al. (2018); Chen et al. (2018a); Kim et al. (2018); Chen et al. (2019a); DeYoung et al. (2019); Gu et al. (2019); Wickramanayake et al. (2019); Nam et al. (2020); Hemamou et al. (2021); Park & Wallraven (2021); Pornprasit et al. (2021); Agarwal et al. (2022c); Hameed et al. (2022); Bommer et al. (2024)

Statistical significance testing is a common strategy for verifying whether an explanation is meaningfully different from random or naïve baseline explanations. This approach is frequently applied to FAs[a].

For CEs, statistical tests are typically used to assess whether the extracted concepts carry significantly more information than randomly sampled concepts. This is done by comparing average relevance or activation scores between the true and random concepts (Adel et al., 2018; Kim et al., 2018).

In a more advanced variant, Hemamou et al. (2021) train a classifier to distinguish between real and synthetic (random) explanations. High accuracy in this task indicates that the generated explanantia contain statistically meaningful structure.

While commonly reported for FA and CE, the general idea of statistical significance testing could, in principle, be adapted to other explanation types as well. However, doing so may require explanation-specific reformulations.

---

[a]Samek et al. (2016); Chen et al. (2018a; 2019a); DeYoung et al. (2019); Gu et al. (2019); Wickramanayake et al. (2019); Nam et al. (2020); Hemamou et al. (2021); Park & Wallraven (2021); Pornprasit et al. (2021); Agarwal et al. (2022c); Hameed et al. (2022); Bommer et al. (2024)

## Metric II.19: (Counter-)Factual Relevance

**Desiderata:** Fidelity
**Explanation Type:** ExE                                    **References:** (1)
Liu et al. (2021b)

This metric applies to explanations that generate both a factual and counterfactual explanans, aiming to evaluate how well they reflect and contrast the model's reasoning.

Originally proposed by Liu et al. (2021b) for the graph domain, requiring both explanantia being subgraphs, it can be generalized to other input types.

The method computes the model's output for the factual and counterfactual explanantia and compares them to the original prediction using the negative symmetric Kullback-Leibler divergence. The difference between these two scores is normalized by the distance between the factual and counterfactual explanantia. A high normalized score indicates that both explanans are informative: the factual preserves the original reasoning, while the counterfactual shifts the decision appropriately.

## Metric II.20: Prediction Validity

**Desiderata:** Fidelity
**Explanation Type:** ExE                                    **References:** (18)
Wachter et al. (2017); Karlsson et al. (2018); Dhurandhar et al. (2019); Guidotti et al. (2019); Mahajan et al. (2019); Dandl et al. (2020); Le et al. (2020); Molnar (2020); Mothilal et al. (2020); Nguyen & Martínez (2020); Pedapati et al. (2020); Pawelczyk et al. (2021); Rasouli & Yu (2021); Ma et al. (2022); Tan et al. (2022); Vermeire et al. (2022); Guidotti (2024); Verma et al. (2024)

By definition, a counterfactual must result in a different prediction from the original input. For untargeted counterfactuals, this simply requires $\hat{y}_z \neq \hat{y}_x$, while targeted counterfactuals must satisfy $\hat{y}_z = y_z^*$ (Wachter et al., 2017; Molnar, 2020; Guidotti, 2024).

Most authors assess the fraction of generated counterfactuals that meet this condition[a]. To capture more nuance, authors also propose measuring the model's confidence in the target class for the counterfactual (Rasouli & Yu, 2021), or using a continuous loss between the predicted and target class probabilities, such as the $L_1$ distance (Dandl et al., 2020).

For factual explanations, the class should remain unchanged, i.e., $\hat{y}_z = \hat{y}_x$, which can also be verified either binary (Dhurandhar et al., 2019) or using loss-based similarity measures (Nguyen & Martínez, 2020).

[a]Karlsson et al. (2018); Dhurandhar et al. (2019); Guidotti et al. (2019); Mahajan et al. (2019); Le et al. (2020); Mothilal et al. (2020); Pedapati et al. (2020); Pawelczyk et al. (2021); Ma et al. (2022); Tan et al. (2022); Vermeire et al. (2022); Verma et al. (2024)

## Metric II.21: Sufficiency

**Desiderata:** Fidelity
**Explanation Type:** CE                                                    **References:** (2)
Yeh et al. (2020); Dasgupta et al. (2022)

An explanation should be complete enough to serve as a sufficient reason for a given model output. Ideally, the explanans alone should allow us to predict the outcome. This property reflects the completeness of the explanans and can be assessed in different ways.

Yeh et al. (2020) train a secondary model that maps the explanans back to the black-box's activation space. The predictive performance using the mapped explanans indicates how much information the explanation preserves about the original model's decision process.

Alternatively, Dasgupta et al. (2022) evaluate whether explanations are sufficient for consistent outcomes: Given an explanans $e_0$ for an input $x_0$, we identify other instances whose explanantia are equivalent (or sufficiently similar) to $e_0$ and compute the fraction that shares the same model prediction as $x_0$. A higher agreement indicates a more sufficient explanation.

## Metric II.22: Output Faithfulness

**Desiderata:** Fidelity
**Explanation Type:** WBS                                                    **References:** (34)
Andrews et al. (1995); Craven & Shavlik (1995); Stefanowski & Vanderpooten (2001); Barakat et al. (2010); Augasta & Kathirvalavakumar (2012); Zilke et al. (2016); Bastani et al. (2017); Krishnan & Wu (2017); Lakkaraju et al. (2017); Guo et al. (2018b); Laugel et al. (2018); Peake & Wang (2018); Plumb et al. (2018); Tan et al. (2018); Wu et al. (2018a); Zhang et al. (2018b); Chen et al. (2019a); Guidotti et al. (2019); Kanehira & Harada (2019); Lakkaraju et al. (2019); Zhou et al. (2019); Anders et al. (2020); Hatwell et al. (2020); Lakkaraju et al. (2020); Panigutti et al. (2020); Pedapati et al. (2020); Rajapaksha et al. (2020); Rawal & Lakkaraju (2020); Amparore et al. (2021); Chen et al. (2021); Moradi & Samwald (2021); Pornprasit et al. (2021); Bayrak & Bach (2023b); Bo et al. (2024)

A surrogate model should closely mimic the behavior of the original black-box model. Therefore, a common evaluation strategy is to compare the outputs of the surrogate and black-box models using a similarity or performance measure.

This can be assessed with standard measures such as accuracy, $F_1$-score, MSE, or $L_p$ distances[a], or with similarity measures such as SSIM, Pearson correlation, or KL divergence (Chen et al., 2019a; Anders et al., 2020). Arbitrary loss functions may also be used (Amparore et al., 2021).

For imbalanced datasets, Moradi & Samwald (2021) calculate fidelity per class, using either the true labels or the black-box predictions as reference. For individual rules, fidelity may be expressed as the rule's precision (Stefanowski & Vanderpooten, 2001; Hatwell et al., 2020).

When evaluating local surrogates, *Local Output Fidelity* is defined by computing fidelity within a neighborhood of the input instance (Laugel et al., 2018; Plumb et al., 2018; Guidotti et al., 2019; Rajapaksha et al., 2020), or within a synthetic neighborhood (Panigutti et al., 2020; Pornprasit et al., 2021). For unseen data, Lakkaraju et al. (2020) validate explanations by comparing outputs to those of the nearest training instances.

[a]Andrews et al. (1995); Craven & Shavlik (1995); Barakat et al. (2010); Augasta & Kathirvalavakumar (2012); Zilke et al. (2016); Bastani et al. (2017); Krishnan & Wu (2017); Lakkaraju et al. (2017); Guo et al. (2018b); Laugel et al. (2018); Plumb et al. (2018); Tan et al. (2018); Zhang et al. (2018b); Guidotti et al. (2019); Kanehira & Harada (2019); Lakkaraju et al. (2019); Zhou et al. (2019); Anders et al. (2020); Lakkaraju et al. (2020); Panigutti et al. (2020); Pornprasit et al. (2021); Bayrak & Bach (2023b)

**Metric II.23: Internal Faithfulness**

**Desiderata:** WBS
**Explanation Type:** Fidelity **References:** (3)
Messalas et al. (2019); Anders et al. (2020); Amparore et al. (2021)

Since the WBS models can achieve similar predictive performance as the original black-box without relying on the same underlying reasoning (Messalas et al., 2019; Anders et al., 2020), it is essential to evaluate their internal fidelity, meaning the similarity in how both models justify their predictions. This can be achieved by comparing post-hoc explanations of the original and surrogate model for the same inputs. Using feature attribution methods (e.g., SHAP from Lundberg (2017)), a typical approach is to measure the average overlap of the top-$k$ features between both models' explanantia (Messalas et al., 2019). Other similarity metrics and explanation types may also be used.
Alternatively, Amparore et al. (2021) compare counterfactuals generated from each model, treating their similarity as a proxy for the alignment of decision boundaries. This provides a structural view of how well the surrogate captures the black-box model's rationale beyond mere output agreement.

**Metric II.24: Setup Consistency**

**Desiderata:** Consistency
**Explanation Type:** FA, ExE, WBS, (CE, NLE) **References:** (11)
Bastani et al. (2017); Honegger (2018); Guidotti et al. (2019); Rajapaksha et al. (2020); Warnecke et al. (2020); Amparore et al. (2021); Graziani et al. (2021); Margot & Luta (2021); Velmurugan et al. (2021b); Dai et al. (2022); Vermeire et al. (2022)

This metric evaluates how consistent the explanantia remain when generated multiple times for the same input and model. This, however, is only a necessary consideration for nondeterministic explanation methods.
Some authors assess this axiomatically, by counting the fraction of identical explanantia produced across repeated runs (Honegger, 2018; Vermeire et al., 2022). Others calculate distances or similarity scores between different runs and report the aggregated variation, using general or task-specific metrics such as the variance of feature weights or feature presence[a].
Although most work focuses on local instance-wise explanations, the same principle applies to global explanations, where the similarity between globally constructed explanantia is measured (Bastani et al., 2017; Margot & Luta, 2021).
While we did not identify examples of this metric applied to every explanations type in the literature, the core idea is general and can easily be extended to any XAI algorithm.

---

[a]Guidotti et al. (2019); Rajapaksha et al. (2020); Warnecke et al. (2020); Amparore et al. (2021); Graziani et al. (2021); Velmurugan et al. (2021b); Dai et al. (2022)

**Metric II.25: Hyperparameter Sensitivity**

**Desiderata:** Consistency
**Explanation Type:** FA, (ExE, CE, WBS, NLE) **References:** (6)
Chen et al. (2019a); Verma & Ganguly (2019); Bansal et al. (2020); Mishra et al. (2020); Sanchez-Lengeling et al. (2020); Graziani et al. (2021)

This metric evaluates how robust an explanation method is to changes in its configuration. Since many XAI methods rely on hyperparameters, high sensitivity may indicate instability, making tuning more difficult and reducing user trust in the method (Bansal et al., 2020).
A common approach is to generate explanantia for the same input across different hyperparameter settings and measure the similarity between them[a]. Alternatively, Chen et al. (2019a) propose to compare the stability of performance metrics such as fidelity over varying hyperparameters. Beyond

identifying general sensitivity, Graziani et al. (2021) use this technique to guide hyperparameter selection: by progressively adjusting hyperparameters and tracking when the generated explanantia converge, one can identify stable regions in the hyperparameter space.

Although most work focuses on FAs, the general idea is applicable to any method with tunable hyperparameters.

---

[a]Verma & Ganguly (2019); Bansal et al. (2020); Mishra et al. (2020); Sanchez-Lengeling et al. (2020); Graziani et al. (2021)

---

**Metric II.26: Execution Time**

**Desiderata:** Efficiency
**Explanation Type:** FA, ExE, WBS, (CE, NLE)                          **References:** (34)
Ross et al. (2017); Ribeiro et al. (2018); Zhang et al. (2018c); Chen et al. (2019a); Cheng et al. (2019); Fusco et al. (2019); Ignatiev et al. (2019); Shakerin & Gupta (2019); Slack et al. (2019); Topin & Veloso (2019); Albini et al. (2020); Guo et al. (2020); Marques-Silva et al. (2020); Rajapaksha et al. (2020); Ramon et al. (2020); Warnecke et al. (2020); Abrate & Bonchi (2021); Bajaj et al. (2021); Faber et al. (2021); Lin et al. (2021); Malik et al. (2021); Pawelczyk et al. (2021); Rasouli & Yu (2021); Van Looveren & Klaise (2021); Wang et al. (2021); Amoukou et al. (2022); Belaid et al. (2022); Ma et al. (2022); Mercier et al. (2022); Vermeire et al. (2022); Bayrak & Bach (2023a); Brandt et al. (2023); Jin et al. (2023); Verma et al. (2024)

---

Most authors measure the time required to generate an explanans for a fixed input, model, and dataset, on specified hardware[a].

In addition to empirical runtime, some authors analyze algorithmic complexity using $\mathcal{O}$-notation to characterize the worst-case or average-case number of steps required to run an explanation[b]. Chen et al. (2019a) further evaluate parallelizability to understand potential runtime improvements through hardware acceleration or distributed computation.

---

[a]Ross et al. (2017); Ribeiro et al. (2018); Zhang et al. (2018c); Cheng et al. (2019); Fusco et al. (2019); Ignatiev et al. (2019); Shakerin & Gupta (2019); Guo et al. (2020); Marques-Silva et al. (2020); Rajapaksha et al. (2020); Ramon et al. (2020); Warnecke et al. (2020); Bajaj et al. (2021); Faber et al. (2021); Lin et al. (2021); Malik et al. (2021); Pawelczyk et al. (2021); Rasouli & Yu (2021); Van Looveren & Klaise (2021); Wang et al. (2021); Amoukou et al. (2022); Belaid et al. (2022); Ma et al. (2022); Mercier et al. (2022); Vermeire et al. (2022); Bayrak & Bach (2023a); Brandt et al. (2023); Jin et al. (2023); Verma et al. (2024)
[b]Slack et al. (2019); Topin & Veloso (2019); Albini et al. (2020); Abrate & Bonchi (2021); Malik et al. (2021); Van Looveren & Klaise (2021)

---

### C.3.III  Input Intervention

The metrics listed in this group all require changes to the input, which is then presented to the model again and potentially explained once more.

---

**Metric III.27: Unguided Perturbation Fidelity**

**Desiderata:** Fidelity
**Explanation Type:** FA                                               **References:** (10)
Ancona et al. (2017); Shrikumar et al. (2017); Sundararajan et al. (2017); Alvarez-Melis & Jaakkola (2018b); Arya et al. (2019); Cheng et al. (2019); Yeh et al. (2019); Zhang et al. (2019b); Bhatt et al. (2020); Elkhawaga et al. (2023)

---

Some of the most influential XAI metrics evaluate how well an explanans aligns with observed model behavior when the input is perturbed. Together, these metrics assess whether the explanation faithfully captures how the model responds to its inputs.

This includes *Sensitivity-n* from Ancona et al. (2017), *Infidelity* from Yeh et al. (2019), and *Faithfulness* presented by Alvarez-Melis & Jaakkola (2018b) and Arya et al. (2019). In these approaches, input perturbations are introduced either randomly across all features (Yeh et al., 2019), by zeroing

features individually or in small groups (Alvarez-Melis & Jaakkola, 2018b; Arya et al., 2019; Cheng et al., 2019; Bhatt et al., 2020), or by manipulating subsets of fixed size (Ancona et al., 2017). The model's change in prediction is then compared to the explanans, using different strategies: directly against the raw attribution vector, against a version scaled by perturbation magnitude, or by summing attributions of the changed features. The deviation is measured using standard metrics like Pearson correlation[a] or mean squared error (Yeh et al., 2019).

Complementing these are completeness-based approaches such as *Completeness* (Sundararajan et al., 2017) and *Summation-to-Delta* (Shrikumar et al., 2017), which assume that the explanans must fully account for the model's behavior. These compare the difference in output between an instance and a baseline (e.g., a fully perturbed input) with the sum of the attributions across changed features. Ideally, the two should match exactly. For practical purposes, however, the relative deviation between attribution sum and output delta can be used as a softer criterion.

---

[a]Ancona et al. (2017); Alvarez-Melis & Jaakkola (2018b); Arya et al. (2019); Cheng et al. (2019); Bhatt et al. (2020)

## Metric III.28: Guided Perturbation Fidelity

**Desiderata:** Fidelity
**Explanation Type:** FA,CE                                                     **References:** (75)

Bach et al. (2015); Samek et al. (2016); Ancona et al. (2017); Dabkowski & Gal (2017); Shrikumar et al. (2017); Chattopadhay et al. (2018); Chu et al. (2018); Guo et al. (2018a;b); Liu et al. (2018b); Nguyen (2018); Petsiuk (2018); Wang et al. (2018b); Yang et al. (2018a); Annasamy & Sycara (2019); Arya et al. (2019); Brahimi et al. (2019); DeYoung et al. (2019); Fong et al. (2019); Kanehira et al. (2019); Kapishnikov et al. (2019); Lin et al. (2019); Pope et al. (2019); Schlegel et al. (2019); Serrano & Smith (2019); Wagner et al. (2019); Yuan et al. (2019); Ayush et al. (2020); Brunke et al. (2020); Carton et al. (2020); Chen et al. (2020); Cong et al. (2020); Fan et al. (2020); Hsieh et al. (2020); Nguyen et al. (2020); Pan et al. (2020b); Rieger & Hansen (2020); Schlegel et al. (2020); Schulz et al. (2020); Singh et al. (2020); Wang et al. (2020a); Warnecke et al. (2020); El Shawi et al. (2021); Ge et al. (2021); Jethani et al. (2021); Jung & Oh (2021); Liu et al. (2021b); Luss et al. (2021); Poppi et al. (2021); Singh et al. (2021); Situ et al. (2021); Sun et al. (2021); Velmurugan et al. (2021a;b); Vu et al. (2021); Wang et al. (2021); Yin et al. (2021); Albini et al. (2022); Atanasova et al. (2022); Dai et al. (2022); De Silva et al. (2022); Funke et al. (2022); Hameed et al. (2022); Müller et al. (2022); Ngai & Rudzicz (2022); Rong et al. (2022); Tan et al. (2022); Veerappa et al. (2022); Šimić et al. (2022); Zou et al. (2022); Agarwal et al. (2023); Alangari et al. (2023b); Jin et al. (2023); Schlegel & Keim (2023); Awal & Roy (2024)

A common and versatile strategy for evaluating explanation quality is to assess how perturbing features based on the explanans affects model predictions. This approach captures both the necessity (correctness) and sufficiency (completeness) of features highlighted by an explanans (Carton et al., 2020; DeYoung et al., 2019; Alangari et al., 2023b). Details on perturbation strategies are outlined in Subsection C.2.1.

**Perturbation Scope:** Perturbations can be applied either in a fixed-size or iterative manner. Fixed-size perturbations change a predefined amount of information at once[a]. The amount of removed features can be selected as a fixed number[b] or based on a threshold over cumulative importance scores[c]. Iterative perturbation gradually removes features one by one or in batches[d].

**Perturbation Order:** The order in which features are perturbed is critical. Most relevant first (MoRF) evaluates correctness: removing the most important features should lead to a sharp performance drop if the explanation correctly highlights necessary features[e]. Least relevant first (LeRF) tests completeness: performance should remain high if unimportant features are removed first[f]. Alternatively, some authors reverse the entire process by starting from a blank input and adding features incrementally[g], which is functionally equivalent, with MoRF addition corresponding to LeRF deletion, and vice-versa (for an illustration see Figure 7).

**Measure Score:** The performance change is measured using various scoring mechanisms. These include raw prediction difference $\theta(x, \hat{y}) - \theta(\dot{x}, \hat{y})$[h], normalized or relative variants[i], or evaluation of loss and accuracy metrics[j]. Some works use statistical distances such as Kullback-Leibler divergence

(Liu et al., 2021b; Agarwal et al., 2023), Kendall's Tau (Singh et al., 2020; 2021), or Pearson correlation (Poppi et al., 2021).

**Aggregation:** For iterative perturbations, the results can be aggregated as the average performance change (Samek et al., 2016; Ancona et al., 2017), the area over the perturbation curve (AOPC) (Samek et al., 2016; Brahimi et al., 2019; DeYoung et al., 2019; Rieger & Hansen, 2020), or the area under it (AUPC) (Petsiuk, 2018; Ngai & Rudzicz, 2022; Šimić et al., 2022; Jin et al., 2023). The area between MoRF and LeRF (ABPC) may also be computed (Nguyen et al., 2020; Schulz et al., 2020), optionally with decay weighting to emphasize early changes (Šimić et al., 2022).

**Baselines:** To contextualize results, explanation can be compared against various baselines, including random feature selections[k], zero-inputs (Schulz et al., 2020), or naïve explanation such as edge detectors (Hooker et al., 2019). Advanced setups may compare against models trained on random labels or with inserted irrelevant features to bound the quality of explanation (Hameed et al., 2022).

**Variants:**

- Evaluating explanantia across all classes per instance, not only the predicted class (Pope et al., 2019; Awal & Roy, 2024).
- Sanity checks verifying whether the perturbation curve behaves monotonically (Arya et al., 2019; Fong et al., 2019; Luss et al., 2021), or whether MoRF always performs worse than LeRF (Šimić et al., 2022).
- Normalizing the performance drop by the input difference to mitigate distribution shift: $\frac{\delta\left(\theta(x),\theta(\dot{x})\right)}{\delta(x,\dot{x})}$ (Ge et al., 2021; Schlegel & Keim, 2023).
- Plotting performance over remaining entropy rather than number of perturbed features (Kapishnikov et al., 2019).
- Training with randomized feature masking to improve model stability, although this may reduce causality fidelity (Jethani et al., 2021).

**Alternatives:** Further, we may change the perspective through alternative formulations:

- Counting the minimum number of features needed to change the model's prediction (in MoRF deletion) (Nguyen, 2018) or using differences between class-specific explanantia to guide perturbations (Shrikumar et al., 2017).
- Treating perturbed inputs as counterfactuals and applying metrics such as Minimality (see Metric 4) (Fan et al., 2020; Ge et al., 2021; Albini et al., 2022).
- Replacing fixed perturbations with adversarial optimization over $k$ features to evaluate minimum change necessary for altering predictions (Hsieh et al., 2020; Vu et al., 2021).

The approach may be extended to CEs by either mapping concepts to features (e.g., concept-based saliency maps by Lucieri et al. (2020)), or perturbing internal activations at the concept layer (El Shawi et al., 2021).

---

[a]Guo et al. (2018b); DeYoung et al. (2019); Warnecke et al. (2020); Jethani et al. (2021); Jung & Oh (2021); Velmurugan et al. (2021a;b); Wang et al. (2021); De Silva et al. (2022); Agarwal et al. (2023)

[b]Bach et al. (2015); Samek et al. (2016); Ancona et al. (2017); Chu et al. (2018); Schlegel et al. (2019); Nguyen et al. (2020); Alangari et al. (2023b); Schlegel & Keim (2023)

[c]Brahimi et al. (2019); Pope et al. (2019); Schlegel et al. (2019); Ngai & Rudzicz (2022); Schlegel & Keim (2023)

[d]Bach et al. (2015); Samek et al. (2016); Ancona et al. (2017); Petsiuk (2018); DeYoung et al. (2019); Rieger & Hansen (2020); Jin et al. (2023)

[e]Bach et al. (2015); Samek et al. (2016); Chu et al. (2018); DeYoung et al. (2019); Pope et al. (2019); Schlegel et al. (2019); Carton et al. (2020); Rieger & Hansen (2020); Ngai & Rudzicz (2022); Alangari et al. (2023b); Jin et al. (2023); Schlegel & Keim (2023)

[f]Bach et al. (2015); Ancona et al. (2017); Dabkowski & Gal (2017); Guo et al. (2018b); Annasamy & Sycara (2019); DeYoung et al. (2019); Wagner et al. (2019); Carton et al. (2020); Singh et al. (2020); Wang et al. (2020a); Jethani et al. (2021); Liu et al. (2021b); Singh et al. (2021); Wang et al. (2021); Dai et al. (2022); De Silva et al. (2022); Alangari et al. (2023b)

[g]Petsiuk (2018); Arya et al. (2019); Kapishnikov et al. (2019); Luss et al. (2021); Situ et al. (2021); Sun et al. (2021); Atanasova et al. (2022); Funke et al. (2022); Tan et al. (2022)

[h]Bach et al. (2015); Chu et al. (2018); DeYoung et al. (2019); Kapishnikov et al. (2019); Chen et al. (2020); Cong et al. (2020); Schulz et al. (2020); Ngai & Rudzicz (2022)

[i]Chattopadhay et al. (2018); Kapishnikov et al. (2019); Schulz et al. (2020); Jung & Oh (2021); Velmurugan et al. (2021a;b); Ngai & Rudzicz (2022)

[j]Bach et al. (2015); Chu et al. (2018); Kapishnikov et al. (2019); Lin et al. (2019); Schlegel et al. (2019); Serrano & Smith (2019); Wagner et al. (2019); Warnecke et al. (2020); Sun et al. (2021); Wang et al. (2021); De Silva et al. (2022); Schlegel & Keim (2023)

[k]Samek et al. (2016); Schlegel et al. (2019); Schlegel & Keim (2023); Serrano & Smith (2019); Jin et al. (2023)

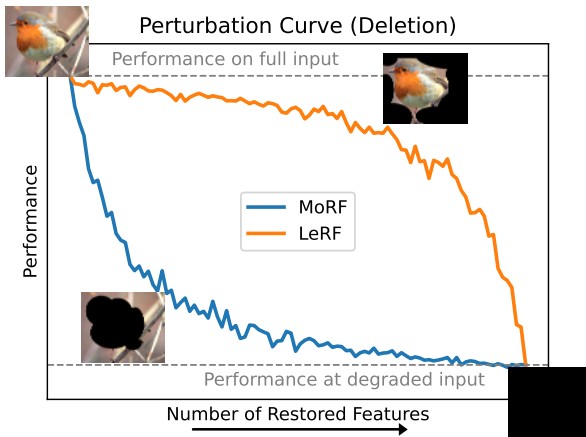 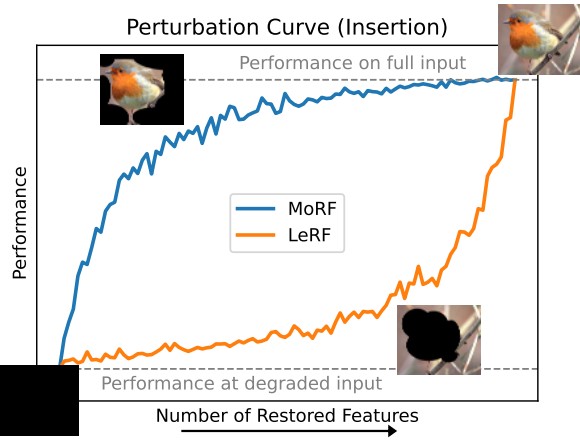

(a) Deletion perturbs features iteratively.     (b) Insertion restores features iteratively.

Figure 7: Examples for ideal Perturbation Curves to evaluate feature attributions. Insertion and Deletion approaches are equivalent when switching MoRF for LeRF ordering and vice versa. Deletion MoRF (Insertion LeRF) can be used to evaluate correctness, and Insertion MoRF (Deletion LeRF) to evaluate completeness.

---

**Metric III.29: Counterfactuability**

**Desiderata:** Fidelity
**Explanation Type:** WBS                                                   **References:** (1)
Pornprasit et al. (2021)

To assess the expressiveness and impact of rule-based explanations (either generated directly or extracted from surrogate models such as trees), we can use them to guide the generation of counterfactual perturbations. This evaluates whether the rules have predictive leverage and reflect real decision logic, rather than being purely descriptive or spurious.

Specifically, given an input instance that is covered by a rule, Pornprasit et al. (2021) perturb the input such that the rule conditions are violated. If the rule truly captures important rationale, breaking it should influence the black-box model's prediction. This can be quantified in two ways:

- By counting the number of perturbed instances that change the predicted class label.
- By measuring the aggregate change in predicted class probability before and after perturbation.

---

**Metric III.30: Prediction Neighborhood Continuity**

**Desiderata:** Fidelity, Continuity
**Explanation Type:** WBS                                                   **References:** (1)
Lakkaraju et al. (2020)

A WBS is only useful if its behavior remains consistent with the black-box even under slight perturbations to the input.

To assess this, Lakkaraju et al. (2020) propose measuring the difference in Output Faithfulness (see Metric 22) between the original and perturbed inputs. Specifically, for each input a perturbed variant is created (e.g., through noise or adversarial modification). The metric then compares the agreement between the black-box model and the surrogate model before and after the perturbations

A lower difference indicates a more robust surrogate, as it preserves faithfulness across perturbations. High differences may signal that the surrogate captures only superficial model behavior or overfits to specific input patterns.

---

**Metric III.31: Quantification of Unexplainable Features**

**Desiderata:** Continuity, (Fidelity)
**Explanation Type:** FA, (CE)                                    **References:** (2)
Zhang et al. (2019a); Chen et al. (2022)

---

To assess the continuity of explanations, Zhang et al. (2019a) perturb the unimportant features of an input based on the explanans, and then generate a new one for the perturbed input. The similarity between the original and perturbed explanantia serves as a measure of continuity: a high similarity suggests that irrelevant changes do not affect the explanation, indicating robustness. In addition, it implies a high completeness, as relevant features must have been captured well enough to maintain the model's behavior despite noise.

A special case of this approach is the Attack Capture Rate proposed by Chen et al. (2022), which applies to FAs in NLP (particularly rationale extraction). Here, insertion attacks introduce distractor phrases to the input, which ideally should not influence the rationale. The metric measures how often the inserted tokens appear in the extracted explanans. This variant relies on artificial attacks, which limits its generalizability beyond NLP.

This metric may be extended to CE by perturbing the concept layer corresponding to unimportant concepts, or by mapping concepts back to input features via concept-based localization (e.g., see Lucieri et al. (2020)).

---

**Metric III.32: Neighborhood Continuity**

**Desiderata:** Continuity
**Explanation Type:** FA, ExE, WBS, NLE, (CE)                     **References:** (19)
Alvarez-Melis & Jaakkola (2018a;b); Chu et al. (2018); Honegger (2018); Yeh et al. (2019); Zhang et al. (2019a); Artelt & Hammer (2020); Fan et al. (2020); Lakkaraju et al. (2020); Artelt et al. (2021); Bajaj et al. (2021); Situ et al. (2021); Yin et al. (2021); Agarwal et al. (2022a); Atanasova et al. (2022); Fouladgar et al. (2022); Agarwal et al. (2023); Bayrak & Bach (2023a); Tekkesinoglu & Pudas (2024)

---

Neighborhood Continuity evaluates whether explanantia remain similar for similar explananda. The underlying assumption is that small changes in the input should not lead to disproportionately large differences in the output, thereby increasing user trust through Continuity.

The main distinction between proposed metrics lies in how "similar instances" are defined. Some classic metrics include *Local Lipschitz Stability* from Alvarez-Melis & Jaakkola (2018a;b), *Sensitivity* from Yeh et al. (2019), and *RIS/RRS/ROS* from Agarwal et al. (2022a).

The **choice of neighborhood** depends on the domain and explanation use case. Some approaches define neighborhoods using fixed-radius input distances or $k$-nearest neighbors (Chu et al., 2018; Situ et al., 2021; Fouladgar et al., 2022; Tekkesinoglu & Pudas, 2024), while others restrict similarity to instances sharing the same predicted label (Honegger, 2018; Fan et al., 2020).

To **generate similar inputs**, perturbation-based strategies are often used (see Subsection C.2.1). Perturbations may be bounded in magnitude[a] or derived from domain-specific semantics (Zhang et al., 2019a; Yin et al., 2021; Fouladgar et al., 2022). Some metrics require that perturbed inputs

preserve the original prediction $\hat{y}_x$ (Alvarez-Melis & Jaakkola, 2018a;b; Agarwal et al., 2022a), or maintain similar logits $\theta(x)$ (Agarwal et al., 2023).

**Explanans similarity** is typically calculated using distance or correlation-based metrics (see Subsection C.2.3) and may be normalized by input distance (Alvarez-Melis & Jaakkola, 2018a;b; Agarwal et al., 2022a), or based on distances in model activation space (Agarwal et al., 2022a; 2023).

**Variations**: Fan et al. (2020) compare similarity for nearby vs. distant inputs , while Zhang et al. (2019a) restrict comparisons to unchanged features to reduce noise from perturbations.

Although originally proposed for FAs, this concept is broadly transferable to other explanation types. For instance, Lakkaraju et al. (2020) evaluate WBSs by comparing surrogate models trained on perturbed data, using model-specific similarity measures such as coefficient mismatch for linear models.

---

[a]Alvarez-Melis & Jaakkola (2018a;b); Yeh et al. (2019); Bajaj et al. (2021); Agarwal et al. (2022a); Atanasova et al. (2022)

---

**Metric III.33: Adversarial Input Resilience**

**Desiderata:** Continuity

**Explanation Type:** FA, (ExE, CE, WBS, NLE)     **References:** (10)

Singh et al. (2018); Wang et al. (2018b); Chen et al. (2019b); Dombrowski et al. (2019); Ghorbani et al. (2019); Subramanya et al. (2019); Boopathy et al. (2020); Kuppa & Le-Khac (2020); Zhang et al. (2020); Huang et al. (2023a)

Explanations can be vulnerable to Adversarial Attacks (AA), where the goal is to manipulate either the explanans (Ghorbani et al., 2019) or prediction (Singh et al., 2018; Wang et al., 2018b; Subramanya et al., 2019). More sophisticated approaches aim to manipulate one, while additionally restricting changes to the other. Two main types exist:

- **Explanans manipulation**: The explanans is altered while the prediction remains fixed (Dombrowski et al., 2019; Boopathy et al., 2020; Kuppa & Le-Khac, 2020; Huang et al., 2023a).
- **Prediction manipulation**: The prediction changes while the explanans remains similar (Zhang et al., 2020; Huang et al., 2023a).

Evaluation typically measures the distance between the adversarial output (explanans or prediction) and either its original or targeted counterpart. The input change is usually bounded. Performance is reported either as aggregated distance metrics (Dombrowski et al., 2019; Ghorbani et al., 2019; Zhang et al., 2020; Huang et al., 2023a) or as attack success rates based on predefined thresholds (Kuppa & Le-Khac, 2020; Zhang et al., 2020; Huang et al., 2023a).

While the reported authors focus on FAs, the approach can be easily adapted to other explanation types by selecting appropriate similarity measures.

### C.3.IV   Model Intervention

For some categories of metrics, it is necessary to change the underlying model, e.g., through retraining or weight randomization.

---

**Metric IV.34: Model Parameter Randomization Test**

**Desiderata:** Fidelity

**Explanation Type:** FA, (ExE, CE, WBS, NLE)     **References:** (5)

Adebayo et al. (2018); Kindermans et al. (2019); Binder et al. (2023); Bommer et al. (2024); Hedström et al. (2024)

To verify that explanations are truly reflective of the black-box model's learned reasoning, the model's internal parameters are systematically randomized, and the resulting changes in explanantia are

analyzed. The rationale is that if an explanation remains unchanged under randomization, it is likely generic and not informative of the model's decision logic (Adebayo et al., 2018).

Most commonly, the weights of a neural network are randomized either entirely, by layer, or iteratively in top-down or bottom-up order. The similarity between the original and randomized explanantia is then computed (Adebayo et al., 2018; Hedström et al., 2024). Similarity may be evaluated using SSIM and correlation for heatmaps (Adebayo et al., 2018; Binder et al., 2023; Bommer et al., 2024), or any suitable similarity metric (see Subsection C.2.3). Hedström et al. (2024) additionally average the results across noisy inputs (e.g., via input perturbations) to stabilize local explanations.

For gradient-based methods, Sixt et al. (2020) propose inserting a random activation vector at a specific layer, to break the causal connection without modifying the network weights.

A complementary strategy proposed by Kindermans et al. (2019) keeps the weights fixed but applies a controlled shift to the input distribution, adapting the input-layer biases such that internal computations and outputs remain unchanged. Since the model behavior is invariant by design, any change in the explanans indicates unwanted sensitivity. This shift-invariance test can be quantified using standard similarity metrics between pre- and post-shift explanantia.

Although originally proposed for FAs, the approach is applicable to any explanation type for which suitable similarity metrics can be defined.

---

**Metric IV.35: Data Randomization Test**

**Desiderata:** Fidelity
**Explanation Type:** FA, (ExE, CE, WBS, NLE)                    **References:** (2)
Adebayo et al. (2018); Sanchez-Lengeling et al. (2020)

---

Explanations should highlight meaningful structures in the data, not artifacts of memorization. To verify this, the training data labels are randomized, forcing the model to fit noise rather than learn semantically relevant features (Adebayo et al., 2018). Label randomization can be applied to the full training set (Adebayo et al., 2018) or a subset only (Sanchez-Lengeling et al., 2020).

This test compares explanantia generated by a model trained on the randomized data to those from a model trained on correctly labeled data. A strong explanation method should yield low similarity between the two, as the latter explanantia reflect meaningful decision features, while the former do not. Similarity can be computed using SSIM, correlation (Adebayo et al., 2018), rank-based metrics such as Kendall's Tau (Sanchez-Lengeling et al., 2020), or other suitable measures (see Subsection C.2.3). Although only reported for FAs, this approach can be extended to any explanation type, provided that a suitable similarity measure between explanantia is available.

---

**Metric IV.36: Retrained Model Evaluation**

**Desiderata:** Fidelity
**Explanation Type:** FA                                        **References:** (10)
Guo et al. (2019); Hooker et al. (2019); Cheng et al. (2020); Han et al. (2020); Schiller et al. (2020); Hemamou et al. (2021); Shah et al. (2021); Li et al. (2022); Khalane et al. (2023); Raval et al. (2023)

---

A key limitation of perturbation-based evaluation methods lies in their potential to introduce distribution shifts though the alteration of features. To mitigate this, two related strategies retrain models on the perturbed datasets.

The first approach is most famously known as Remove and Retrain (ROAR) by Hooker et al. (2019), with related variants proposed by Han et al. (2020) and Shah et al. (2021). It builds on the perturbation strategies of Metric 28, but ROAR removes features (e.g., the most important ones according to the explanans) from the training data and then retrains the model from scratch on the altered dataset. Performance degradation of this retrained model, compared to the original model, is then used to infer the quality of the explanation: a faithful explanans should identify features whose removal severely affects performance.

The second strategy leverages explanations for knowledge distillation. Here, various authors train a new model on a dataset reduced to only the features deemed important by the explanation[a].

Evaluation follows the general structure of Metric 28, measuring how much predictive performance is retained when only the relevant features are used.

While both ROAR and distillation-based setups are valuable for benchmarking explanation methods in a more robust setting, they do not directly assess the original explanans for a specific instance. Instead, they evaluate the utility of the explanation method by measuring the average informativeness of the features it selects.

---

[a]Guo et al. (2019); Cheng et al. (2020); Schiller et al. (2020); Hemamou et al. (2021); Li et al. (2022); Khalane et al. (2023); Raval et al. (2023)

---

### Metric IV.37: Influence Fidelity

**Desiderata:** Fidelity
**Explanation Type:** ExE                                    **References:** (2)
Krishnan & Wu (2017); Guo et al. (2020)

To evaluate the fidelity of ExEs, the model is retrained on a dataset modified according to the explanation. The effect of these modifications on model predictions is used to assess the explanatory quality.

Guo et al. (2020) remove the identified influential training points from the dataset, retrain the model, and measure the change in loss for the explanandum. If the removed instances were truly helpful, model performance should degrade. Conversely, if they were misleading, removal should lead to an improvement. Thus, a greater change in loss signals a more correct and complete set of influential instances.

In an alternative approach, Krishnan & Wu (2017) retain the identified influential instances but randomly flips the labels of all remaining training examples before retraining. If the explanation captures all relevant information, the prediction should remain stable. Hence, lower prediction variance after retraining indicates a more complete and accurate explanans.

---

### Metric IV.38: Normalized Movement Rate

**Desiderata:** Continuity
**Explanation Type:** FA                                     **References:** (2)
Salih et al. (2022; 2024b)

To assess the robustness of FAs in the presence of collinear or redundant features, Salih et al. (2022) introduce a metric that evaluates the stability of feature rankings as the most important features are iteratively removed and the model is retrained. This procedure mirrors the setup of the Remove and Retrain paradigm (Metric 36), but shifts the focus from prediction performance to the behavior of the explanation itself.

After each retraining step, the explanans is recomputed, and the ranks of the remaining features are compared to the previous ones. A large shift in ranking indicates that redundant or weakly relevant features have taken over the role of more informative ones, suggesting that the attribution method lacks robustness in the face of redundancy and may not reflect the true rationale behind the model's prediction.

To address this issue, Salih et al. (2024b) propose the Modified Informative Position (MIP), which aims to stabilize the interpretation by providing a more resilient ranking structure across iterative retraining steps.

**Metric IV.39: Adversarial Model Resilience**

**Desiderata:** Continuity
**Explanation Type:** FA, (ExE, CE, WBS, NLE)    **References:** (4)
Heo et al. (2019); Pruthi et al. (2019); Viering et al. (2019); Dimanov et al. (2020)

Explanation methods should depend meaningfully on the internal parameters of the black-box model. However, this sensitivity can be exploited to adversarially manipulate them. Specifically, small but carefully chosen changes to the model's weights can alter the generated explanations without significantly changing predictions. This manipulation can serve to obfuscate undesirable behaviors or bias within a model.

To assess the robustness of explanation methods against such attacks, various strategies perturb the model parameters and observe the resulting changes in explanantia. These attacks can be:

- *Targeted*, where the modified model is encouraged to produce explanantia similar to a predefined target (Heo et al., 2019; Viering et al., 2019).
- *Untargeted*, where the goal is to produce explanantia that differ substantially from the original ones (Heo et al., 2019; Pruthi et al., 2019; Dimanov et al., 2020).

To preserve prediction behavior, constraints may be imposed on the model modification. Dimanov et al. (2020) bound the change in prediction scores , but other restrictions are thinkable, such as limiting the weight difference norm to ensure the model remains close to the original.

The success of the manipulation is measured depending on the attack setup. For targeted attacks, its the similarity between the manipulated explanans and the predefined target (Viering et al., 2019). Untargeted attacks are evaluated using the dissimilarity between the manipulated and original explanans (Dimanov et al., 2020; Pruthi et al., 2019).

All approaches were originally introduced for FA, but by applying suitable similarity measures (see Subsection C.2.3), they can be extended to other explanation types as well.

### C.3.V    A Priori Constrained

The final group of metrics is those that have additional requirements to the desideratum, namely the necessity for specific annotations in the dataset or using a specific type of models.

**Metric V.40: GT Dataset Evaluation**

**Desiderata:** Plausibility, Fidelity
**Explanation Type:** FA, CE, NLE    **References:** (119)
**Human-Annotated Dataset** (69): Simonyan et al. (2013); Cao et al. (2015); Lapuschkin et al. (2016); Zhou et al. (2016); Das et al. (2017); Fong & Vedaldi (2017); Selvaraju et al. (2017); Bargal et al. (2018); Baumgartner et al. (2018); Camburu et al. (2018); Chen et al. (2018a); Chuang et al. (2018); Jha et al. (2018); Liu et al. (2018a); Park et al. (2018); Poerner et al. (2018); Wang et al. (2018a); Wu & Mooney (2018); Zhang et al. (2018a); Bastings et al. (2019); Chen et al. (2019d;c); DeYoung et al. (2019); Fong et al. (2019); Kanehira & Harada (2019); Kanehira et al. (2019); Mitsuhara et al. (2019); Puri et al. (2019); Rajani et al. (2019); Shu et al. (2019); Sydorova et al. (2019); Taghanaki et al. (2019); Trokielewicz et al. (2019); Verma & Ganguly (2019); Wang & Nvasconcelos (2019); Wickramanayake et al. (2019); Zeng et al. (2019); Bass et al. (2020); Cheng et al. (2020); Li et al. (2020a); Liu et al. (2020); Nam et al. (2020); Pan et al. (2020a); Rio-Torto et al. (2020); Schulz et al. (2020); Subramanian et al. (2020); Sun et al. (2020); Wang & Vasconcelos (2020); Xu et al. (2020a;b); Bany Muhammad & Yeasin (2021); Barnett et al. (2021); Bykov et al. (2021); Jang & Lukasiewicz (2021); Joshi et al. (2021); Mathew et al. (2021); Nguyen et al. (2021); Wiegreffe & Marasović (2021); Asokan et al. (2022); Cui et al. (2022); Du et al. (2022); Mücke & Pfahler (2022); Theiner et al. (2022); Nematzadeh et al. (2023); Rasmussen et al. (2023); Ribeiro et al. (2023); Tritscher et al. (2023); Atanasova (2024b); Saifullah et al. (2024)
**Synthetic Dataset** (50): Cortez & Embrechts (2013); Chen et al. (2017); Oramas et al. (2017); Ross et al. (2017); Chen et al. (2018b); Kim et al. (2018); Mascharka et al. (2018); Yang et al. (2018b); Antwarg et al. (2019); Arras et al. (2019); Camburu et al. (2019); Ismail et al. (2019); Jia et al. (2019); Lin et al. (2019); Subramanya et al. (2019); Takeishi (2019); Yang & Kim (2019); Ying et al. (2019); Amiri et al. (2020); Ismail et al. (2020); Jia et al. (2020); Kohlbrenner et al. (2020); Lucieri et al. (2020); Luo et al. (2020); Nguyen & Martínez (2020); Tritscher et al. (2020); Wang et al. (2020a); Bohle et al. (2021); Faber et al. (2021); Kim et al. (2021); Lin et al. (2021); Liu et al. (2021a); Shah et al. (2021); Yalcin et al. (2021); Agarwal et al. (2022b); Amoukou et al. (2022); Arias-Duart et al. (2022); Arras et al. (2022); Fan et al. (2022); Khakzar et al. (2022); Rao et al. (2022); Tjoa & Guan (2022); Wilming et al. (2022);

Zhou et al. (2022); Agarwal et al. (2023); Hesse et al. (2023); Miró-Nicolau et al. (2023); Sun et al. (2023); Ya et al. (2023); Rao et al. (2024)

To evaluate the quality of explanations, many authors propose comparing them to dataset-based ground truths. This strategy can follow two distinct paradigms, depending on how the ground truth is derived. If the rationale is uniquely defined through the data generation process, we may evaluate the explanation's **Fidelity**. If, however, the ground truth stems from human judgment or heuristic annotation, we evaluate the **Plausibility** of the explanation.

The former is feasible with synthetic datasets that encode precisely one explanatory rationale. The latter is more common in practice but inherently less reliable, as black-box models may learn spurious correlations as shown by Ribeiro et al. (2016) or identify valid rationales beyond those provided by humans (Mücke & Pfahler, 2022; Ya et al., 2023).

**Synthetic Datasets** are constructed such that the relationship between input features and labels is controlled and known. Ideally, these datasets admit only a single valid rationale. Although this assumption is not always met in practice[a]. The ground-truth explanantia may either be explicitly specified, e.g., by indicating the features responsible for prediction[b], or derived from the data-generation process itself, for instance using gradients or Shapley values[c]. Figure 8 presents selected examples.

Typical strategies for synthetic data include:

- Creating inputs from structured primitives (e.g., objects or shapes), where the target labels are deterministic functions of those primitives[d].
- Inserting backdoor triggers into inputs, which act as the decisive explanans for altered predictions[e].
- Performing adversarial attacks restricted to known feature subsets, which then serve as the explanatory evidence for prediction changes (Subramanya et al., 2019).
- Generating labels as noise-free functions over randomly sampled input features, commonly in tabular domains[f].
- Constructing mosaic images or input grids where the rationale corresponds to a localized sub-region of the input[g].

Advantages of synthetic datasets include reduced distribution shift when applying perturbation-based metrics (Metric 28) (Ismail et al., 2019; 2020; Hesse et al., 2023), support for concept-level manipulations (Yang & Kim, 2019; Lin et al., 2021; Hesse et al., 2023), and the inclusion of meaningful counterfactuals or rationale variants (Yang & Kim, 2019).

**Human-Annotated Datasets** instead provide plausible rationales grounded in human intuition or derived from proxy labels. In some cases, annotators are explicitly asked to justify their decisions or to highlight which features they consider relevant for a particular prediction (Chen et al., 2019c; Xu et al., 2020b; Cui et al., 2022; Tritscher et al., 2023). In other settings, rationales are implicit, where existing annotations such as segmentation maps, bounding boxes, or other metadata are repurposed to approximate human reasoning[h]. These datasets are typically used to evaluate the plausibility of explanations rather than their fidelity, as the ground-truth provided reflects human expectations rather than the actual decision-making logic of the model.

**Evaluation**:

Across both paradigms, the central assumption is that accurate model predictions imply alignment with the intended rationale, thereby justifying a comparison between the generated explanantia and the ground-truth annotations. This comparison can be performed using any suitable similarity metric (see Subsection C.2.3). While FA evaluations often rely on feature-level scoring, CEs and NLEs require domain-appropriate measures.

In the context of **FAs**, several dedicated evaluation strategies were reported. Precision and recall can be computed over truly important features to assess how well the explanation captures relevant inputs (Bastings et al., 2019). Other techniques evaluate the ranking of important and unimportant features (Chen et al., 2017; 2018b; Antwarg et al., 2019; Camburu et al., 2019), or quantify the symmetric difference between the sets of selected and annotated features (Nguyen & Martínez, 2020). A

particularly common metric involves summing the attribution values over known important features, often with normalization or weighting adjustments[i].

For **visual and natural language domains**, the well-known *Pointing Game* checks whether the most highly attributed features lie within predefined ground-truth regions such as bounding boxes or key tokens[j]. Closely related is *Weakly Supervised Localization*, where the explanans is compared directly to segmentation masks or bounding boxes, typically using Intersection-over-Union (IoU)[k]. In multi-label or multi-object scenarios, further alignment can be tested by verifying whether the explanation focuses on the input features associated with the predicted label (Du et al., 2019).

For **CEs**, evaluation typically involves test datasets that explicitly annotate the presence or absence of each concept, enabling precise assessment of concept identification accuracy (Kim et al., 2018; Lucieri et al., 2020; Asokan et al., 2022).

For **NLEs**, comparisons to reference justifications rely on general natural language processing measures such as BLEU or ROUGE[l], as well as similarity measures specifically designed for NLEs (Park et al., 2018; Du et al., 2022)

Finally, to ensure that the evaluation isolates explanation quality from predictive accuracy, Fan et al. (2022) recommend to limit evaluation to those samples for which the model's prediction is both correct and sufficiently confident.

---

[a]Kim et al. (2018); Arras et al. (2019); Yang & Kim (2019); Tritscher et al. (2020); Faber et al. (2021)

[b]Chen et al. (2017); Oramas et al. (2017); Ross et al. (2017); Chen et al. (2018b); Yang et al. (2018b); Ying et al. (2019); Luo et al. (2020); Faber et al. (2021); Kim et al. (2021); Tjoa & Guan (2022); Wilming et al. (2022); Zhou et al. (2022)

[c]Cortez & Embrechts (2013); Jia et al. (2019; 2020); Liu et al. (2021a); Amoukou et al. (2022)

[d]Oramas et al. (2017); Ross et al. (2017); Kim et al. (2018); Yang et al. (2018b); Ying et al. (2019); Lucieri et al. (2020); Luo et al. (2020); Faber et al. (2021); Kim et al. (2021); Yalcin et al. (2021); Khakzar et al. (2022); Tjoa & Guan (2022); Agarwal et al. (2023); Hesse et al. (2023); Miró-Nicolau et al. (2023)

[e]Lin et al. (2019; 2021); Fan et al. (2022); Sun et al. (2023); Ya et al. (2023)

[f]Cortez & Embrechts (2013); Chen et al. (2017; 2018b); Ismail et al. (2019); Jia et al. (2019); Amiri et al. (2020); Ismail et al. (2020); Jia et al. (2020); Nguyen & Martínez (2020); Tritscher et al. (2020); Liu et al. (2021a); Agarwal et al. (2022b); Amoukou et al. (2022)

[g]Bohle et al. (2021); Shah et al. (2021); Arias-Duart et al. (2022); Rao et al. (2022; 2024)

[h]Selvaraju et al. (2017); Kanehira et al. (2019); Wang & Nvasconcelos (2019); Cheng et al. (2020); Saifullah et al. (2024)

[i]Lapuschkin et al. (2016); Yang & Kim (2019); Kohlbrenner et al. (2020); Nam et al. (2020); Rio-Torto et al. (2020); Wang et al. (2020a); Xu et al. (2020a); Bohle et al. (2021); Kim et al. (2021); Arias-Duart et al. (2022); Arras et al. (2022); Zhou et al. (2022)

[j]Bargal et al. (2018); Poerner et al. (2018); Zhang et al. (2018a); Fong et al. (2019); Sydorova et al. (2019); Taghanaki et al. (2019); Takeishi (2019); Schulz et al. (2020); Barnett et al. (2021); Arras et al. (2022); Theiner et al. (2022)

[k]Simonyan et al. (2013); Cao et al. (2015); Zhou et al. (2016); Fong & Vedaldi (2017); Selvaraju et al. (2017); Wickramanayake et al. (2019); Bany Muhammad & Yeasin (2021); Nguyen et al. (2021); Fan et al. (2022)

[l]Camburu et al. (2018); Chuang et al. (2018); Liu et al. (2018a); Wu & Mooney (2018); Chen et al. (2019d); Rajani et al. (2019); Wickramanayake et al. (2019); Li et al. (2020a); Sun et al. (2020); Jang & Lukasiewicz (2021); Wiegreffe & Marasović (2021); Ribeiro et al. (2023); Atanasova (2024b)

---

## Metric V.41: White Box Model Check

**Desiderata:** Fidelity

**Explanation Type:** FA, (ExE, NLE)                                    **References:** (12)

Ribeiro et al. (2016); Antwarg et al. (2019); Dhurandhar et al. (2019); Jia et al. (2019); Zhou et al. (2019); Crabbe et al. (2020); Jia et al. (2020); Guidotti (2021); Velmurugan et al. (2021a); Dai et al. (2022); Brandt et al. (2023); Carmichael & Scheirer (2023)

---

In white-box models, the internal logic is fully accessible and interpretable, providing a ground-truth rationale against which generated explanantia can be directly evaluated. Typical white-box models used for comparison include linear regressors (Crabbe et al., 2020; Dai et al., 2022), feature-additive models (Carmichael & Scheirer, 2023), small neural networks with manually set parameters (Antwarg et al., 2019; Brandt et al., 2023), or symbolic models such as decision trees (Ribeiro et al., 2016; Dhurandhar et al., 2019).

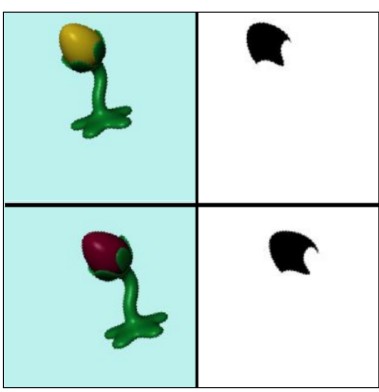

(a) The an8Flower dataset (Oramas et al., 2017).
Left: Input images of flowers, with differing petals. Right: Ground-Truth explanans of relevant region.

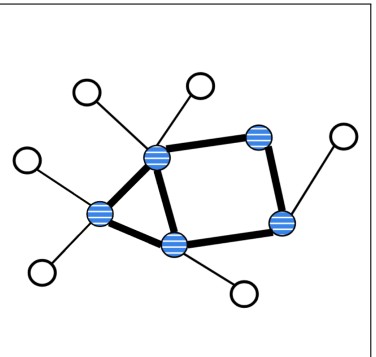

(b) The ShapeGGen dataset (Agarwal et al., 2023).
The graph's label depends on relevant structures or motifs (e.g. house-shape) that serve as ground-truth explanans.

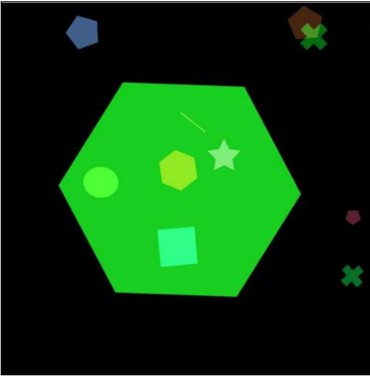

(c) The SCDB dataset (Lucieri et al., 2020).
The label is defined by the combination of small objects inside the are of the larger one and serve as ground-truth concepts.

Figure 8: Examples of synthetic datasets containing ground-truth explanantia of relevant features.

> Given that the model's reasoning is known, explanation methods can be assessed by comparing their output against this ground-truth explanans. This can be done using general similarity metrics (Guidotti, 2021; Jia et al., 2019) or using error metrics like MSE (Crabbe et al., 2020; Dai et al., 2022; Brandt et al., 2023).
>
> Further, if the white-box model relies only on a constrained feature subset, one can also measure explanation fidelity through accuracy, precision, or recall between explanans and the truly influential features (Ribeiro et al., 2016; Zhou et al., 2019; Jia et al., 2020; Velmurugan et al., 2021a).
>
> Although originally proposed for FAs, this approach may be extended to ExEs and NLEs, as the white-box model enables verification of whether the provided explanantia are consistent with the known reasoning. While user inspection is straightforward, adapting this check into an automatic, functionality-grounded evaluation remains challenging but potentially feasible.

## D   Excluded Metrics

We excluded a total of four metrics from our framework. While initially identified during our literature review, we later dismissed them after closer inspection. Some do not evaluate the explanation itself, but rather the model or downstream tasks, while others rely on assumptions that are overly narrow or reward properties we consider misleading. For transparency, we briefly describe each and illustrate our reasons for exclusion below.

- **Alignment**: Etmann et al. (2019) assess plausibility by measuring how well a saliency map aligns with the original input, using correlation as a proxy. However, we argue that input reconstruction does not necessarily enhance interpretability, as simply returning the input or an edge map offers little explanatory value. Furthermore, the metric mixes explanation with input similarity, rewarding explanation that may lack selectivity or meaningful abstraction. Therefore, we exclude this metric due to its flawed underlying assumption.
- **Explanatory Power** Arras et al. (2017) evaluate the model–explanation pair by computing $k$-nearest-neighbor accuracy on document vectors constructed from word-attribution explanation. While the score reflects how semantically useful these explanations are for related tasks, it measures downstream utility rather than explanation quality. As such, it evaluates a combination of

model and explanation performance, not the fidelity (or other desideratum) of the explanation itself. Further, it is not applicable beyond text domain or in scenarios without downstream tasks.

- **Feature Diversity**: Smyth & Keane (2022) calculate the fraction of different features altered across all counterfactuals in a dataset, where low diversity implies the same features are consistently changed. However, this appears to be a purely statistical property of the explanation, and it remains unclear why higher or lower diversity should indicate better explanation quality. Without a clear link to any desideratum, its practical value as an evaluation metric is doubtful.

- **Mutual Information**: Le et al. (2020) measure how independent the features changed in counterfactuals are, with the goal of encouraging maximal expressiveness. It uses Symmetrical Uncertainty (Press, 2007) to quantify pairwise dependencies among changed features. However, enforcing independence between altered features can easily lead to implausible counterfactuals; especially when high correlations reflect natural dependencies. Breaking these can result in unrealistic or invalid inputs.

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
