# OpenReview forum: "Unifying VXAI: A Systematic Review and Framework for the Evaluation of Explainable AI"
_TMLR — Accepted by TMLR_

### Review · Reviewer_bbUY · 2025-08-26

**Summary Of Contributions:**

This paper presents a systematic literature review of functionality-grounded metrics for evaluating explainable AI (XAI) methods, following PRISMA guidelines. The authors identify 362 relevant publications and aggregate their contributions into 41 functionally similar metric groups. The main contributions are: (1) a comprehensive systematic review identifying evaluation metrics from 362 sources, (2) aggregation of these into 41 metric groups based on methodological similarity, (3) a three-dimensional categorization scheme spanning explanation type, evaluation contextuality, and explanation quality desiderata, and (4) a unified framework (VXAI) claimed to be the most comprehensive structured overview to date.

The paper proposes seven desiderata for XAI evaluation organized into Technical (Coverage, Fidelity, Continuity, Consistency, Efficiency) and Interpretability (Parsimony, Plausibility) dimensions. The contextuality dimension ranges from explanans-centric (level I) to a priori constrained evaluation (level V). Five explanation types are considered: Feature Attributions, Concept Explanations, Example Explanations, White-Box Surrogates, and Natural Language Explanations.

**Audience:**

Yes

**Broader Impact Concerns:**

The paper appropriately addresses how misleading evaluation metrics can either reduce trust in well-functioning models or reinforce trust in flawed ones

**Claims And Evidence:**

Yes

**Requested Changes:**

I think the survey is solid and acceptable according to TMLR guidelines as is, but for more impact I'd recommend:
- **Specify the criteria** used to group individual metrics into 41 categories. If multiple reviewers participated, report inter-rater agreement. If single-reviewer, describe the systematic approach used.
- **Include concrete recommendations** for metric subset selection given specific research contexts. The comprehensive framework could benefit from decision heuristics to become practically useful.
- **Improve visual accessibility**. Use larger font sizes in figures where possible. The extensive gray boxes make large portions difficult to read. Consider breaking complex horizontal figures into multiple focused subfigures for better readability.
- **Reorganize structure**. Move technical details like exact search queries to appendices. Promote results and substantive content to improve readability.

**Strengths And Weaknesses:**

## Strengths
- **Systematic** review methodology is exemplary, following PRISMA guidelines with transparent search strategies and inclusion criteria. Impressive coverage of 362 sources is one of the most comprehensive literature syntheses in XAI evaluation to date.
	- the two-stage approach (database search + recursive snowballing) is sound
	- work spans the entire landscape of functionality-grounded XAI evaluation
- **Addresses a genuine need** - the lack of standardized evaluation approaches in XAI; XAI metrics are often embedded within method papers rather than dedicated evaluation studies.
- Three-dimensional categorization is **conceptually clear**, distinction between technical and interpretability dimensions is intuitive, and the contextuality levels offer useful gradations
	- contextuality dimension is particularly valuable, clearly distinguishing between in-situ and ex-situ evaluation
- **Good scoping** - focusses on functionality-grounded metrics, while acknowledging complementary human-centered evaluation approaches exist.
- **Thoughtful critiques** of ground-truth-based evaluation and perturbation methods


## Weaknesses

- No empirical validation of the proposed taxonomy. The framework remains theoretical without evidence that the categorization helps practitioners or that metrics within categories behave similarly.
- Motivation for five-level contextuality scale seems ad-hoc - the paper does not establish why these particular levels represent the most useful distinctions.
- **Limited practical guidance**. With 41 metric groups across multiple dimensions, the framework may be too complex for practical adoption. No guidance on metric subset selection: practitioners would profit from guidance on prioritizing for specific use cases.
	- similarly, guidance on how to combine metrics measuring different desiderata would be useful, building on the brief discussion of aggregation approaches
- **Aggregation methodology lacks detail**, the process for grouping individual metrics into 41 categories is not clearly specified.
- **Missing concrete research agenda** beyond general calls for standardization. What specific gaps in evaluation methodology does this review reveal? What specific organizational or conceptual advantages does VXAI provide over existing approaches?


## Minor

**Visual accessibility**:

- Use larger font sizes where possible, especially in figures (tables may be constrained by size requirements).
- The red-green color scheme is not color-blind friendly, though since it's not used for distinguishing information this is likely acceptable.
- The extensive gray boxes make large portions of the paper visually heavy and difficult to read.
- Consider breaking complex figures into multiple focused subfigures. The horizontal format is not ideal - Figure 3 could flip rows and columns to be readable without turning one's head, and Figure 6 could be broken into multiple subfigures.

**Structure and readability**:

- Reading would be more enjoyable if technical decisions such as inclusion criteria and exact queries were moved to the appendix, making results and content more prominent.
- The paper generally uses clear terminology but occasionally switches between "explanation quality" and "explanation evaluation".
- Many sentences could be more direct. For example, "metrics are provided" could become "we provide metrics." Academic language often hides the author behind passives, making text harder to read.
- The paper frequently uses phrases like "In contrast" and "However" at paragraph starts even when the contrast isn't strong - some can be removed for better flow.

---

> ### Author Response · Authors · 2025-11-06
> **Reply to Reviewer bbUY**
>
> We thank the reviewer for the positive review and address the raised issues as follows:
>
> ---
>
> ### **Structure of paper**
> - **Move Method to Appendix:** All details about the literature review will be moved to the appendix, leaving only a concise outline in the main text. The section will briefly note that we conducted a systematic survey following PRISMA guidelines, focused on identifying papers introducing new original methods, and report the total number of reviewed articles.
>
> ---
>
> ### **Practical guidance**
> - **Add information to Discussion:** We will include a new paragraph or subsection titled “Authors’ Advice” offering guidance on metric selection (fixing Explanation Type and defining relevant Desiderata). We advocate for more research on when and how to choose suitable metrics. While no universal guidelines exist yet, we encourage applying available metrics rather than trusting explanations blindly. We further recommend using multiple metrics and varying their hyperparameters. Even without benchmarks, explanations can be compared against naïve or random baselines to gauge whether they perform better, as each metric indicates whether higher or lower values are preferable.
>
> ---
>
> ### **Highlight evaluation gaps and advantage of VXAI**
> - **Comment:** Research gaps are already covered in the Discussion (e.g., lack of *Portability* metrics under *Efficiency*, and the dominance of *Feature Attribution* metrics over *Natural Language Explanations*). We also stress the need to extend existing metrics to “potentially applicable” explanation types and revisit these gaps in the Conclusion.
> - **Add statement in Conclusion:** We will highlight that the VXAI framework not only provides the most comprehensive metric collection to date but also introduces a detailed, extensible categorization scheme. The framework is horizontally expandable (new dimensions, desiderata, explanation types, contextualities) and vertically extendable (new metrics), supporting future adaptation and continuous growth.
>
> ---
>
> ### **Details for aggregation method**
> - **Add detail to Method:** We will clarify that many identified low-level metrics were similar or minor variants (e.g., differing only in hyperparameters). These were grouped into higher-level aggregated metrics to ensure a coherent and interpretable overview.
>
> ---
>
> ### **Empirical evaluation of VXAI framework**
> - **Comment:** An empirical validation would be valuable but lies beyond this work’s scope. Evaluating whether the framework “helps practitioners” would require a controlled user study, which risks bias and would further expand the manuscript. We thus deliberately keep the work theoretical, expecting its practical impact to emerge through community adoption.
>
> ---
>
> ### **Visual improvements**
> - **Larger fonts in figures:**
>   - *XAI Desiderata (F3):* Increase font size and reduce gray heaviness.
>   - *Tables 1 & 2:* Improve readability where margins allow.
>   - *Metric statistics (F5):* Increase text size.
>   - *MoRF/LeRF (F7):* Enlarge text and place panels side by side.
> - **Green–red color scheme:**
>   - The distinction is not critical; full recoloring is unnecessary. We verified accessibility using a color-blindness checker but welcome further input.
> - **Gray boxes:**
>   - Maintain the structure (each box represents a metric) but reduce gray intensity for better readability.
> - **Complex figures:**
>   - *Figure 3 (XAI Desiderata):* Rotate for easier reading.
>   - *Figure 6:* Rearrangement under consideration.
>
> ---
>
> ### **Clarify terminology and language**
> - **General update to language:** We will polish the manuscript for clarity and directness. Terminology will be used consistently: *evaluation* refers to the process of assessing explanation quality. If this distinction was unclear, we are happy to adapt phrasing accordingly.
>
> ---
>
> We hope to address all open questions and criticisms raised by the reviewer and look forward to the continued exchange.

---

> > ### Comment · Reviewer_bbUY · 2025-11-10
> > **Overall good**
> >
> > I appreciate the concrete commitments to restructuring the paper, improving visual accessibility, and adding practical guidance through an "Authors' Advice" section.
> >
> > Moving technical details to the appendix while promoting substantive results is easier to read. The commitment to larger fonts and reduced gray boxing should address my concerns.
> >
> > Regarding empirical validation, I think the improvements (Author's Advice) you've committed to should strengthen the work's accessibility for practitioners. I understand the practical constraints of conducting user studies within manuscript scope. The addition of concrete metric selection heuristics ("fix explanation type and define relevant desiderata") moves the framework closer to actionable guidance.
> >
> > The clarifications on aggregation methodology provide useful context and discussion of evaluation gaps and the VXAI framework's extensibility should make this work a solid reference for standardizing XAI evaluation practice.
> >
> > Overall, I believe your revisions adequately address the presentation and guidance concerns.

---

### Review · Reviewer_2n19 · 2025-09-07

**Summary Of Contributions:**

The paper is a systematic review of evaluation metrics for explainable AI and presents a unified framework for the evaluation of XAI algorithms called VXAI. Specifically, 362 papers were aggregated into 41 functionally-similar metric groups that were then categorized based on (1) the desiderata each group contributes to; (2) the degree of dependency on the model and data; (3) the explanation types the metric group applies to.

**Audience:**

Yes

**Claims And Evidence:**

Yes

**Requested Changes:**

As per the weaknesses above, I think the main requested changes are several clarifications for the setting and methodology of the paper:
- Clarification of the method section as per my questions above
- Clarification regarding models that are interpretable by design
- Clarification regarding the scope of XAI algorithms covered by this review paper

**Strengths And Weaknesses:**

Strengths:
- An up-to-date and very comprehensive systematic review on the topic of evaluation of explainable AI
- The systematic review follows the Preferred Reporting Items for Systematic Reviews and Meta-Analyses (PRISMA) guidelines
- The paper is clear and justifications for most design choices in the systematic reviews are provided

---

Weaknesses:
- Despite serving as a very comprehensive systematic review, the majority of insights (e.g., in Section 6) remains on the summary level and does not provide non-trivial higher-level insight on the field such as identifying important gaps in the literature or directions for future work. I don’t think this is critical for a systematic review, but it could have increased the potential value and impact of such paper.

- I found the Method section (summarized in Figure 4) a bit confusing. Specifically, I would appreciate some clarification about the following questions:
	* The step of database search has eventually lead to only 6 papers included in the study, while the citation steps has led to 317 included paper. This raise the question of the effectiveness of the database search (including the databases chosen, the keywords used, etc.)
	* The manual search procedure is not clearly explained in the paper and is not very transparent at the moment: first, google scholar is likely to include all the papers that were retrieved in the database search. Second, I am surprised that google has only retrieved 65 references. How were they filtered? This is critical to clarify this as it seems like perhaps this step may be more effective than the database search.
	* It is not clear how 160 papers from database search lead to 515 citations while 65 papers from the manual search have only lead to 20 citations: is this after removing duplicates found in the citations from the database search?
	* How preprints were treated? Was ArXiv used to search for papers? Were papers that only appear on ArXiv (or other preprint servers) but not published in peer-review venues included in the study?

- I was not sure how models that are interpretable by design (decision trees, linear regression, etc.) whose global explanation is the model itself are handled and categorized. For example, what is the explanation type in such case? White box surrogate does not make sense as these are not surrogate models.

- Scope: it seems like the emphasis is on evaluating XAI algorithms for tasks like classification and regression, and not for generation tasks (text generation, image generation from text, etc) but I did not see it clearly defined in the paper.

---

> ### Author Response · Authors · 2025-11-06
> **Reply to Reviewer 2n19**
>
> We thank the reviewer for the positive and constructive feedback and address the raised issues as follows:
>
> ---
>
> ### **Evaluation of interpretable-by-design models**
> - **Add clarification in Introduction:** We focus on in-hoc and post-hoc explanations *for black-box models* alike and therefore do not consider directly interpretable (white-box) models. However, many of the presented metrics can also be applied to such models, depending on their formulation.
> - **Add clarification to Explanation Types:** The VXAI framework is agnostic to whether a specific explanation is integrated into the model (e.g., Attention Mechanisms or Concept Bottleneck Models) or derived *post-hoc* from it.
> - **Add clarification to Discussion:** Although not the main focus of this work, many of the identified metrics can be extended to white-box models that are interpretable by design. The main exception are metrics under the *Fidelity* desideratum, as fidelity measures the alignment between explanans and explanandum, which presupposes a black-box reference. In such cases, the *Fidelity* desideratum could be replaced by a *Performance* desideratum, using standard metrics such as accuracy or mean squared error.
>
> ---
>
> ### **Scope applicable to Classification/Regression or more**
> - **Add clarification to Discussion:** Most XAI algorithms are designed with discriminative tasks such as classification in mind. However, the metrics within the VXAI framework are general enough to also apply to explanations generated for other types of tasks, including regression or generative models. The framework itself focuses on the evaluation through these metrics, which operate around the explanation rather than the specific task type. Certain adaptations may be required—for instance, metrics that rely on accuracy or similar measures would need to use appropriate task-dependent alternatives.
>
> ---
>
> ### **Motivation for Contextuality scale**
> - **Add details to Categorization Scheme:** We will expand the explanation of the Contextuality dimension and clarify its rationale. Contextuality helps practitioners select appropriate metrics and interpret their results, as it captures the scope of dependency each metric assumes. Since all metrics are covered by these Contextualities, we consider the taxonomy complete for now.
> - **Add details to Discussion:** We will also highlight that evaluating some desiderata (e.g., *Fidelity* or *Continuity*) requires broader contextuality, which increases methodological complexity and reduces comparability across metrics.
>
> ---
>
> ### **Limited insights, research gaps, and future work**
> - **Add information to Discussion:** We will include a new paragraph or subsection titled “Authors’ Advice.” We strongly advocate for more research on suitable metrics and for practical guidance on when to choose which. While such standards are not yet available, we encourage researchers to apply existing metrics rather than trust explanations blindly. We also recommend using multiple metrics with varied hyperparameters and comparing results against random baselines, since for each metric it is known whether higher or lower scores indicate improvement.
> - **Comment:** We believe key research gaps are already discussed in the paper, such as the absence of a metric for *Portability* under *Efficiency*, or the strong focus on *Feature Attribution* while neglecting other explanation types such as *Natural Language Explanations*. We also stress the need to extend existing metrics to “potentially applicable” explanation types and reiterate these directions in the Conclusion.
> - **Comment:** We acknowledge the request for more in-depth metric discussion but emphasize that doing so in the main text would exceed length limits. Detailed formulations, categorizations, and citations remain accessible in the appendix.
>
> ---
>
> ### **Systematic Review methodology**
> - **Add details to Method:** When moving the method to the appendix, we will expand on the multi-stage process. Stage 1 focused on identifying secondary literature, while stage 2 snowballed into primary works. Manual searches via Google Scholar and recommendations supplemented the structured database search; duplicates were excluded. Relevant preprints were included only if referenced by other sources.
> - **Add details to Method:** We will clarify that numerous minor or hyperparameter-based metric variants were grouped into higher-level categories to enhance interpretability and avoid redundancy.
>
> ---
>
> We hope these clarifications address all raised concerns and improve the manuscript’s clarity and completeness.

---

### Review · Reviewer_NeRQ · 2025-11-01

**Summary Of Contributions:**

This paper conducts a systematic review of the evaluation of black-box models' explanations. The review covers 362 relevant publications and aggregates the proposed evaluation metrics into 41 groups. They further propose a three-dimensional scheme spanning desiderata, contextuality, and explanation types. The contributions of these 41 groups to the three dimensions are comprehensively discussed.

**Audience:**

Yes

**Claims And Evidence:**

Yes

**Requested Changes:**

See weakness/question part.

**Strengths And Weaknesses:**

**Strength**
- The contribution of this work to the XAI community is strong and timely. With the rapid growth of XAI methods and algorithms, definitions of a good explanation and a rigorous evaluation framework remain unclear. This systematic and structured review provides a unified evaluation framework that helps bridge this gap.
- The literature curation for both related review works and publications proposing new metrics is rigorous and transparent, following the PRISMA guidelines. I’m impressed by the wide coverage and clear organization, and I appreciate the authors’ diligence and the substantial amount of work invested.
- The three-dimensional scheme is both novel and useful. XAI is intrinsically complex, with diverse methods, goals, contexts, and stakeholders. These complexities all propagate to downstream evaluation — there are no simple, universal metrics. I’m glad to see the authors tailor evaluation metrics to different aspects and provide a comprehensive yet clean structure.
- I like many of the insights and comments in the paper. Here are some examples:
```
* Although evaluation has different desiderata, there is a hierarchy and priority among them. Fidelity is the foundation of the whole evaluation process.
* Different models can have different reasoning, even generating the same or similar outputs. There is no reason to provide consistent explanations across models even if they behave the same.
* Some explanation methods embed evaluation metrics as part of their algorithm, which raises concerns about circularity.
* There is also no reason to assume the human rationale and the model rationale are aligned, which challenges many human-centered evaluations.
```

**Weakness/questions**
- Evaluation of interpretable-by-design models: The paper exclusively focuses on evaluating post-hoc explanations for black-box models and suggests using interpretable models if their performance is comparable. This seems to implicitly assume that an interpretable-by-design model automatically produces faithful explanations. I’m not sure I fully agree with this, especially regarding grey-box models. For instance, an attention mechanism can generate saliency maps, but there are ongoing debates about whether attention weights truly reflect model reasoning. The concept bottleneck model is also considered interpretable-by-design, yet there are significant concerns about the faithfulness of its concept scores and potential information leakage. From my point of view, evaluation of interpretable-by-design models is also necessary, and many of the metrics covered in this paper could be adapted to them. I look forward to seeing the authors’ opinions and comments on this point.

- Structure of the paper: The paper is long and dense, which is fine for a systematic review. However, it becomes less reader-friendly by not presenting the methods and main results until around page 14. The presentation of the three dimensions is highly unbalanced: desiderata are explained in great detail (spanning seven pages), whereas the other two dimensions, explanation type and contextuality, are much more briefly described, composing only three pages in total. This is especially noticeable for contextuality, where each level is introduced in only a few sentences. I’m not sure this is sufficient to clarify the meaning and rationale of each grouping. The main text also provides no room — even briefly — to explain the 41 groups of metrics, presenting only their statistics. I’m not convinced this is the optimal way to organize the content.

---

> ### Author Response · Authors · 2025-11-06
> **Reply to Reviewer NeRQ**
>
> We thank the reviewer for the positive feedback and want to address the raised issues as follows:
> ---
> ### **Evaluation of interpertable-by-design models**:
> - **Add clarification in Introduction:** We focus on in-hoc and post-hoc explanations for black-box models alike and therefore do not consider directly interpretable (white-box) models. However, many of the presented metrics can also be applied to such models, depending on their formulation.
> - **Add clarification to Explanation Types:** The VXAI framework is agnostic to whether a specific explanation for a black-box model is integrated into the model (e.g., Attention Mechanisms or Concept Bottleneck Models) or derived post-hoc from it.
> - **Add clarification to Discussion:** Although not the main focus of this work, many of the identified metrics can be extended to white-box models that are interpretable by design. The main exception are metrics under the Fidelity desideratum, as fidelity measures the alignment between explanans and explanandum, which presupposes a black-box reference. In such cases, the Fidelity desideratum could be replaced by a Performance desideratum, using standard metrics such as accuracy or mean squared error.
> ---
> ### **Structure of the paper**:
> - **Move Method to Appendix:** All details about the literature review will be moved to the appendix, leaving only a short outline of the process in the main text. The main section will briefly state that we performed a systematic survey following the PRISMA guidelines, aiming to identify papers that introduced a new original method, and the number of reviewed articles.
> - **Move Desiderata of XAI to Appendix**: Consider moving the detailed descriptions of the XAI desiderata to the appendix. In the main text (Results, Categorization Scheme), only list the desiderata briefly as currently done and refer readers to the appendix for their full explanations.
> - **Add details to Categorization Scheme:** We will provide more detail on the Contextuality dimension and clarify its rationale. The Contextuality dimension is intended to help practitioners select appropriate metrics and interpret their results, as it captures a key technical property of each metric. Since all metrics are covered by these Contextualities, we consider it also complete  (for now).
> - **Add details to Discussion**: We will also emphasize this aspect in the Discussion section, highlighting one of the central limitations of current metrics: evaluating certain desiderata (e.g., Fidelity or Continuity) requires a broader contextuality, which makes reliable evaluation more difficult.
> ---
> ### **No discussion of metrics in text**:
> - **Comment:** While we recognize the reviewers’ interest in more detailed discussion of individual metrics, providing such detail beyond the information summarized in the tables would substantially increase the manuscript length. For this reason, the in-depth descriptions of all metrics remain available in the appendix, where their formulations, categorizations, and specific references are presented in full.
>
> We hope to address all open quesitons and critizisms brought up by the reviewer and are looking forward to the exchange.

---

### Decision · Action_Editor_xSzD · 2025-12-15

**Recommendation:** Accept as is

**Audience:**

Yes

**Audience Explanation:**

All expert reviewers agree that the proposed unified framework for the eValuation of XAI (VXAI) will be valuable to the XAI community and have unanimously recommended acceptance. This consensus strongly suggests that the findings of this paper will be of significant interest to TMLR’s audience.

**Claims And Evidence:**

Yes

**Claims Explanation:**

The paper presents a comprehensive survey on metrics and evaluation methods for Explainable AI (XAI). The authors follow the Preferred Reporting Items for Systematic Reviews and Meta-Analyses (PRISMA) guidelines when conducting such survey, ensuring systematicity and broad coverage of the literature.

---

> ### Author Response · Authors · 2026-01-06
> **Camera Ready submitted**
>
> We have submitted the camera-ready version of the manuscript.
> We would like to sincerely **thank** the Action Editor and the Editors-in-Chief for their support throughout the review process. We are also very grateful to the reviewers for their constructive and insightful feedback, which substantially improved the quality and clarity of the paper.
> As the date, we have indicated 01/2026 and would be happy to adjust it if necessary.